# HESS Opinions: Incubating deep-learning-powered hydrologic science advances as a community

Chaopeng Shen[1], Eric Laloy[2], Amin Elshorbagy[3], Adrian Albert[4], Jerad Bales[5], Fi-John Chang[6],
Sangram Ganguly[7], Kuo-lin Hsu[8], Daniel Kifer[9], Zheng Fang[10], Kuai Fang[1], Dongfeng Li[10], Xiaodong
Li[11], and Wen-Ping Tsai[1]

*1. Civil and Environmental Engineering, Pennsylvania State University, University Park, PA 16802*
*2. Institute for Environment, Health and Safety, Belgian Nuclear Research Centre, Mol, Belgium*
*3. Dept. of Civil, Geological, and Environmental Engineering, University of Saskatchewan, Saskatoon, Canada*
*4. Energy Technologies Area, Lawrence Berkeley National Laboratory, Berkeley, CA 94720*
*5. Consortium of Universities for the Advancement of Hydrologic Science, Inc. (CUAHSI), Cambridge, MA*
*6. Department of Bioenvironmental Systems Engineering, National Taiwan University, Taipei, 10617, Taiwan*
*7. NASA Ames Research Center/ BAER Institute, Moffett Field, CA 94035*
*8. Civil and Environmental Engineering, University of California, Irvine, Irvine, CA 92697*
*9. Computer Science and Engineering, Pennsylvania State University, University Park, PA 16802*
*10. Civil Engineering, University of Texas at Arlington, Arlington, TX 76013*
*11. State Key Laboratory of Hydraulics and Mountain River Engineering, Sichuan University, Sichuan, China*

*Correspondence to*: Chaopeng Shen (cshen@engr.psu.edu)

**Abstract.** Recently, deep learning (DL) has emerged as a revolutionary and versatile tool transforming industry applications and generating new and improved capabilities for scientific discovery and model building. The adoption of DL in hydrology has so far been gradual, but the field is now ripe for breakthroughs. This paper suggests that DL-based methods can open a complementary avenue toward knowledge discovery in hydrologic sciences. In the new avenue, machine-learning algorithms present competing hypotheses that are consistent with data. Interrogative studies are then invoked to interpret DL models for scientists to further evaluate. However, hydrology presents many challenges to for DL methods, such as data limitations, heterogeneity and co-evolution, and the general inexperience of the hydrologic field with DL. The roadmap toward DL-powered scientific advances will require the coordinated effort from a large community involving scientists and citizens. Integrating process-based models with DL models will help alleviate data limitations. The sharing of data and baseline models will improve the efficiency of the community as a whole. Open competitions could serve as the organizing events to greatly propel growth and nurture data science education in hydrology, which demands a grass-root collaboration. The area of hydrologic DL present numerous research opportunities which could, in turn, stimulate advances in machine learning as well.

## 1. Overview

Deep learning (DL) is a suite of tools centered on artfully designed large-size artificial neural networks. The deep networks at the core of DL are said to have "depth" due to their multi-layered structures, which help deep networks represent abstract concepts about the data (Schmidhuber, 2015). Given input attributes that describe an instance, deep networks can be trained to make predictions of some dependent variables, either continuous or categorical, about this instance. For example, for standard computer vision problems, deep networks can recognize the theme or objects from a picture (Guo et al., 2016; He et al., 2016; Simonyan and Zisserman, 2014) or remotely sensed images (Zhu et al., 2017). For sequential data, DL can associate natural language sequence to commands (Baughman et al., 2014; Hirschberg and Manning, 2015) or predict the action of an actor in the next video frame (Vondrick et al., 2016). DL can also generate (or synthesize) images that carry certain artistic styles (Gatys et al., 2016) or a natural language response to questions (Leviathan and Matias, 2018; Zen and Sak, 2015). With the support of deep architectures, deep networks can automatically engineer relevant concepts and features from large datasets, instead of requiring human experts to define these features (Section 2.2.2). As a foundational component of modern artificial intelligence (AI), DL has made substantial strides in recent years and helped to solve problems that have resisted AI for decades (LeCun et al., 2015).

While DL has stimulated exciting advances in many disciplines and has become the method of choice in some areas, hydrology so far have only had a very limited set of DL applications (Shen, 2018) (hereafter referred to as Shen18). Despite scattered reports of promising DL results (Fang et al., 2017; Laloy et al., 2017, 2018; Tao et al., 2016; Vandal et al., 2017; Zhang et al., 2018), hydrologists have not widely adopted these new tools. This collective opinion paper argues that there are many opportunities in hydrological sciences where DL can help provide both stronger predictive capabilities and a complementary avenue toward scientific discovery. We then reflect on why it has been challenging to harness the power of DL and big data in hydrology and explore what we can do as a community to incubate progress. Readers who are less familiar with machine learning or deep learning are referred to a companion review paper (Shen18), which provides a more comprehensive and technical background than this opinion paper. Many details behind the arguments in Section 2 are provided in Shen18.

## 2. The emergence of a complementary research avenue

We are witnessing the growth of three pillars needed for DL to support a research avenue that is complementary to traditional hypothesis-driven research: big hydrologic data, powerful machine learning algorithms, and interrogative methods (such as visualization and  techniques) to extract interpretable knowledge from the trained networks. This new avenue starts from data, uses DL methods to generate hypotheses, and applies interrogative methods to help us understand hydrologic system functioning. We discuss these aspects in the following sections.

## 2.1. With more data, opportunities arise

The fundamental supporting factor for emerging opportunities with DL is the growth of big hydrologic data, with all surface, sub-surface, urban, infrastructure, and ecosystem dimensions. In this paper, hydrology refers to both the complete natural and engineered water cycle, and associated processes in the ecosystem and geologic media. There are ever increasing amounts of hydrologic data available through remote sensing (see a summary in Srinivasan, (2013)) and data compilations. For example, satellite-based datasets include precipitation, surface soil moisture (Entekhabi, 2010; Jackson et al., 2016; Mecklenburg et al., 2008), vegetation states and indices, e.g., (Knyazikhin et al., 1999), and derived evapotranspiration products (Mu et al., 2011), terrestrial water storage (Wahr et al., 2006), snowcover (Hall et al., 2006), and a planned mission for estimating streamflows (Pavelsky et al., 2014), etc. On the data compilation side, there are now compilations of geologic (Gleeson et al., 2014) and soil datasets; centralized management of streamflow and groundwater data in the United States, Europe, parts of South America and Asia, or globally for some large rivers (GRDC, 2017); water chemistry, groundwater samples and other biogeophysical datasets. The Consortium of Universities for the Advancement of Hydrologic Science, Inc. (CUAHSI) operates two systems for the discovery and archival of water data: the Hydrologic Information System (CUAHSI, 2018c) for time series, and HydroShare for all water data types (Horsburgh et al., 2016). An Internet of Water (Aspen, 2017) has been proposed and is beginning to develop, thereby improving access to these emerging data sets.

Moreover, unconventional data sources are starting to emerge. High-resolution sensing of Earth will be provided by increasing amount of CubeSats, Unmanned Aerial Vehicles, balloons, inexpensive photogrammetric sensing and many other sources (McCabe et al., 2017). These new sources provide new forms of measurements not envisioned before. For example, cell phone signal strength and cell-phone pictures can contribute to high resolution monitoring of rainfall intensity (Allamano et al., 2015). Inexpensive infrared camera images can detect water levels in complex urban water flows (Hiroi and Kawaguchi, 2016). Internet-of-Things (IoT) sensors embedded in water infrastructure can transmit data about the states of water in our environment (Zhang et al., 2018). These new sources of information provided unprecedented volumes and multi-faceted coverages of the natural and built environment. However, since each new data source has its own characteristics and peculiarities, the identification of the appropriate approaches to fully exploit their value, especially synergistically, creates a significant challenge. In contrast, DL models can be built, without significant human expertise and extensive manual labour, to rapidly derive useful information from these data.

## 2.2. DL: A big step forward

### 2.2.1. Rapid adoption

The field of hydrology has witnessed flows and ebbs of several generations of machine learning methods in the past few decades. From regularized linear regression (Tibshirani and Tibshirani, 1994) to Support Vector Regression (Drucker et al., 1996), from genetic programming (Koza, 1992) to artificial neural networks (Chang et al., 2014; Chen et al., 2018; Hsu et al.,

1995, 1997, 2002), from classification and regression tree to random forest (Ho, 1995), from Gaussian Process (Snelson and Ghahramani, 2006) to Radial Basis Function Network (Moradkhani et al., 2004), each approach offered useful solutions to a set of problems, but each also faced its own limitations. As a result, over time, some may have grown dispassionate about progress in machine learning, while some others may have concerns about whether DL represents real progress or is just 5 "hype."

The progress in AI brought forth by DL to various industries and scientific disciplines is revolutionary (Section 4 in Shen18) and can no longer be ignored by the hydrology community. Major technology firms have rapidly adopted and commercialized DL-powered AI (Evans et al., 2018). For example, Google has re-oriented its research priority from "mobile-first" to "AI-first" (Dignan, 2018). The benefits of these industrial investments can now be felt by ordinary users of their services such as 10 machine translation and digital assistants who can engage in conversations sounding like a human (Leviathan and Matias, 2018). Moreover, AI patents of industries and scientific disciplines grew at a 34% compound annual growth rate between 2013 and 2017, apparently after DL's breakthroughs in 2012 (Columbus, 2018). Also reported in (Columbus, 2018), more than 65% of data professionals responded to a survey indicating AI as their company's most significant data initiative for next year.

**Table 1. Number of papers returned from searches on ISI Web of Science.**

| year | DL-nonCS | DL-CS | ML-nonCS | ML-CS | DL/ML-CS | DL/ML-nonCS |
|------|----------|-------|----------|-------|----------|-------------|
| 2011 | 0 | 23 | 1068 | 1838 | 1% | 0% |
| 2012 | 15 | 25 | 1310 | 1899 | 1% | 1% |
| 2013 | 35 | 80 | 1677 | 2360 | 3% | 2% |
| 2014 | 84 | 238 | 2228 | 3050 | 8% | 4% |
| 2015 | 308 | 709 | 3074 | 4405 | 16% | 10% |
| 2016 | 841 | 1462 | 4414 | 5361 | 27% | 19% |
| 2017 | 2035 | 2723 | 6125 | 5860 | 46% | 33% |

15 *DL-CS results were obtained by searching for "Topic" (TS)="Deep Learning" AND "Research area" (SU)= "Computer Science"; ML-CS was obtained the same way as DL-CS, only that "Deep learning" was replaced by "machine learning"; DL-nonCS was obtained by TS="Deep Learning" NOT SU="Computer Science" NOT SU=education. Education was removed because entries in this category were not related to our definition of DL. There were also 19 articles in 2011 where deep learning was about education in disciplines other than SU=Education. Therefore, 19 was used as a blank value and also* 20 *subtracted from the DL-nonCS column. DL/ML-CS is ratio of DL-CS to ML-CS expressed as a percentage. DL/ML-nonCS was obtained similarly.*

DL is gaining adoption in a wide range of scientific disciplines and, in some areas, has started to substantially transform those disciplines. The fast growth is clearly witnessed from literature searches. Since 2011, the number of entries with DL as a topic 25 increased almost exponentially, showing around 100% compound annual growth rate before 2017 (Table 1). DL evolved from occupying less than 1% of machine learning (ML) entries in computer science (CS) in 2011 to 46% in 2017. This change showcases massive conversion from traditional machine learning to DL within computer science. Other disciplines lagged

slightly behind, but also experienced exponential increase. They also saw the DL/ML ratio jumping from 0% in 2011 to 33% in 2017. As reviewed in Shen18, DL has enhanced the statistical power of data in high energy physics, and the use of DL can be considered to be equivalent to a 25% increase in the experimental dataset (Baldi et al., 2015). In biology, DL has been used to predict potential pathological implications from genetic sequences (Angermueller et al., 2016). DL models fed with raw-level data have been shown to outperform those using expert-defined features when they predict high-level outcomes, e.g., toxicity, from molecular compositions (Goh et al., 2017). Just like other methods, DL may eventually be replaced by newer ones, but that is not a reason to hold out on possible progress.

Many of the abovementioned advances were driven by DL's domination in AI competitions:

- The ImageNet Challenges is an open competition to evaluate algorithms for object detection and image classification (Russakovsky et al., 2014). Topics change during each contest, and a dataset of ~14M tagged images and videos were cumulatively compiled, with convenient and uniform data access provided by the organizers. The 2010 Challenge was won by a large-scale Support Vector Machine (SVM). Convolutional Neural Network, a kind of deep network, first won this contest in 2012 (Krizhevsky et al., 2012a). This victory heralded the exponential growth of DL in popularity. Since then, and until 2017 (the last contest), the vast majority of entrants and all contest winners used CNNs, which edges out other methods by large margins (Schmidhuber, 2015).

- The IJCNN traffic sign recognition contest, which is composed of 50,000 images (48 pixels x 48 pixels), witnessed superhuman  visual recognition performance (greater than human recognition) from CNN-based methods (Stallkamp et al., 2011). CNNs also performed better than humans on recognition of cancers from medical images (Yu et al., 2016).

- The TIMIT speech corpus is a dataset that holds the recordings from 630 English speakers. LSTM-based models showed a large edge over Hidden Markov Model (HMM) results (Graves et al., 2013) in recognizing the speeches. Similarly, LSTM-based methods significantly outperformed all statistical approaches in keyword spotting (Indermuhle et al., 2012), optical character recognition, language identification, text-to-speech synthesis, social signal classification, machine translation and Chinese handwriting recognition.

- An LSTM-based speech recognition system has achieved "human parity" in conversational speech recognition on the Switchboard corpus (Xiong et al., 2016). A parallel version achieved best-known pixel-wise brain image segmentation results on the MRBrainS13 dataset (Stollenga et al., 2015). The improvement in language translation software can be witnessed by ordinary web users.

- A time-series forecasting contests, Computational Intelligence in Forecasting Competition, was won by a combination of fuzzy and exponential models in 2015 when no LSTM was present, but LSTM won the contest in 2016 (CIF, 2016).

In contrast, only a handful of applications of big data DL could be found in hydrology, but they already demonstrated great promise. Vision DL has been employed to retrieve precipitation from satellite images, where it demonstrated a materially-superior performance than earlier-generation neural networks (Tao et al., 2017, 2018). GAN was used to imitate and generate scanning images of geologic media (Laloy et al., 2018), where the authors showed realistic replication of training image patterns. Time-series deep learning network was employed to temporally extend satellite-sensed soil moisture observations (Fang et al., 2017) and was found to be more reliable than simpler methods. Time series DL rainfall-runoff models which are confined to certain geographic divisions have been created (Kratzert et al., 2018). There are also DL studies, based on smaller datasets, to help predict water flows in the urban environment (Assem et al., 2017) and water infrastructure (Zhang et al., 2018). In addition to utilizing big data, DL was able to create valuable, big datasets that could not have been otherwise possible. For example, utilizing DL, researchers were able to generate new datasets for Tropical Cyclones, Atmospheric Rivers and Weather Fronts (Liu et al., 2016; Matsuoka et al., 2017) by tracking them. Machine learning has also been harnessed to tackled the convection parameterization issue in climate modelling (Gentine et al., 2018).

*2.2.2. Technical advances*

Underpinning the powerful performance of DL are its technical advances. The deep architectures have several distinctive advantages: (1) deep networks are designed with the capacity to represent extremely complex functions. (2) After training, the intermediate layers can perform modular functions which can be migrated to other tasks, in a process called transfer learning, and extend the value of the training data. (3) The hidden layer structures have been designed to automatically extract features, which helps dramatically reducing labour, expertise and the trial and error time needed for feature engineering. (4) Compared to earlier models like classification trees, most of the deep networks are differentiable, meaning that we can calculate derivatives of outputs with respect to inputs or the parameters in the network. This feature enables highly efficient training algorithms that exploit these derivatives. Moreover, the differentiability of neural networks enables querying DL models for sensitivity analysis of outputs to input parameters, a task of key importance in hydrology.

Metaphorically, the intermediate (or hidden) layers in DL algorithms can be understood as workbenches or placeholders for tools that are to be built by deep networks themselves. These hidden layers are trained to calculate certain features from the data, which are then used by downstream layers to predict the dependent variables. For example, Yosinski (2015) showed that some intermediate layers in a deep vision recognition network are responsible for identifying the location of human or animal faces; Karpathy et al., (2015) showed that some hidden cells in a text prediction network act as length counters of a line while some others keep track of whether the text is in quotes or not. These functionalities were not bestowed by the network designers, but emerged by themselves after network training. Earlier network architectures either did not have the needed depth, or were not designed in an artful way such that the intermediate layers could be effectively trained. For more technical details, refer to an introduction in Schmidhuber (2015) and Shen (2018).

Given that deep networks can identify features without human guide, it follows that they may extract features that the algorithm designers were unaware of, or did not intentionally encode the network to do. If we could believe that there is latent knowledge about the hydrologic system that humans are not yet aware of, but can be determined from data, the automatic extraction of features leads to a potential pathway toward knowledge discovery. For example, deep networks recently showed that grid-like neuron response structures automatically emerge at intermediate network layers for a network trained to imitate how mammals perform navigation, providing strong support to a Nobel-winning neuroscience theory about the functioning of these structures (Banino et al., 2018).

Deep networks may be more robust than simpler models despite their large size, if they are regularized properly (regularization techniques apply penalty to model complexity to make the model more robust) and are chosen based on validation errors in a two-stage approach (Kawaguchi et al., 2017). Effective regularization techniques include (i) early stopping: monitor the training progress on a separate validation set and stop the training once validation metrics start to deteriorate; and/or (ii) novel regularization techniques such as dropout (Srivastava et al., 2014). DL models can be easier to train than previous networks, as their architectures and new stochastic gradient techniques (Kingma and Ba, 2014) address issues like vanishing gradient (Hochreiter, 1998). Training large networks used today was computationally implausible until scientists started to exploit the parallel processing power of graphical processing units (GPUs). Nowadays, new application-specific integrated circuits have also been created to specifically tackle DL, although DL architectures are still evolving.

Primary types of successful deep learning architectures include convolutional neural networks (CNN) for image recognition (Krizhevsky et al., 2012b; Ranzato et al., 2006), Long short-term memory (LSTM) (Greff et al., 2015; Hochreiter and Schmidhuber, 1997) for time series modeling, variational auto-encoders (VAE) (Kingma and Welling, 2013), and deep belief networks for pattern recognition and data (typically image but also text or sound, etc) generation (section 3.2 in Shen18). Besides these new architectures, a novel generative model concept called the generative adversarial network (GAN) has become an active area of research. The key characteristic of GANs is that they are learned by creating a competition between the actual generative model or ''generator'' and a discriminator in a zero-sum game framework (Goodfellow et al., 2014), in which these components are learned jointly. Compared to other generative models, GANs potentially offer much greater flexibility in the patterns to be generated. The power of GANs has been recognized recently in the geoscientific community, especially in machine learning research inspired by physics, where GANs have been used to generate certain complicated physical, environmental, and socio-economic systems (Albert et al., 2018; Laloy et al., 2018).

While showing many advantages, DL models will require substantial amount of computing expertise. The tuning of hyper-parameters, e.g. network size, learning rate, batch size, etc., often require a priori experiences and trial and error. The computational paradigm, e.g., computing on graphical processing units, is also substantially different from ordinary hydrologists' educational background. The fundamental theories on why DL generalizes so well have not been maturely developed (Section 2.7 in Shen18). In the ongoing debates, some argued that a large part of DL's power comes from

memorization while others countered that DL prioritizes learning simple patterns (Arpit et al., 2017; Krueger et al., 2017) and a two-stage procedure (training and testing) also helped (Kawaguchi et al., 2017). Despite these explanations, it has been found in vision DL that deep networks can be fooled by adversarial examples where small, unperceivable perturbations to input images sometimes cause large changes in predictions, leading to incorrect outcomes (Goodfellow et al., 2015; Szegedy et al., 2013). It remains to be seen whether such adversarial examples exist for hydrologic DL applications. If we can recreate adversarial examples, they can be added into the training dataset to improve the robustness of the model (Ororbia et al., 2016).

## 2.3. Network interrogative methods to enable knowledge extraction from deep networks

Conventionally, neural networks were primarily used to approximate mappings between inputs and outputs. The focus was put on improving predictive accuracy. In terms of the use of neural networks in scientific research, then, there have been a major concern that DL and more generally machine learning are referred to as black boxes that cannot be understood by humans and, thus, cannot serve to advance scientific understanding. At the same time, data-driven research may lack clearly-stated hypotheses which is in contrast to traditional hypothesis-driven scientific methods. There has been significant pressure from both inside and outside the DL community to make the network decisions more explainable. For example, new (as of January 2018) European data privacy laws dictate that automated individual decision making, which significantly influences the algorithm's users must provide a "right to explanation" where a user can ask for an explanation of an algorithmic decision (Goodman and Flaxman, 2016).

Some recent progress in DL research focused on addressing these concerns. Notably, a new sub-discipline, known as "AI neuroscience" has produced useful interrogative techniques to help scientists interpret the DL model (see literature in Section 3.2 in Shen18). The main classes of interpretive methods include (i) reverse-engineer the hidden layers: attributing deep network decisions to input features or a subset of inputs; (ii) transferring knowledge from deep networks to interpretable, reduced-order models. (iii) visualization of network activations. Many scientists have also devised case-by-case adhoc methods, e.g., to investigate the correlation between inputs and cell activations (Shen, 2018; Voosen, 2017).

Interpretive DL methods have so far not been employed in hydrology or even geosciences. However, to give some examples from other domains, in medical image diagnosis, some researchers used reverse engineering methods to show which pixel on an image led the network to make its decision about anatomy classifications (Kumar and Menkovski, 2016). They found that the network traced its decisions to image landmarks mostly often used by human experts. In more recent research, AI researchers trained their network to not only classify an image, but also didactically explain why the decision was made and why an image is one class instead of another (Figure 1). Extending this idea to the precipitation retrieval problem in hydrology as in (Tao et al., 2017, 2018), we could let DL inform us what features on the satellite cloud image is helpful for reducing bias in precipitation retrieval.

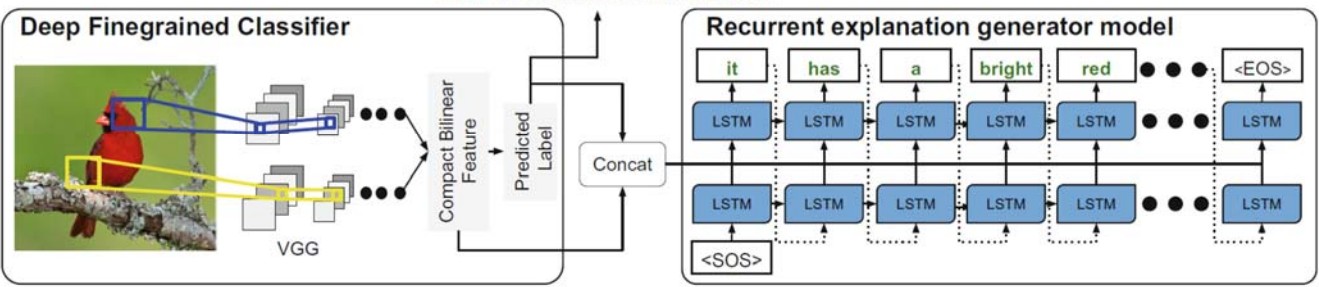

This is a **cardinal** because ...

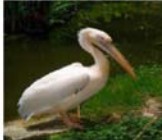

This is a **White Pelican** because...

*Description*: this bird is white and black in color with a long curved beak and white eye rings.
*Explanation-Dis.*: this is a large white bird with a **long neck** and a **large orange beak**.

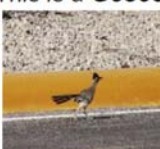

This is a **Geococcyx** because...

*Description*: this bird has a long black bill a white throat and a brown crown.
*Explanation-Dis.*: this is a black and white spotted bird with a **long tail feather** and a pointed beak.

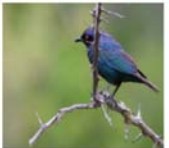

This is a **Cape Glossy Starling** because...

*Description*: this bird is blue and black in color with a stubby beak and black eye rings.
*Explanation-Dis.*: this is a blue bird with a **red eye** and a blue crown.

**Figure 1. (Reprinted from Hendricks et al., 2016 with permission) Authors trained a joint classification and explanation network for image classification. The bolded text is a "class-relevant" attribute (a distinguishing attribute for the class) in the explanation. Their classification network extracts visual features (regions on the image) responsible for the decision. Then, the explanation network links these regions to distinguishing words in a dictionary to form an explanation that explains the reason for the classification, and why it is not other classes. This level of explanation may be difficult to achieve for hydrologic problems due to limited supervising data (annotated dictionary for classes), but it is possible to borrow the idea of associating features in the input data with some descriptive words.**

## 2.4. The complementary research avenue

As the interrogative methods further grow, there emerges a research avenue toward attaining knowledge that is complementary to the traditional hypothesis-driven one (Figure 2). The data-driven research avenue can be divided into four steps: (i) hypotheses are generated by machine learning algorithms from data; (ii) the validation step is where data withheld from training, and different from training, are employed to evaluate the machine-learning-generated hypotheses; (iii) interpretive methods are employed to extract data-consistent and human-understandable hypotheses (Mount et al., 2016) (described in Section 2.3); and (iv) the retained hypotheses are presented to scientists for analysis and further data collection, and the process iterates.

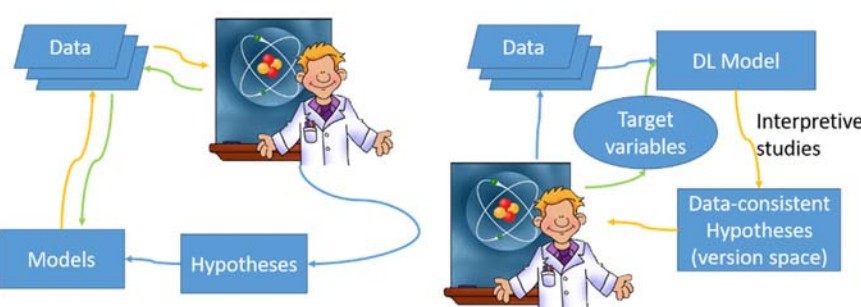

**Figure 2. Comparing two alternative avenues toward gaining knowledge from data. In the classical avenue, scientists compile and interpret data, form hypotheses, (optionally) build models to describe data and hypotheses (the green pathway). Then the model results with data to affirm or reject the hypotheses and the feedbacks (the yellow pathway) allow the scientist to revise the model**
**and iterate. In the data-driven avenue, scientists collect data and define the target variables of DL models (the green path). Then interpretive methods are invoked to extract data-consistent and human-understandable hypotheses (the yellow path). There must be a hypotheses validation step where data withheld from training is used to evaluate or reject the hypotheses.**

The classical avenue, especially when applied to modelling studies, attracts non-uniqueness and subjectivity. To give a
concrete example, consider a classical problem of rainfall-runoff modelling. Suppose a hydrologist found that hydrologic responses in several nearby basins are different. Some basins produce flashier peaks while others have smaller peaks in summer, large seasonal fluctuation and large peak streamflows only in winter. Taking a modelling approach, the hydrologist might invoke a conceptual hydrologic model, e.g., Topmodel (Beven, 1997), only to find that the model results do not adequately describe the observed heterogeneity in the rainfall-runoff response. The hydrologist might hypothesize that the
different behaviours are due to heterogeneity in soil texture, which is not well represented in the model. Subsequently, the hydrologist incorporates processes that represent soil spatial heterogeneity, such as refined soil pedo-transfer functions that can differentiate between the soil types in different regions. Perhaps with some parameter adjustment, this model can provide streamflow predictions that are qualitatively similar to the observations. This procedure then increases the hydrologist's confidence that the heterogeneity in soil hydraulic parameters is indeed responsible for their different hydrologic responses.
However, this improvement is not conclusive due to process equifinality: there can be alternative processes that can also result in similar outcomes, e.g., the influence of soil thickness, Karst geology, terrain or drainage density. The identification of potential improvement might be dependent on the hydrologist's intuition or pre-conceptions, which are nonetheless important but potentially biased. While the intention of a process-based model may be deductive (Beven, 1989), the example process given above is, in fact, abductive reasoning (Josephson and Josephson, 1994), as it seeks a plausible but not exhaustive
thorough of the phenomenon. Furthermore, incorporating all the physics into the model may prove technically challenging or too time-consuming.

Compared to the classical avenue, the inductive data-driven approach allows us to more efficiently explore a larger set of hypotheses. Although it cannot be said that the machine learning algorithms present no human bias (because inputs are human-defined and some hyper-parameters are empirically adjusted), the larger set of hypotheses presented will at least greatly reduce that risk. First, let us examine a Classification and Regression Tree algorithms-based (CART-based) data-driven approach

(Fang and Shen, 2017). We could start with physiographic data for many basins in this region, including terrain, soil type, soil thickness, etc. We can use CART to model the process-based model's errors, which allows us to separate out the conditions under which these errors occur more frequently. We let the pattern emerge out of data without enforcing a strong human pre-conceived hypothesis. Attention must be paid to the robustness of the data mining and utilize holdout dataset or cross-validation to verify the generality of the conclusion. Data may suggest that soil thickness is the main reason for the error. Or, if data do

not prefer one hypothesis over the other, then all hypotheses are equally possible and cannot be ruled out. This advantage of DL can be summarized in a short phrase, "*an algorithm has no ego.*" On a practical level, this approach can more efficiently and simultaneously examine multiple competing hypotheses.

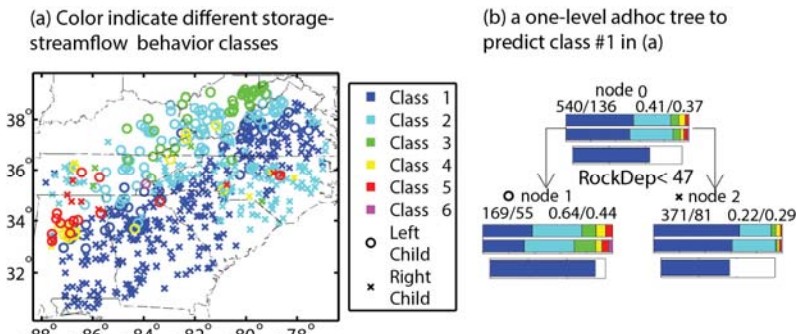

**Figure 3. (adapted from Fang and Shen 2017. Reprint permission obtained). We calculated storage-streamflow correlation patterns**
**over continental United States (CONUS) and divided small or mesoscale basins into multiple classes. We studied what physical factors most cleanly separate different correlation patterns. In this case, what separates the blue class (storage and streamflow are highly correlated across all flow regimes) and the green class turned out to be soil thickness. It suggests the blue basins in the south have high correlation because they have thick soils, which facilitates infiltration, water storage, and groundwater-dominated streamflow.**

One example of such analyses was carried out in Fang and Shen, (2017) where differences in basin storage-streamflow correlations were explained by physical factors using CART, an earlier-generation data mining method (Figure 3). The data mining analysis allowed patterns to emerge, which inspired hypotheses about key factors that control the hydrologic functioning of different systems, such as soil thickness and soil bulk density are important controls of streamflow-storage

relationships. For another example, data-mining analysis showed that drought recovery time is associated to temperature and precipitation, while biodiversity only has secondary importance (Schwalm et al., 2017). Scientists need to define the predictors and general model types, but they do not pose strongly constraining hypotheses about the controlling factors, and instead "let the data speak". The key to this approach is a large amount of data from which patterns emerge.

However, working with DL models, we need to further resort to interrogative methods to make the results understandable (Figure 2 right panel). For example, we can construct DL models to predict the errors of the process-based model, and then use visualization techniques to see which variable, under which condition, lead to the error. Because DL can absorb a large amount of data, it can find commonality among data as well as identify differences. Whereas CART models are limited by the amount of data and face stability problems in lower branches (data are exponentially less at lower branches), DL models may produce a more robust interpretation.

The machine learning paradigm lends us to finding "unrecognized linkages" (Wagener et al., 2010) or find complex patterns in the data that humans could not easily realize or capture. Owning to the strong capability of DL, it can better approximate the "best achievable model" (BAM) for the mapping relations between inputs and output. As such, it lends support to measuring the information content contained in the inputs about the output. Nearing et al., (2016) utilized Gaussian Process regression to approximate the BAM. DL can play similar roles and can also allow for modelling, perhaps in a more thorough way. The simplicity of building DL model and altering inputs makes it an ideal testbed for new ideas.

Outputs from the hidden layers of deep networks can now be visualized to gain insights about the transformations performed on the input data by the network (Samek et al., 2017). For image recognition tasks, one can invert the DL model to find out the parts of the inputs that led the network to make a certain decision (Mahendran and Vedaldi, 2015). There are also means to visualize outputs from recurrent networks, e.g., showing the conditions under which certain cells are activated (Karpathy et al., 2015). These visualizations can illustrate the relationships that the data-driven model has identified.

Considering the above potential benefits, the data-driven avenue should at least be considered or given an opportunity to play a role in hydrological sciences discovery. However, this avenue may be uncomfortable to some. In the classical avenue, the scientist must originate the hypotheses before constructing models; in the data-driven one, the data mining/knowledge discovery process is a precursor step to the main hypotheses formation-- hypotheses cannot be generated before the data mining analysis (Mount et al., 2016). Especially, hypotheses can no longer be unequivocally stated during the proposal stage of research.

Granted, the interrogative methods as a whole are new and time is required for them to grow. We need to note that the nascent "DL neuroscience" literature did not exist until 2015. However, if we outright reject the complementary avenue based on the habitual thinking that neural networks are black boxes, we may deny ourselves an opportunity for breakthroughs.

## 3. Challenges and opportunities for DL in hydrology

The field of hydrology has a unique set of challenges that are also research opportunities for DL. Many of these science challenges have, to date, not been effectively addressed using traditional methods, and cannot be sufficiently tackled by individual research groups. Some challenges for which DL approaches might be exploited are presented below.

Observations in hydrology and water science generally are regionally and temporally imbalanced. For example, while streamflow observations are relatively dense in the United States, such data are sparse in many other parts of the world, either because measurements have not been made or are not made accessible. There is often a dearth of observations that can be used as comprehensive training datasets for DL algorithms. Few hydrologic applications have as much data as the data available to standard AI research applications such as imagine recognition or natural language processing. Remote sensing of hydrologic variables also has limitations, including effects of canopy and clouds which can limit observations, temporal density of observations because of orbital paths, and observation footprints, which create challenges when trying to validate satellite observations with field point measurements. A body of literature studying this problem across different geographic regions can be loosely summarized under the topic of "prediction in ungauged basins" (PUB) (Hrachowitz et al., 2013). PUB problems pose a significant challenge to data-driven methods.

Global change is altering the hydrologic and related cycles, and hydrologists must now make predictions in anticipation of changes, beyond previously observed ranges (Wagener et al., 2010). Especially, more frequent extremes have been observed for many parts of the world and such extremes have been projected to occur more frequently in the future (Stocker et al., 2013). Data-driven methods must demonstrate their capability to make reasonable predictions when applied out of the range of the training dataset.

Observations of the water cycle tend to focus on one aspect of the water cycle, and seldom offer a complete description. For example, we can estimate total terrestrial water storage (Wahr, 2004) or top 5-cm surface soil moisture via multiple satellite missions. It is difficult, however, to directly combine such observations of components of the water cycle into a complete picture of the water cycle. A challenge, then, is merging distinct observations, with all their space-time discontinuities to aid predictions, model validation, and to provide a more complete understanding of the global water cycle.

Hydrologic data are accompanied by a large amount of strongly heterogeneous (Blöschl, 2006) "contextual variables" such as land use, climate, geology, and soil properties. The proper scale at which to represent heterogeneity in natural systems is a vexing problem (Archfield et al., 2015), as micro-scale of soil heterogeneity, for example, is not computationally realistic in hydrologic models. The scale at which heterogeneity should be represented varies with setting and elements of the water cycle (Ajami et al., 2016). Moreover, while we recognize that heterogeneity exists in contextual features, many of these features, such as soil properties and hydrogeology, are poorly characterized across landscapes, but both features play important role in controlling water movement. Heterogeneity needs to be adequately represented without radically bloating the parameter space of the models and thus data demand.

Furthermore, the heterogeneous physiographic factors co-evolve and covary (Troch et al., 2013) with complicated causal and non-causal connections (Faghmous and Kumar, 2014). The relationships of soil, terrain and vegetation are further conditioned on geologic and climate history and often do not transfer to other regions (Thompson et al., 2006). Consequently, training with insufficient data may result in overfitted data-driven models or many alternative DL models that cannot be rejected. On the

flip side, such complexity due to co-evolution also precludes a reductionist approach where all or most of these relationships are clearly described from fundamental laws.

Hydrologic problems fit poorly into the template of problems for which standard network structures (Section 3.2 in Shen18) are designed, i.e., purely image recognition or time series prediction problems, or a mixture of both. For example, catchment hydrologic problems are characterized by both spatially heterogeneous but temporally static attributes (topography and hydrogeology) and temporal (atmospheric forcing) dimensions. Such input dimensions are not efficiently represented by with typical input dimensions of LSTM or CNN.

Because large and diverse datasets are needed for DL application, access to properly pre-processed and formatted present practical challenges. These steps of data compilation, pre-processing, and formatting often occupy too much unnecessary time for researchers. Many of the processing tasks for images cannot be handled by individual research groups. Compared to the DL community in AI and chemistry, etc., DL learning community in hydrology is not sufficiently coordinated, resulting in significant waste of effort and "reinvention of wheels".

Deep generative models such as GANs can be used for the stochastic generation of natural textures. This capability has recently led to methodological advances in subsurface hydrology (Laloy et al., 2017, 2018; Mosser et al., 2017) where the ability to efficiently and accurately simulate complicated geologic structures with given (non-Gaussian) geostatistical properties is of paramount importance for uncertainty quantification of subsurface flow and transport models. However, amongst other directions for future research, more work is needed (i) to generate the complete range of structural complexity observed in geologic layers, (ii) deal efficiently with large 3D domains and (iii), account for various types of direct (e.g., observed geologic facies at a given location, mean property value over a specific area, etc) and indirect (e.g., measured hydrologic state variables to be used within an inverse modelling procedure) conditioning data in the simulation.

## 4. A community roadmap toward DL-powered scientific advances in hydrology

Despite the challenges articulated above, here we offer a shared vision for a community roadmap for advancing hydrologic sciences using DL (Figure 4). A well-coordinated community is much more efficient and powerful in resolving problems, as we have seen in other scientific endeavours. Montanari et al. (2013) noted, "future science must be based on an interdisciplinary approach" and "the research challenges in hydrology for the next 10 years should be tackled through a collective effort". We see that several steps are crucial in this roadmap: devising ways to integrate physical knowledge, use DL to infer unknown quantities, process-based models (PBMs) and DL, community approaches in sharing and accessing data, open and transparent model competitions, baseline models and visualization packages and an education program that introduces data-driven methods at various levels.

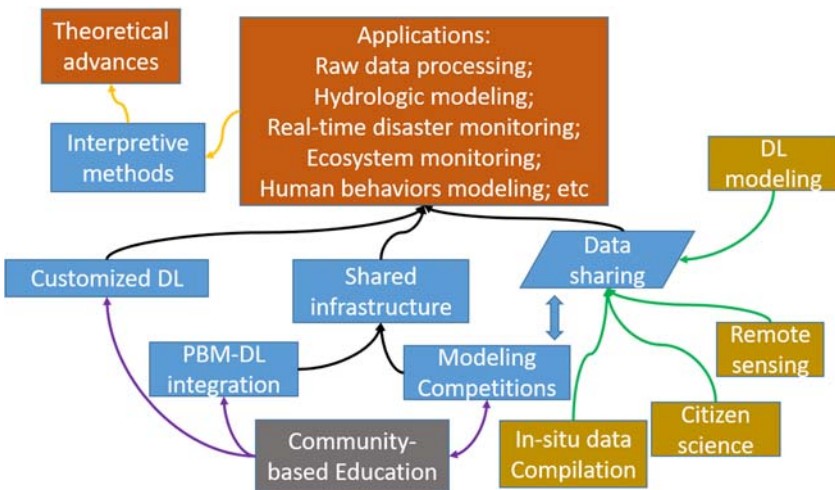

**Figure 4. A roadmap toward DL-powered scientific discovery in hydrologic science. Data availability can be increased by (green arrows) collecting and compiling existing data, incorporate novel data sources such as those collected by citizen scientists, remote sensing and modelled dataset. DL can be employed to predict data that are currently difficult to observe. The modelling competitions and the integration between PBM and DL will build important shared computing and analytic infrastructure, which, together with data sources, support a wide range of hydrologic applications. Interpretive methods should be attempted to extract knowledge from trained deep networks (orange arrows). Underpinning these activities is the enhanced, community-based educational program for machine learning in hydrology (purple arrows). However, these activities, especially the modelling competitions, might in turn feedback to the educational activity.**

## 4.1. Integrating physical knowledge, process-based models, and DL models

To address the challenge of data limitations (data quantity), we envision that a critical and necessary step is to more organically integrate hydrologic knowledge, process-based models, and DL. Process-based models, as they are derived from underlying physics, require less data for calibration than data-driven models. They can provide estimates for spatial and temporal data gaps and unobserved hydrologic processes. Well-constructed PBMs should also be able to represent temporal changes and trends. However, because data-driven models directly target observations, these models may have better performance in locations and periods where data are available. Also, as discussed earlier, data-driven models are less prone to *a priori* model structural error than are PBMs. We should aim to maximally utilize the best features of each type of models.

There will be a diversity of approaches with which PBMs and data-driven models could be combined. Karpatne et al., (2017) compiled a list of approaches of what they collectively call "theory-guided data science," which include (i) using knowledge to design data-driven model; (ii) using knowledge to initialize network states; (iii) using physical knowledge to construct priors to constrain the data-driven models; (iv) using knowledge-based constrained optimization (although this may be difficult to implement in practice); (v) using theory as regularization terms for the data-driven model, which will force the model to respect these constraints; and (vi) learn hybrid models, where the data-driven method is used as surrogate for certain parts of the physical model. One may also impose multiple learning objectives based on the knowledge of the problem.

There are a multitude of potential approaches and this list can be further expanded to accommodate various objectives. First, we can focus on PBM errors (difference between PBM simulation and observations). Non-deep machine learning has already shown promise in correcting PBM errors. Abramowitz et al. (2006) developed an ANN to predict the error in net ecosystem exchange from a land surface model, and achieved 95% reduction in annual error. More importantly, an ANN trained to correct the error at one biome corrects the PBM in another biome with a different temperature regime (Abramowitz et al., 2007). In the context of weather forecasts, machine learning methods were used to learn the patterns from past forecasting errors (Delle Monache et al., 2011, 2013), leading to a 20 percent improvement in performance for events of similar characteristics (Junk et al., 2015). Their results suggest PBMs make structural errors that are independent of the state-variable regimes, although there is a lack of theories to guide the separation of error types. We envision that PBMs can better resolve the impacts of regime changes, while DL can better capture state-independent error patterns and do mild state-dependent extrapolations. A co-benefit of modelling PBM error is that insights are gained about the PBM. Using interrogative methods to reverse engineer what DL has learned about PBM error provides possible avenues for improving the underlying PBM processes.

Second, PBMs can augment input data for DL models. PBMs can be used to increase supervising data for DLs, for example, for climate or land-use scenarios that have not existed presently, to augment existing data. Given model structural uncertainty (uncertainty with hydrologic processes), frameworks like the Structure for Unifying Multiple Modeling Alternatives (Clark et al., 2015) and automated model building (Marshall, 2017) could be employed to generate a range of outputs. Furthermore, if the DL training is limited by available data, we may not be able to reject alternative DL models that could generate unphysical or unrealistic outputs. Providing PBM simulations as either training data or regularization terms can help to nudge DL models to generate physically meaningful outputs. The extent to which errors in PBM model results affects DL outcomes remains to be explored. A theoretical framework is lacking for separately estimating aleatory uncertainty (resulting from data noise), and epistemic uncertainty (resulting from PBM error and training data paucity) and uncertainty due to regime-shift. The advantages and disadvantages of various approaches could be systematically and efficiently evaluated in community-coordinated fashion.

### 4.2. Multi-faced, community-coordinated hydrologic modelling competitions

There are many possible approaches and many alternative model structures for using DL to make hydrologic predictions and to provide insight into hydrologic processes. In the light of these challenges, we argue that open, fast and standardized competitions are one effective way of accelerating the progress. The competitions can evaluate the models not only in terms of predictive performance but also the attainment of understanding.

The impacts of competitions are best evidenced in the community-coordinated AI challenges, which use a standardized set of problems. These competitions have strongly propelled the advances in AI. Some have argued that the contributions of the ImageNet dataset and the competition may be more significant than the winning algorithms arising from the contests (Gershgorn, 2017). New methods can be evaluated objectively and disseminated rapidly through competitions. Because the

problems are standardized, they remove biases due to data sources and pre-processing. The community can quickly learn advantages and disadvantages of alternative model design through these competitions, which also encourage reproducibility.

We envision multi-faceted hydrologic modelling competitions where various models ranging from process-based ones to DL ones are evaluated and compared. The coordinators can, for example, provide a set of standard atmospheric forcings, landscape characteristics, and observed variables and provide targeted questions which participants must address. Importantly, the evaluation criteria should include not only performance-type criteria such as model efficiency coefficients and bias but also **qualitative/explanatory** ones such as explanations for control variables and model errors. Over-simplified or poorly-constructed models may provide more accessible explanations, but they might be misleading because the models may be overfitted to a given situation. Their simplicity may also constrain their ability to digest large datasets as a way of reducing uncertainty. Multi-faceted competitions allow us to also identify a "Pareto front" of explainability and performance and help rule out "false explanations". The objective of the competition is not only to seek the best simulation performance, but also those methods that offer deeper insight into hydrologic processes.

Another important value of competitions is that organizers will provide a standard input dataset and well-defined tasks. The entire community can leverage such effort. A substantial amount of effort is required to establish such a dataset, which may only be possible under a specifically designed project. Moreover, open competitions in the computer science field has produced well-known models such as AlexNet (Krizhevsky et al., 2012b), GoogLeNet (Szegedy et al., 2015), etc. These models serve as benchmarks and quick entry points for others. They can greatly improve reproducibility and the effectiveness of comparisons. Standard models, datasets and evaluation metrics will greatly improve DL adoption and hydrologic sciences.

### 4.3. Community-shared resources and broader involvement

A useful approach to address the major obstacle of data limitation is to increase our data repositories and to open access to existing data. Data value can be greatly enhanced by centralized data compilations, a task many institutions are already undertaking. For example, the Consortium of Universities for the Advancement of Hydrologic Science, Inc. (CUAHSI) provides access to large amounts of hydrologic data (CUAHSI, 2018a). As another example, in 2015, a project called Collaborative Research Actions (Endo et al., 2015) was proposed in Belmont Forum, which is a group of the world's major and emerging funders of global environmental-change research. Many scientists from different countries join the project and focus on the same issue, Food-Energy-Water Nexus. They shared their data (heterogeneous data) and research results from different regions.

ML has already been used to create useful hydrologic datasets such as soil properties (Chaney et al., 2016; Schaap, 1999) and landcover (Helber et al., 2017; Zhu et al., 2017) and cyclones (Liu et al., 2016). We envision there will continue to be significant progress in this regard, and the key to success will be the availability of ground-truth datasets. Using data sharing standards will advance data sharing across domains (WaterML2, 2018). Providing access to data through web services, such as used by

CUAHSI, negates the problem of storing data in a single location and enhances discoverability. Data brokers also provide more channels to share experiences, scholarly discussions, and debates along with the generation of data.

An important area where DL is expected to deliver significant value is the analysis of big and sub-research-quality data such as those collected by citizen scientists. Many aspects of the water cycle are directly accessible by everyone. Citizen scientists already gather data about precipitation (CoCoRaHS, 2018), temperature, humidity, soil moisture, river stage (CrowdHydrology, 2018), and potentially groundwater levels. These quantities can be measured using inexpensive instruments such as cameras, pressure gauges and moisture sensors. Volunteer scientists can also be solicited for data in places where such data can best reduce the uncertainty of the DL model, as in a framework called active learning (Settles, 2012). Social data have been used to help monitoring flood inundation (Sadler et al., 2018; Wang et al., 2018). Crowd-sourced data have played roles in DL research, where a large but noisy dataset was argued to be more useful than a much smaller but well-curated dataset (Huang et al., 2016; Izadinia et al., 2015). Even though there are problems related to data quality which can be overcome using AI approaches. An important co-benefit of involving citizen scientists is the education and outreach to the public. The active engagement is much more effective when the public has a stake in the research outcomes.

### 4.4. Education

A major barrier to realizing the benefits of data science and DL lies in our undergraduate and graduate curriculum. Little in hydrologists' standard curriculum prepares them for a future with substantially more data-driven science. Statistical courses often do not cover machine learning basics, while data mining courses offered by computer science departments lack the connections to the water discipline. Given the interdisciplinary nature of hydrology, it has been long recognized that it takes a community to raise a hydrologist (Merwade and Ruddell, 2012; Wagener et al., 2012). We propose a concerted effort by current hydrologic machine learning researchers, with participation from computer scientists, to pool and share educational content. Such effort will form the basis of a hydrologic data mining curriculum and leverage the wit of the community. Collaborations may form through either grassroot collaborations or institutionally-supported education projects, e.g., (CUAHSI, 2018b). The open competitions would be a great source of education materials. A diversity of models that have been evaluated and contrasted help clarify Pros and Cons of different methods. Shared datasets, DL algorithms and data pre-processing software can be leveraged in classrooms.

As with the design of any education effort, it is important to consider inclusiveness and diversity. Especially, for hydrologic DL, the source field of AI appears to have an extremely poor track record of gender balance (Simonite, 2018). The reason to such imbalance could be rooted in introductory computer science classes, undergraduate curriculum and social stereotypes. Research has found that the introductory computer science classes, especially those taken by non-majors, are instrumental in developing a desire to stay in the field (Lehman, 2017). In addition, the portrayal of gender stereotypes regarding computing and the increase in weed-out courses (Aspray, 2016) have both discouraged women students in computer science (Sax et al., 2017). To counter such negative impacts, the introductory courses in the curriculum need to assume little prior programming

experiences. Special attention must be paid by the educators to shatter the stereotypes. On the other hand, the richness of natural beauty in hydrology and the connection between data and the real world may be employed to help bridge the gender gap.

## 5. Concluding remarks

In this opinion paper, we argue that hydrologic scientists ought to give thoughts to a research avenue that complements traditional approaches, wherein DL-powered data mining is used to generate hypotheses, predictions, and insights. Although in the past there may have been strong reservations toward black-box approaches, recent efforts have been put in the interpretation and understanding of deep learning networks. The black-box perception of ML in hydrology is perhaps a self-reinforcing curse on this juvenile field, as a rejection of the research avenue based on this perception will, in turn, jeopardize

the development of more transparent algorithms. Progress in hydrology and other disciplines show that there is substantial promise in incorporating DL into hydrologists' toolbox. However, challenges such as data limitation and model variability demand a community-coordinated approach.

We have also argued for open hydrologic competitions that emphasize both performance and explainability. These competitions will greatly improve the growth of the field as a whole. They serve as valuable "organizing events", where

different threads in algorithm development, model evaluation and comparison, reproducibility tests, dataset compilation, resource sharing and community organization all come to a convergence, to spur growth in the field.

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

# HESS Opinions: Incubating deep-learning-powered hydrologic science advances as a community

Chaopeng Shen[1], Eric Laloy[2], Amin Elshorbagy[3], Adrian Albert[4], Jerad Bales[5], Fi-John Chang[6], Sangram Ganguly[7], Kuo-lin Hsu[8], Daniel Kifer[9], Zheng Fang[10], Kuai Fang[1], Dongfeng Li[10], Xiaodong Li[11], and Wen-Ping Tsai[1]

*1. Civil and Environmental Engineering, Pennsylvania State University, University Park, PA 16802*
*2. Institute for Environment, Health and Safety, Belgian Nuclear Research Centre, Mol, Belgium*
*3. Dept. of Civil, Geological, and Environmental Engineering, University of Saskatchewan, Saskatoon, Canada*
*4. Energy Technologies Area, Lawrence Berkeley National Laboratory, Berkeley, CA 94720*
*5. Consortium of Universities for the Advancement of Hydrologic Science, Inc. (CUAHSI), Cambridge, MA*
*6. Department of Bioenvironmental Systems Engineering, National Taiwan University, Taipei, 10617, Taiwan*
*7. NASA Ames Research Center/ BAER Institute, Moffett Field, CA 94035*
*8. Civil and Environmental Engineering, University of California, Irvine, Irvine, CA 92697*
*9. Computer Science and Engineering, Pennsylvania State University, University Park, PA 16802*
*10. Civil Engineering, University of Texas at Arlington, Arlington, TX 76013*
*11. State Key Laboratory of Hydraulics and Mountain River Engineering, Sichuan University, Sichuan, China*

*Correspondence to*: Chaopeng Shen (cshen@engr.psu.edu)

**Abstract.** Recently, deep learning (DL) has emerged as a revolutionary and versatile tool transforming industry applications and generating new and improved capabilities for scientific discovery and model building. The adoption of DL in hydrology has so far been gradual, but the field is now ripe for breakthroughs. This paper suggests that DL-based methods can open a
complementary avenue toward knowledge discovery in hydrologic sciences. In the new avenue, machine-learning algorithms present competing hypotheses that are consistent with data. Interrogative studies are then invoked to interpret DL models for scientists to further evaluate. However, hydrology presents many challenges to for DL methods, such as data limitations, heterogeneity and co-evolution, and the general inexperience of the hydrologic field with DL. The roadmap toward DL-powered scientific advances will require the coordinated effort from a large community involving scientists and citizens.
Integrating process-based models with DL models will help alleviate data limitations. The sharing of data and baseline models will improve the efficiency of the community as a whole. Open competitions could serve as the organizing events to greatly

propel growth and nurture data science education in hydrology, which demands a grass-root collaboration. The area of hydrologic DL present numerous research opportunities which could, in turn, stimulate advances in machine learning as well.

## 1. Overview

Deep learning (DL) is a suite of tools centered on artfully designed large-size artificial neural networks. The deep networks at the core of DL are said to have "depth" due to their multi-layered structures, which help deep networks represent abstract concepts about the data (Schmidhuber, 2015). Given input attributes that describe an instance, deep networks can be trained to make predictions of some dependent variables, either continuous or categorical, about this instance. For example, for standard computer vision problems, deep networks can recognize the theme or objects from a picture (Guo et al., 2016; He et al., 2016; Simonyan and Zisserman, 2014) or remotely sensed images (Zhu et al., 2017). For sequential data, DL can associate natural language sequence to commands (Baughman et al., 2014; Hirschberg and Manning, 2015) or predict the action of an actor in the next video frame (Vondrick et al., 2016). DL can also generate (or synthesize) images that carry certain artistic styles (Gatys et al., 2016) or a natural language response to questions (Leviathan and Matias, 2018; Zen and Sak, 2015). With the support of deep architectures, deep networks can automatically engineer relevant concepts and features from large datasets, instead of requiring human experts to define these features (Section 2.2.2). As a foundational component of modern artificial intelligence (AI), DL has made substantial strides in recent years and helped to solve problems that have resisted AI for decades (LeCun et al., 2015).

While DL has stimulated exciting advances in many disciplines and has become the method of choice in some areas, hydrology so far have only had a very limited set of DL applications (Shen, 2018) (hereafter referred to as Shen18). Despite scattered reports of promising DL results (Fang et al., 2017; Laloy et al., 2017, 2018; Tao et al., 2016; Vandal et al., 2017; Zhang et al., 2018), hydrologists have not to have reservations aboutwidely adopted these new tools, perhaps with some good reasoning. This collective opinion paper argues that there are many opportunities in hydrological sciences where DL can help provide both stronger predictive capabilities and a complementary avenue toward scientific discovery. We then reflect on why it has been challenging to harness the power of DL and big data in hydrology and explore what we can do as a community to incubate progress. Readers who are less familiar with machine learning or deep learning are referred to a companion review paper (Shen18), which provides a more comprehensive and technical background than this opinion paper. Many details behind the arguments in Section 2 are provided in Shen18.

We first voice the opinions that elements of a complementary machine learning-based scientific discovery avenue are taking shape, and this avenue should at least be considered for problems with large data (section 2). Then, we propose several ways to accelerate this avenue (section 3). Finally, we argue that hydrology offers a unique set of challenges for DL research (section 4).

## 2. The emergence of a complementary research avenue

We are witnessing the growth of three pillars needed for DL to support a research avenue that is complementary to traditional hypothesis-driven research: big hydrologic data, powerful machine learning algorithms, and interrogative methods (such as

visualization and techniques) to extract interpretable knowledge from the trained networks. This new avenue starts from data, uses DL methods to generate hypotheses, and applies interrogative methods to help us understand hydrologic system functioning. We discuss these aspects in the following sections.

### 2.1. With more data, opportunities arise

The fundamental supporting factor for emerging opportunities with DL is the growth of big hydrologic data, with all surface,

sub-surface, urban, infrastructure, and ecosystem dimensions. In this paper, hydrology refers to both the complete natural and engineered water cycle, and associated processes in the ecosystem and geologic media. There are ever increasing amounts of hydrologic data available through remote sensing (see a summary in Srinivasan, (2013)) and data compilations. For example, satellite-based datasets include precipitation, surface soil moisture (Entekhabi, 2010; Jackson et al., 2016; Mecklenburg et al., 2008), vegetation states and indices, e.g., (Knyazikhin et al., 1999), and derived evapotranspiration products (Mu et al., 2011),

terrestrial water storage (Wahr et al., 2006), snowcover (Hall et al., 2006), and a planned mission for estimating streamflows (Pavelsky et al., 2014), etc. On the data compilation side, there are now compilations of geologic (Gleeson et al., 2014) and soil datasets; centralized management of streamflow and groundwater data in the United States, Europe, parts of South America and Asia, or globally for some large rivers (GRDC, 2017); water chemistry, groundwater samples and other biogeophysical datasets. The Consortium of Universities for the Advancement of Hydrologic Science, Inc. (CUAHSI) operates two

datasystems for the discovery and archival of water data: the Hydrologic Information System (CUAHSI, 2018c) for time series, and HydroShare for all water data types (Horsburgh et al., 2016). An Internet of Water (Aspen, 2017) has been proposed and is beginning to develop, thereby improving access to these emerging data sets.

Moreover, unconventional data sources are starting to emerge. High-resolution sensing of Earth will be provided by increasing amount of CubeSats, Unmanned Aerial Vehicles, balloons, inexpensive photogrammetric sensing and many other sources

(McCabe et al., 2017). These new sources provide new forms of measurements not envisioned before. For example, cell phone signal strength and cell-phone pictures can contribute to high resolution monitoring of rainfall intensity (Allamano et al., 2015). Inexpensive infrared camera images can detect water levels in complex urban water flows (Hiroi and Kawaguchi, 2016). Internet-of-Things (IoT) sensors embedded in water infrastructure can transmit data about the states of water in our environment (Zhang et al., 2018). These new sources of information provided unprecedented volumes and multi-faceted

coverages of the natural and built environment. However, since each new data source has its own characteristics and peculiarities, the identification of the appropriate approaches to fully exploit their value, especially synergistically, creates a

significant challenge. In contrast, DL models can be built, without significant human expertise and extensive manual labour, to rapidly derive useful information from these data.

## 2.2. DL: A big step forward

### 2.2.1. Rapid adoption

The field of hydrology has witnessed flows and ebbs of several generations of machine learning methods in the past few decades. From regularized linear regression (Tibshirani and Tibshirani, 1994) to Support Vector Regression  (Drucker et al., 1996), from genetic programming (Koza, 1992) to artificial neural networks (Chang et al., 2014; Chen et al., 2018; Hsu et al., 1995, 1997, 2002), from classification and regression tree to random forest (Ho, 1995), from Gaussian Process (Snelson and Ghahramani, 2006) to Radial Basis Function Network (Moradkhani et al., 2004), each approach offered useful solutions to a

set of problems, but each also faced its own limitations. As a result, over time, some may have grown dispassionate about progress in machine learning, while some others may have concerns about whether DL represents real progress or is just a "hype."

The progress in AI brought forth by DL to various industries and scientific disciplines is revolutionary (Section 4 in Shen18) and can no longer be ignored by the hydrology community. Major technology firms have rapidly adopted and commercialized

DL-powered AI (Evans et al., 2018). For example, Google has re-oriented its research priority from "mobile-first" to "AI-first" (Dignan, 2018). The benefits of these industrial investments can now be felt by ordinary users of their services such as machine translation and digital assistants who can engage in conversations sounding like a human (Leviathan and Matias, 2018). Moreover, AI patents of industries and scientific disciplines grew at a 34% compound annual growth rate between 2013 and 2017, apparently after DL's breakthroughs in 2012 (Columbus, 2018). Also reported in (Columbus, 2018), more than 65%

of data professionals responded to a survey indicating AI as their company's most significant data initiative for next year.

DL is gaining adoption in a wide range of scientific disciplines and, in some areas, has started to substantially transform those disciplines. The fast growth is clearly witnessed from literature searches. Since 2011, the number of entries with DL as a topic increased almost exponentially, showing around 100% compound annual growth rate before 2017 (Table 1). DL evolved from occupying less than 1% of machine learning (ML) entries in computer science (CS) in 2011 to 46% in 2017. This change

showcases massive conversion from traditional machine learning to DL within computer science. Other disciplines lagged slightly behind, but also experienced exponential increase. They also saw the DL/ML ratio jumping from 0% in 2011 to 33% in 2017. As reviewed in Shen18, DL has enhanced the statistical power of data in high energy physics, and the use of DL can be considered to be equivalent to a 25% increase in the experimental dataset (Baldi et al., 2015). In biology, DL has been used to predict potential pathological implications from genetic sequences (Angermueller et al., 2016). DL models fed with raw-

level data have been shown to outperform those using expert-defined features when they predict high-level outcomes, e.g.,

toxicity, from molecular compositions (Goh et al., 2017). Just like other methods, DL may eventually be replaced by newer ones, but that is not a reason to hold out on possible progress.

**Table 1. Number of papers returned from searches on ISI Web of Science.**

| year | DL-nonCS | DL-CS | ML-non-CS | ML-CS | DL/ML-CS | DL/ML-nonCS |
|------|----------|-------|-----------|-------|----------|-------------|
| 2011 | 0 | 23 | 1068 | 1838 | 1% | 0% |
| 2012 | 15 | 25 | 1310 | 1899 | 1% | 1% |
| 2013 | 35 | 80 | 1677 | 2360 | 3% | 2% |
| 2014 | 84 | 238 | 2228 | 3050 | 8% | 4% |
| 2015 | 308 | 709 | 3074 | 4405 | 16% | 10% |
| 2016 | 841 | 1462 | 4414 | 5361 | 27% | 19% |
| 2017 | 2035 | 2723 | 6125 | 5860 | 46% | 33% |

*DL-CS results were obtained by searching for "Topic" (TS)="Deep Learning" AND "Research area" (SU)= "Computer Science"; ML-CS was obtained the same way as DL-CS, only that "Deep learning" was replaced by "machine learning"; DL-nonCS was obtained by TS="Deep Learning" NOT SU="Computer Science" NOT SU=education. Education was removed because entries in this category were not related to our definition of DL. There were also 19 articles in 2011 where deep learning was about education in disciplines other than SU=Education. Therefore, 19 was used as a blank value and also*
*subtracted from the DL-nonCS column. DL/ML-CS is ratio of DL-CS to ML-CS expressed as a percentage. DL/ML-nonCS was obtained similarly.*

Many of the abovementioned advances were driven by DL's domination in AI competitions:

- The ImageNet Challenges is an open competition to evaluate algorithms for object detection and image classification
(Russakovsky et al., 2014). Topics change during each contest, and a dataset of ~14M tagged images and videos were cumulatively compiled, with convenient and uniform data access provided by the organizers. The 2010 Challenge was won by a large-scale Support Vector Machine (SVM). Convolutional Neural Network, a kind of deep network, first won this contest in 2012 (Krizhevsky et al., 2012a). This victory heralded the exponential growth of DL in popularity. Since then, and until 2017 (the last contest), the vast majority of entrants and all contest winners used
CNNs, which edges out other methods by large margins (Schmidhuber, 2015).

- The IJCNN traffic sign recognition contest, which is composed of 50,000 images (48 pixels x 48 pixels), witnessed superhuman visual recognition performance (greater than human recognition) from CNN-based methods (Stallkamp et al., 2011). CNNs also performed better than humans on recognition of cancers from medical images (Yu et al., 2016).

- The TIMIT speech corpus is a dataset that holds the recordings from 630 English speakers. LSTM-based models showed a large edge over Hidden Markov Model (HMM) results (Graves et al., 2013) in recognizing the speeches. Similarly, LSTM-based methods significantly outperformed all statistical approaches in keyword spotting

(Indermuhle et al., 2012), optical character recognition, language identification, text-to-speech synthesis, social signal classification, machine translation and Chinese handwriting recognition.

- An LSTM-based speech recognition system has achieved "human parity" in conversational speech recognition on the Switchboard corpus (Xiong et al., 2016). A parallel version achieved best-known pixel-wise brain image segmentation results on the MRBrainS13 dataset (Stollenga et al., 2015). The improvement in language translation software can be witnessed by ordinary web users.

- A time-series forecasting contests, Computational Intelligence in Forecasting Competition, was won by a combination of fuzzy and exponential models in 2015 when no LSTM was present, but LSTM won the contest in 2016 (CIF, 2016).

In contrast, only a handful of applications of big data DL could be found in hydrology, but they already demonstrated great promise. Vision DL has been employed to retrieve precipitation from satellite images, where it demonstrated a materially-superior performance than earlier-generation neural networks (Tao et al., 2017, 2018). GAN was used to imitate and generate scanning images of geologic media (Laloy et al., 2018), where the authors showed realistic replication of training image patterns. Time-series deep learning network was employed to temporally extend satellite-sensed soil moisture observations (Fang et al., 2017) and was found to be more reliable than simpler methods. Time series DL rainfall-runoff models which are confined to certain geographic divisions have been created (Kratzert et al., 2018). There are also DL studies, based on smaller datasets, to help predict water flows in the urban environment (Assem et al., 2017) and water infrastructure (Zhang et al., 2018). In addition to utilizing big data, DL was able to create valuable, big datasets that could not have been otherwise possible. For example, utilizing DL, researchers were able to generate new datasets for Tropical Cyclones, Atmospheric Rivers and Weather Fronts (Liu et al., 2016; Matsuoka et al., 2017) by tracking them. Machine learning has also been harnessed to tackled the convection parameterization issue in climate modelling (Gentine et al., 2018).

### 2.2.2. Technical advances

Underpinning the powerful performance of DL are its technical advances. The deep architectures have several distinctive advantages: (1) deep networks are designed with the capacity to represent extremely complex functions. (2) After training, the intermediate layers can perform modular functions which can be migrated to other tasks, in a process called transfer learning, and extend the value of the training data. (3) The hidden layer structures have been designed to automatically extract features, which helps dramatically reducing labour, expertise and the trial and error time needed for feature engineering. (4) Compared to earlier models like classification trees, most of the deep networks are differentiable, meaning that we can calculate derivatives of outputs with respect to inputs or the parameters in the network. This feature enables highly efficient training algorithms that exploit these derivatives. Moreover, the differentiability of neural networks enables querying DL models for sensitivity analysis of outputs to input parameters, a task of key importance in hydrology.

Metaphorically, the intermediate (or hidden) layers in DL algorithms can be understood as workbenches or placeholders for tools that are to be built by deep networks themselves. These hidden layers are trained to calculate certain features from the data, which are then used by downstream layers to predict the dependent variables. For example, Yosinski (2015) showed that some intermediate layers in a deep vision recognition network are responsible for identifying the location of human or animal faces; Karpathy et al., (2015) showed that some hidden cells in a text prediction network act as length counters of a line while some others keep track of whether the text is in quotes or not. These functionalities were not bestowed by the network designers, but emerged by themselves after network training. Earlier network architectures either did not have the needed depth, or were not designed in an artful way such that the intermediate layers could be effectively trained. For more technical details, refer to an introduction in Schmidhuber (2015) and Shen (2018).

Given that deep networks can identify features without human guide, it follows that they may extract features that the algorithm designers were unaware of, or did not intentionally encode the network to do. If we could believe that there is latent knowledge about the hydrologic system that humans are not yet aware of, but can be determined from data, the automatic extraction of features leads to a potential pathway toward knowledge discovery. For example, deep networks recently showed that grid-like neuron response structures automatically emerge at intermediate network layers for a network trained to imitate how mammals perform navigation, providing strong support to a Nobel-winning neuroscience theory about the functioning of these structures (Banino et al., 2018).

Deep networks may be more robust than simpler models despite their large size, if they are regularized properly (regularization techniques apply penalty to model complexity to make the model more robust) and are chosen based on validation errors in a two-stage approach (Kawaguchi et al., 2017). Effective regularization techniques include (i) early stopping: monitor the training progress on a separate validation set and stop the training once validation metrics start to deteriorate; and/or (ii) novel regularization techniques such as dropout (Srivastava et al., 2014). DL models can be easier to train than previous networks, as their architectures and new stochastic gradient techniques (Kingma and Ba, 2014) address issues like vanishing gradient (Hochreiter, 1998). Training large networks used today was computationally implausible until scientists started to exploit the parallel processing power of graphical processing units (GPUs). Nowadays, new application-specific integrated circuits have also been created to specifically tackle DL, although DL architectures are still evolving.

Primary types of successful deep learning architectures include convolutional neural networks (CNN) for image recognition (Krizhevsky et al., 2012b; Ranzato et al., 2006), Long short-term memory (LSTM) (Greff et al., 2015; Hochreiter and Schmidhuber, 1997) for time series modeling, variational auto-encoders (VAE) (Kingma and Welling, 2013), and deep belief networks for pattern recognition and data (typically image but also text or sound, etc) generation (section 3.2 in Shen18). Besides these new architectures, a novel generative model concept called the generative adversarial network (GAN) has become an active area of research. The key characteristic of GANs is that they are learned by creating a competition between the actual generative model or ''generator'' and a discriminator in a zero-sum game framework (Goodfellow et al., 2014), in

which these components are learned jointly. Compared to other generative models, GANs potentially offer much greater flexibility in the patterns to be generated. The power of GANs has been recognized recently in the geoscientific community, especially in machine learning research inspired by physics, where ~~deep generative models~~GANs have been used ~~for~~ to generate certain complicated physical, environmental, and socio-economic systems ~~with deep generative models~~ (Albert et al., 2018; Laloy et al., 2018).

While showing many advantages, DL models will require substantial amount of computing expertise. The tuning of hyper-parameters, e.g. network size, learning rate, batch size, etc., often require a priori experiences and trial and error. The computational paradigm, e.g., computing on graphical processing units, is also substantially different from ordinary hydrologists' educational background. The fundamental theories on why DL generalizes so well have not been maturely developed (Section 2.7 in Shen18). In the ongoing debates, some argued that a large part of DL's power comes from memorization while others countered that DL prioritizes learning simple patterns (Arpit et al., 2017; Krueger et al., 2017) and a two-stage procedure (training and testing) also helped (Kawaguchi et al., 2017). Despite these explanations, it has been found in vision DL that deep networks can be fooled by adversarial examples where small, unperceivable perturbations to input images sometimes cause large changes in predictions, leading to incorrect outcomes (Goodfellow et al., 2015; Szegedy et al., 2013). It remains to be seen whether such adversarial examples exist for hydrologic DL applications. If we can recreate adversarial examples, they can be added into the training dataset to improve the robustness of the model (Ororbia et al., 2016).

### 2.3. Network interrogative methods to enable knowledge extraction from deep networks

Conventionally, neural networks were primarily used to approximate mappings between inputs and outputs. The focus was put on improving predictive accuracy. In terms of the use of neural networks in scientific research, then, there have been a major concern that DL and more generally machine learning are referred to as black boxes that cannot be understood by humans and, thus, cannot serve to advance scientific understanding. At the same time, data-driven research may lack clearly-stated hypotheses which is in contrast to traditional hypothesis-driven scientific methods. There has been significant pressure from both inside and outside the DL community to make the network decisions more explainable. For example, new (as of January 2018) European data privacy laws dictate that automated individual decision making, which significantly influences the algorithm's users must provide a "right to explanation" where a user can ask for an explanation of an algorithmic decision (Goodman and Flaxman, 2016).

Some recent progress in DL research focused on addressing these concerns. Notably, a new sub-discipline, known as "AI neuroscience" has produced useful interrogative techniques to help scientists interpret the DL model (see literature in Section 3.2 in Shen18). The main classes of interpretive methods include (i) reverse-engineer the hidden layers: attributing deep network decisions to input features or a subset of inputs; (ii) transferring knowledge from deep networks to interpretable, reduced-order models. (iii) visualization of network activations. Many scientists have also devised case-by-case adhoc methods, e.g., to investigate the correlation between inputs and cell activations (Shen, 2018; Voosen, 2017).

Interpretive DL methods have so far not been employed in hydrology or even geosciences. However, to give some examples from other domains, in medical image diagnosis, some researchers used reverse engineering methods to show which pixel on an image led the network to make its decision about anatomy classifications (Kumar and Menkovski, 2016). They found that the network traced its decisions to image landmarks mostly often used by human experts. In more recent research, AI researchers trained their network to not only classify an image, but also didactically explain why the decision was made and why an image is one class instead of another (Figure 1). Extending this idea to the precipitation retrieval problem in hydrology as in (Tao et al., 2017, 2018), we could let DL inform us what features on the satellite cloud image is helpful for reducing bias in precipitation retrieval.

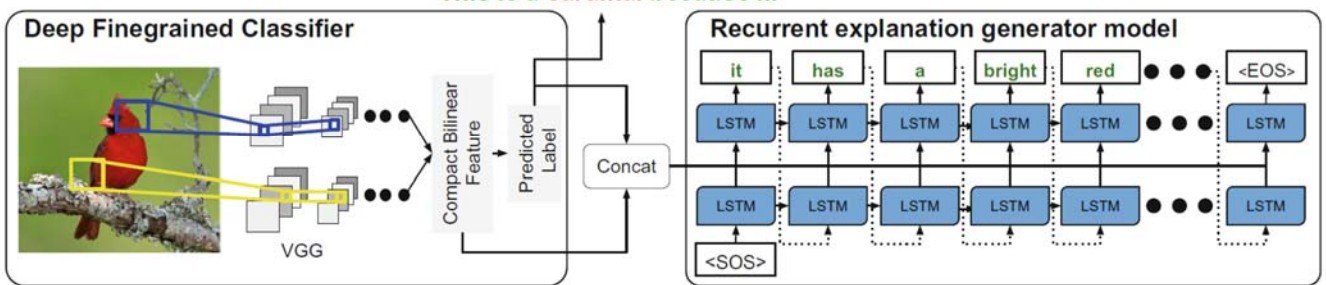

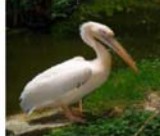
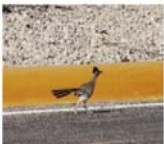
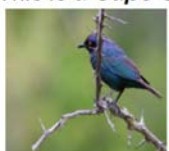

Figure 1. (Reprinted from Hendricks et al., 2016 with permission) Authors trained a joint classification and explanation network for image classification. The bolded text is a "class-relevant" attribute (a distinguishing attribute for the class) in the explanation. Their classification network extracts visual features (regions on the image) responsible for the decision. Then, the explanation network links these regions to distinguishing words in a dictionary to form an explanation that explains the reason for the classification, and why it is not other classes. This level of explanation may be difficult to achieve for hydrologic problems due to limited supervising data (annotated dictionary for classes), but it is possible to borrow the idea of associating features in the input data with some descriptive words.

**2.4. The complementary research avenue**

As the interrogative methods further grow, there emerges a research avenue toward attaining knowledge that is complementary to the traditional hypothesis-driven one (Figure 2). The data-driven research avenue can be divided into four steps: (i) hypotheses are generated by machine learning algorithms from data; (ii) the validation step is where data withheld from training, and different from training, are employed to evaluate the machine-learning-generated hypotheses; (iii) interpretive methods are employed to extract data-consistent and human-understandable hypotheses (Mount et al., 2016) (described in Section 2.3); and (iv) the retained hypotheses are presented to scientists for analysis and further data collection, and the process iterates.

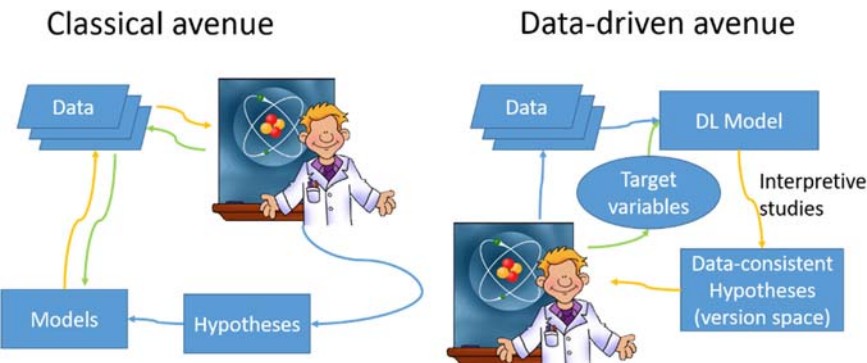

**Figure 2. Comparing two alternative avenues toward gaining knowledge from data. In the classical avenue, scientists compile and interpret data, form hypotheses, (optionally) build models to describe data and hypotheses (the green pathway). Then the model results with data to affirm or reject the hypotheses and the feedbacks (the yellow pathway) allow the scientist to revise the model and iterate. In the data-driven avenue, scientists collect data and define the target variables of DL models (the green path). Then interpretive methods are invoked to extract data-consistent and human-understandable hypotheses (the yellow path). There must be a hypotheses validation step where data withheld from training is used to evaluate or reject the hypotheses.**

The classical avenue, especially when applied to modelling studies, attracts non-uniqueness and subjectivity. To give a concrete example, consider a classical problem of rainfall-runoff modelling. Suppose a hydrologist found that hydrologic responses in several nearby basins are different. Some basins produce flashier peaks while others have smaller peaks in summer, large seasonal fluctuation and large peak streamflows only in winter. Taking a modelling approach, the hydrologist might invoke a conceptual hydrologic model, e.g., Topmodel (Beven, 1997), only to find that the model results do not adequately describe the observed heterogeneity in the rainfall-runoff response. The hydrologist might hypothesize that the different behaviours are due to heterogeneity in soil texture, which is not well represented in the model. Subsequently, the hydrologist incorporates processes that represent soil spatial heterogeneity, such as refined soil pedo-transfer functions that can differentiate between the soil types in different regions. Perhaps with some parameter adjustment, this model can provide streamflow predictions that are qualitatively similar to the observations. This procedure then increases the hydrologist's

confidence that the heterogeneity in soil hydraulic parameters is indeed responsible for their different hydrologic responses. However, this improvement is not conclusive due to process equifinality: there can be alternative processes that can also result in similar outcomes, e.g., the influence of soil thickness, Karst geology, terrain or drainage density. The identification of potential improvement might be dependent on the hydrologist's intuition or pre-conceptions, which are nonetheless important but potentially biased. While the intention of a process-based model may be deductive (Beven, 1989), the example process given above is, in fact, abductive reasoning (Josephson and Josephson, 1994), as it seeks a plausible but not exhaustive thorough of the phenomenon. Furthermore, incorporating all the physics into the model may prove technically challenging or too time-consuming.

Compared to the classical avenue, the inductive data-driven approach allows us to more efficiently explore a larger set of hypotheses. Although it cannot be said that the machine learning algorithms present no human bias (because inputs are human-defined and some hyper-parameters are empirically adjusted), the larger set of hypotheses presented will at least greatly reduce that risk. First, let us examine a Classification and Regression Tree algorithms-based (CART-based) data-driven approach (Fang and Shen, 2017). We could start with physiographic data for many basins in this region, including terrain, soil type, soil thickness, etc. We can use CART to model the process-based model's errors, which allows us to separate out the conditions under which these errors occur more frequently. We let the pattern emerge out of data without enforcing a strong human pre-conceived hypothesis. Attention must be paid to the robustness of the data mining and utilize holdout dataset or cross-validation to verify the generality of the conclusion. Data may suggest that soil thickness is the main reason for the error. Or, if data do not prefer one hypothesis over the other, then all hypotheses are equally possible and cannot be ruled out. This advantage of DL can be summarized in a short phrase, "*an algorithm has no ego.*" On a practical level, this approach can more efficiently and simultaneously examine multiple competing hypotheses.

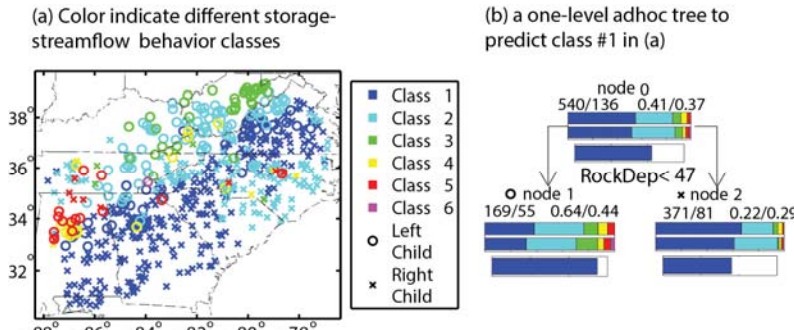

**Figure 3. (adapted from Fang and Shen 2017. Reprint permission obtained). We calculated storage-streamflow correlation patterns over continental United States (CONUS) and divided small or mesoscale basins into multiple classes. We studied what physical factors most cleanly separate different correlation patterns. In this case, what separates the blue class (storage and streamflow are highly correlated across all flow regimes) and the green class turned out to be soil thickness. It suggests the blue basins in the south have high correlation because they have thick soils, which facilitates infiltration, water storage, and groundwater-dominated streamflow.**

One example of such analyses was carried out in Fang and Shen, (2017) where differences in basin storage-streamflow correlations were explained by physical factors using CART, an earlier-generation data mining method (Figure 3). The data mining analysis allowed patterns to emerge, which inspired hypotheses about key factors that control the hydrologic functioning of different systems, such as soil thickness and soil bulk density are important controls of streamflow-storage

relationships. For another example, data-mining analysis showed that drought recovery time is associated to temperature and precipitation, while biodiversity only has secondary importance (Schwalm et al., 2017). Scientists need to define the predictors and general model types, but they do not pose strongly constraining hypotheses about the controlling factors, and instead "let the data speak". The key to this approach is a large amount of data from which patterns emerge.

However, working with DL models, we need to further resort to interrogative methods to make the results understandable

(Figure 2 right panel). For example, we can construct DL models to predict the errors of the process-based model, and then use visualization techniques to see which variable, under which condition, lead to the error. Because DL can absorb a large amount of data, it can find commonality among data as well as identify differences. Whereas CART models are limited by the amount of data and face stability problems in lower branches (data are exponentially less at lower branches), DL models may produce a more robust interpretation.

The machine learning paradigm lends us to finding "unrecognized linkages" (Wagener et al., 2010) or find complex patterns in the data that humans could not easily realize or capture. Owning to the strong capability of DL, it can better approximate the "best achievable model" (BAM) for the mapping relations between inputs and output. As such, it lends support to measuring the information content contained in the inputs about the output. Nearing et al., (2016) utilized Gaussian Process regression to approximate the BAM. DL can play similar roles and can also allow for modelling, perhaps in a more thorough way. The

simplicity of building DL model and altering inputs makes it an ideal testbed for new ideas.

Outputs from the hidden layers of deep networks can now be visualized to gain insights about the transformations performed on the input data by the network (Samek et al., 2017). For image recognition tasks, one can invert the DL model to find out the parts of the inputs that led the network to make a certain decision (Mahendran and Vedaldi, 2015). There are also means to visualize outputs from recurrent networks, e.g., showing the conditions under which certain cells are activated (Karpathy et

al., 2015). These visualizations can illustrate the relationships that the data-driven model has identified.

Considering the above potential benefits, the data-driven avenue should at least be considered or given an opportunity to play a role in hydrological sciences discovery. However, this avenue may be uncomfortable to some researchers. In the classical avenue, the scientist must originate the hypotheses before constructing models; in the data-driven one, the data mining/knowledge discovery process is a precursor step to the main hypotheses formation-- hypotheses cannot be generated

before the data mining analysis (Mount et al., 2016). This feature is a natural consequence of handing part of the work to an algorithm but may cause some disarray for those who follow what has been perceived as structured scientific methods. Especially, hypotheses can no longer be unequivocally stated during the proposal stage of research.

Granted, the interrogative methods as a whole are new and time is required for them to grow. We need to note that the nascent "DL neuroscience" literature did not exist until 2015. However, if we outright reject the complementary avenue based on the habitual thinking that neural networks are black boxes, we may deny ourselves an opportunity for breakthroughs.

## 3. Challenges and opportunities for DL in hydrology

The field of hydrology has a unique set of challenges that are also research opportunities for DL. Many of these science challenges have, to date, not been effectively addressed using traditional methods, and cannot be sufficiently tackled by individual research groups. Some challenges for which DL approaches might be exploited are presented below.

Observations in hydrology and water science generally are regionally and temporally imbalanced. For example, while streamflow observations are relatively dense in the United States, such data are sparse in many other parts of the world, either

because measurements have not been made or are not made accessible. There is often a dearth of observations that can be used as comprehensive training datasets for DL algorithms. Few hydrologic applications have as much data as the data available to standard AI research applications such as imagine recognition or natural language processing. Remote sensing of hydrologic variables also has limitations, including effects of canopy and clouds which can limit observations, temporal density of observations because of orbital paths, and observation footprints, which create challenges when trying to validate satellite

observations with field point measurements. A body of literature studying this problem across different geographic regions can be loosely summarized under the topic of "prediction in ungauged basins" (PUB) (Hrachowitz et al., 2013). PUB problems pose a significant challenge to data-driven methods.

Global change is altering the hydrologic and related cycles, and hydrologists must now make predictions in anticipation of changes, beyond previously observed ranges (Wagener et al., 2010). Especially, more frequent extremes have been observed

for many parts of the world and such extremes have been projected to occur more frequently in the future (Stocker et al., 2013). Data-driven methods must demonstrate their capability to make reasonable predictions when applied out of the range of the training dataset.

Observations of the water cycle tend to focus on one aspect of the water cycle, and seldom offer a complete description. For example, we can estimate total terrestrial water storage (Wahr, 2004) or top 5-cm surface soil moisture via multiple satellite

missions. It is difficult, however, to directly combine such observations of components of the water cycle into a complete picture of the water cycle. A challenge, then, is merging distinct observations, with all their space-time discontinuities to aid predictions, model validation, and to provide a more complete understanding of the global water cycle.

Hydrologic data are accompanied by a large amount of strongly heterogeneous (Blöschl, 2006) "contextual variables" such as land use, climate, geology, and soil properties. The proper scale at which to represent heterogeneity in natural systems is a

vexing problem (Archfield et al., 2015), as micro-scale of soil heterogeneity, for example, is not computationally realistic in

hydrologic models.  The scale at which heterogeneity should be represented varies with setting and elements of the water cycle (Ajami et al., 2016). Moreover, while we recognize that heterogeneity exists in contextual features, many of these features, such as soil properties and hydrogeology, are poorly characterized across landscapes, but both features play important role in controlling water movement. Heterogeneity needs to be adequately represented without radically bloating the parameter space

of the models and thus data demand.

Furthermore, the heterogeneous physiographic factors co-evolve and covary (Troch et al., 2013) with complicated causal and non-causal connections (Faghmous and Kumar, 2014). The relationships of soil, terrain and vegetation are further conditioned on geologic and climate history and often do not transfer to other regions (Thompson et al., 2006). Consequently, training with insufficient data may result in overfitted data-driven models or many alternative DL models that cannot be rejected. On the

flip side, such complexity due to co-evolution also precludes a reductionist approach where all or most of these relationships are clearly described from fundamental laws.

Hydrologic problems fit poorly into the template of problems for which standard network structures (Section 3.2 in Shen18) are designed, i.e., purely image recognition or time series prediction problems, or a mixture of both. For example, catchment hydrologic problems are characterized by both spatially heterogeneous but temporally static attributes (topography and

hydrogeology) and temporal (atmospheric forcing) dimensions. Such input dimensions are not efficiently represented by with typical input dimensions of LSTM or CNN.

Because large and diverse datasets are needed for DL application, access to properly pre-processed and formatted present practical challenges. These steps of data compilation, pre-processing, and formatting often occupy too much unnecessary time for researchers. Many of the processing tasks for images cannot be handled by individual research groups. Compared to the

DL community in AI and chemistry, etc., DL learning community in hydrology is not sufficiently coordinated, resulting in significant waste of effort and "reinvention of wheels".

Deep generative models such as GANs can be used for the stochastic generation of natural textures. This capability has recently led to methodological advances in subsurface hydrology (Laloy et al., 2017, 2018; Mosser et al., 2017) where the ability to efficiently and accurately simulate complicated geologic structures with given (non-Gaussian) geostatistical properties is of

paramount importance for uncertainty quantification of subsurface flow and transport models. However, amongst other directions for future research, more work is needed (i) to generate the complete range of structural complexity observed in geologic layers, (ii) deal efficiently with large 3D domains and (iii), account for various types of direct (e.g., observed geologic facies at a given location, mean property value over a specific area, etc) and indirect (e.g., measured hydrologic state variables to be used within an inverse modelling procedure) conditioning data in the simulation.

**4. A community roadmap toward DL-powered scientific advances in hydrology**

Despite the challenges articulated above, here we offer a shared vision for a community roadmap for advancing hydrologic sciences using DL (Figure 4). A well-coordinated community is much more efficient and powerful in resolving problems, as we have seen in other scientific endeavours. Montanari et al. (2013) noted, "future science must be based on an interdisciplinary approach" and "the research challenges in hydrology for the next 10 years should be tackled through a collective effort". We see that several steps are crucial in this roadmap: devising ways to integrate physical knowledge, use DL to infer unknown quantities, process-based models (PBMs) and DL, community approaches in sharing and accessing data, open and transparent model competitions, baseline models and visualization packages and an education program that introduces data-driven methods at various levels.

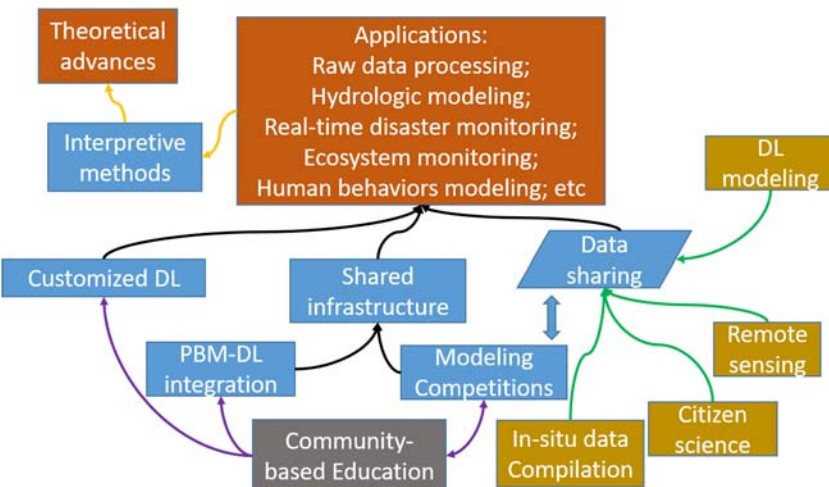

**Figure 4. A roadmap toward DL-powered scientific discovery in hydrologic science. Data availability can be increased by (green arrows) collecting and compiling existing data, incorporate novel data sources such as those collected by citizen scientists, remote sensing and modelled dataset. DL can be employed to predict data that are currently difficult to observe. The modelling competitions and the integration between PBM and DL will build important shared computing and analytic infrastructure, which, together with data sources, support a wide range of hydrologic applications. Interpretive methods should be attempted to extract knowledge from trained deep networks (orange arrows). Underpinning these activities is the enhanced, community-based educational program for machine learning in hydrology (purple arrows). However, these activities, especially the modelling competitions, might in turn feedback to the educational activity.**

**4.1. Integrating physical knowledge, process-based models, and DL models**

To address the challenge of data limitations (data quantity), we envision that a critical and necessary step is to more organically integrate hydrologic knowledge, process-based models, and DL. Process-based models, as they are derived from underlying physics, require less data for calibration than data-driven models. They can provide estimates for spatial and temporal data gaps and unobserved hydrologic processes. Well-constructed PBMs should also be able to represent temporal changes and

trends. However, because data-driven models directly target observations, these models may have better performance in locations and periods where data are available. Also, as discussed earlier, data-driven models are less prone to *a priori* model structural error than are PBMs. We should aim to maximally utilize the best features of each type of models.

There will be a diversity of approaches with which PBMs and data-driven models could be combined. Karpatne et al., (2017) compiled a list of approaches of what they collectively call "theory-guided data science," which include (i) using knowledge to design data-driven model; (ii) using knowledge to initialize network states; (iii) using physical knowledge to construct priors to constrain the data-driven models; (iv) using knowledge-based constrained optimization (although this may be difficult to implement in practice); (v) using theory as regularization terms for the data-driven model, which will force the model to respect these constraints; and (vi) learn hybrid models, where the data-driven method is used as surrogate for certain parts of the physical model. One may also impose multiple learning objectives based on the knowledge of the problem.

There are a multitude of potential approaches and this list can be further expanded to accommodate various objectives. First, we can focus on PBM errors (difference between PBM simulation and observations). Non-deep machine learning has already shown promise in correcting PBM errors. Abramowitz et al. (2006) developed an ANN to predict the error in net ecosystem exchange from a land surface model, and achieved 95% reduction in annual error. More importantly, an ANN trained to correct the error at one biome corrects the PBM in another biome with a different temperature regime (Abramowitz et al., 2007). In the context of weather forecasts, machine learning methods were used to learn the patterns from past forecasting errors (Delle Monache et al., 2011, 2013), leading to a 20 percent improvement in performance for events of similar characteristics (Junk et al., 2015). Their results suggest PBMs make structural errors that are independent of the state-variable regimes, although there is a lack of theories to guide the separation of error types. We envision that PBMs can better resolve the impacts of regime changes, while DL can better capture state-independent error patterns and do mild state-dependent extrapolations. A co-benefit of modelling PBM error is that insights are gained about the PBM. Using interrogative methods to reverse engineer what DL has learned about PBM error provides possible avenues for improving the underlying PBM processes.

Second, PBMs can augment input data for DL models. PBMs can be used to increase supervising data for DLs, for example, for climate or land-use scenarios that have not existed presently, to augment existing data. Given model structural uncertainty (uncertainty with hydrologic processes), frameworks like the Structure for Unifying Multiple Modeling Alternatives (Clark et al., 2015) and automated model building (Marshall, 2017) could be employed to generate a range of outputs. Furthermore, if the DL training is limited by available data, we may not be able to reject alternative DL models that could generate unphysical or unrealistic outputs. Providing PBM simulations as either training data or regularization terms can help to nudge DL models to generate physically meaningful outputs. The extent to which errors in PBM model results affects DL outcomes remains to be explored. A theoretical framework is lacking for separately estimating aleatory uncertainty (resulting from data noise), and epistemic uncertainty (resulting from PBM error and training data paucity) and uncertainty due to regime-shift. The advantages and disadvantages of various approaches could be systematically and efficiently evaluated in community-coordinated fashion.

### 4.2. Multi-faced, community-coordinated hydrologic modelling competitions

There are many possible approaches and many alternative model structures for using DL to make hydrologic predictions and to provide insight into hydrologic processes. In the light of these challenges, we argue that open, fast and standardized competitions are one effective way of accelerating the progress. The competitions can evaluate the models not only in terms of predictive performance but also the attainment of understanding.

The impacts of competitions are best evidenced in the community-coordinated AI challenges, which use a standardized set of problems. These competitions have strongly propelled the advances in AI. Some have argued that the contributions of the ImageNet dataset and the competition may be more significant than the winning algorithms arising from the contests (Gershgorn, 2017). New methods can be evaluated objectively and disseminated rapidly through competitions. Because the problems are standardized, they remove biases due to data sources and pre-processing. The community can quickly learn advantages and disadvantages of alternative model design through these competitions, which also encourage reproducibility.

We envision multi-faceted hydrologic modelling competitions where various models ranging from process-based ones to DL ones are evaluated and compared. The coordinators can, for example, provide a set of standard atmospheric forcings, landscape characteristics, and observed variables and provide targeted questions which participants must address. Importantly, the evaluation criteria should include not only performance-type criteria such as model efficiency coefficients and bias but also **qualitative/explanatory** ones such as explanations for control variables and model errors. Over-simplified or poorly-constructed models may provide more accessible explanations, but they might be misleading because the models may be overfitted to a given situation. Their simplicity may also constrain their ability to digest large datasets as a way of reducing uncertainty. Multi-faceted competitions allow us to also identify a "Pareto front" of explainability and performance and help rule out "false explanations". The objective of the competition is not only to seek the best simulation performance, but also those methods that offer deeper insight into hydrologic processes.

Another important value of competitions is that organizers will provide a standard input dataset and well-defined tasks. The entire community can leverage such effort. A substantial amount of effort is required to establish such a dataset, which may only be possible under a specifically designed project. Moreover, open competitions in the computer science field has produced well-known models such as AlexNet (Krizhevsky et al., 2012b), GoogLeNet (Szegedy et al., 2015), etc. These models serve as benchmarks and quick entry points for others. They can greatly improve reproducibility and the effectiveness of comparisons. Standard models, datasets and evaluation metrics will greatly improve DL adoption and hydrologic sciences.

### 4.3. Community-shared resources and broader involvement

A useful approach to address the major obstacle of data limitation is to increase our data repositories and to open access to existing data. Data value can be greatly enhanced by centralized data compilations, a task many institutions are already

undertaking. For example, the Consortium of Universities for the Advancement of Hydrologic Science, Inc. (CUAHSI) provides access to large amounts of hydrologic data (CUAHSI, 2018a). As another example, in 2015, a project called Collaborative Research Actions (Endo et al., 2015) was proposed in Belmont Forum, which is a group of the world's major and emerging funders of global environmental-change research. Many scientists from different countries join the project and

focus on the same issue, Food-Energy-Water Nexus. They shared their data (heterogeneous data) and research results from different regions.

ML has already been used to create useful hydrologic datasets such as soil properties (Chaney et al., 2016; Schaap, 1999) and landcover (Helber et al., 2017; Zhu et al., 2017) and cyclones (Liu et al., 2016). We envision there will continue to be significant progress in this regard, and the key to success will be the availability of ground-truth datasets. Using data sharing standards

will advance data sharing across domains (WaterML2, 2018). Providing access to data through web services, such as used by CUAHSI, negates the problem of storing data in a single location and enhances discoverability. Data brokers also provide more channels to share experiences, scholarly discussions, and debates along with the generation of data.

An important area where DL is expected to deliver significant value is the analysis of big and sub-research-quality data such as those collected by citizen scientists. Many aspects of the water cycle are directly accessible by everyone. Citizen scientists

already gather data about precipitation (CoCoRaHS, 2018), temperature, humidity, soil moisture, river stage (CrowdHydrology, 2018), and potentially groundwater levels. These quantities can be measured using inexpensive instruments such as cameras, pressure gauges and moisture sensors. Volunteer scientists can also be solicited for data in places where such data can best reduce the uncertainty of the DL model, as in a framework called active learning (Settles, 2012). Social data have been used to help monitoring flood inundation (Sadler et al., 2018; Wang et al., 2018). Crowd-sourced data have played roles

in DL research, where a large but noisy dataset was argued to be more useful than a much smaller but well-curated dataset (Huang et al., 2016; Izadinia et al., 2015). Even though there are problems related to data quality which can be overcome using AI approaches. An important co-benefit of involving citizen scientists is the education and outreach to the public. The active engagement is much more effective when the public has a stake in the research outcomes.

### 4.4. Education

A major barrier to realizing the benefits of data science and DL lies in our undergraduate and graduate curriculum. Little in hydrologists' standard curriculum prepares them for a future with substantially more data-driven science. Statistical courses often do not cover machine learning basics, while data mining courses offered by computer science departments lack the connections to the water discipline. Given the interdisciplinary nature of hydrology, it has been long recognized that it takes a community to raise a hydrologist (Merwade and Ruddell, 2012; Wagener et al., 2012). We propose a concerted effort by

current hydrologic machine learning researchers, with participation from computer scientists, to pool and share educational content. Such effort will form the basis of a hydrologic data mining curriculum and leverage the wit of the community. Collaborations may form through either grassroot collaborations or institutionally-supported education projects, e.g.,

(CUAHSI, 2018b). The open competitions would be a great source of education materials. A diversity of models that have been evaluated and contrasted help clarify Pros and Cons of different methods. Shared datasets, DL algorithms and data pre-processing software can be leveraged in classrooms.

As with the design of any education effort, it is important to consider inclusiveness and diversity. Especially, for hydrologic DL, the source field of AI appears to have an extremely poor track record of gender balance (Simonite, 2018). The reason to such imbalance could be rooted in introductory computer science classes, undergraduate curriculum and social stereotypes. Research has found that the introductory computer science classes, especially those taken by non-majors, are instrumental in developing a desire to stay in the field (Lehman, 2017). In addition, the portrayal of gender stereotypes regarding computing and the increase in weed-out courses (Aspray, 2016) have both discouraged women students in computer science (Sax et al., 2017). To counter such negative impacts, the introductory courses in the curriculum need to assume little prior programming experiences. Special attention must be paid by the educators to shatter the stereotypes. On the other hand, the richness of natural beauty in hydrology and the connection between data and the real world may be employed to help bridge the gender gap.

## 5. Concluding remarks

In this opinion paper, we argue that hydrologic scientists ought to give thoughts to a research avenue that complements traditional approaches, wherein DL-powered data mining is used to generate hypotheses, predictions, and insights. Although in the past there may have been strong reservations toward black-box approaches, recent efforts have been put in the interpretation and understanding of deep learning networks. The black-box perception of ML is perhaps a self-reinforcing curse on this juvenile field, as a rejection of the research avenue based on this perception will, in turn, jeopardize the development of more transparent algorithms. Progress in hydrology and other disciplines show that there is substantial promise in incorporating DL into hydrologists' toolbox. However, challenges such as data limitation and model variability demand a community-coordinated approach.

We have also argued for open hydrologic competitions that emphasize both performance and explainability. These competitions will greatly improve the growth of the field as a whole. They serve as valuable "organizing events", where different threads in algorithm development, model evaluation and comparison, reproducibility tests, dataset compilation, resource sharing and community organization all come to a convergence, to spur growth in the field.

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
