# Peer review of "HESS Opinions: Incubating deep-learning-powered hydrologic science advances as a community"

_Hydrology and Earth System Sciences, 2018_

## Short Comment (SC1) · 30 Apr 2018

1) Even though the word "hydrology" is meant for science ("logy") of water ("hydro"), in the current version of the manuscript, the boundary between water sciences and hydrological/hydrologic sciences is not visible. As per the title of the manuscript, the manuscript is about deep learning in water sciences. However, the content of the manuscript is merely constrained to hydrological sciences (e.g., see P-1 LN-22).

2) What is meant by "HESS" opinions? Are the opinions echoed in the manuscript represented by the journal office? Do the authors represent the journal office? Do the authors represent the editorial board? What is the expected outcome of the review

process of this manuscript? If the null hypothesis is that the HESS opinions are always published regardless of the review process, what should be the appropriate alternate hypothesis that needs to be tested for a given significance level? Are the available data sufficient to conduct this hypothesis testing?

3) As per the authors, deep learning, which has gained widespread attention since 2012(see P-2 LN-1), is a suite of tools centering on artificial neural networks. Is there a specific reason for the authors to prefer the year of 2012? In my opinion, the fusion of information theory and deep learning in hydrological sciences was well rooted even before 2012. Therefore, an appropriate reference is needed to support the authors' statement.

4) The titles of some of the subsections are not acceptable at a significance level of 5%. For example, the tile of subsection 2.1 is with more data, opportunities arise. What is the HESS opinion on writing titles for sections/subsections?

5) As per the authors, compared to classical DL problems, hydrology has a unique set of challenges that are research opportunities for DL (See P-10 LN-16). In the subsequent sentence, the authors state that DL research has not cover these questions extensively. What are those questions? I think, the paragraph (P-10 LN-16) needs to be re-written.

6) As per the authors, DL models have already been used as surrogate models for PBMs, but many novel ways that couple the two (i.e., PBMs and DLs) should be investigated (see P-11 LN-18). In my opinion, this has already been investigated in one of the PBMS (SWAT) in hydrology.

7) As per the authors, the evidence is mounting that when given "enough data", DL can provide the "unique ability" to automatically extract features, sometimes "better than human experts" do(see P-4 LN-28). Subsequent to this statement, the authors provide few bulleted points. What are meant with those bulleted points? Are the bulleted points meant to show that when given "enough data", DL can provide the "unique ability" to

automatically extract features, sometimes "better than human experts" do?

8) Should the abbreviation ML (P-5 LN-33) be introduced in one of the previous pages (see P-3 LN-6)?

9) On P-3 LN-20, except for satellite-based data products of precipitation, references are given for all other large available datasets (e.g., soil moisture, evapotranspiration, and streamflows) mentioned in the manuscript. Is there a specific reason for not citing a research paper for satellite-based data products of precipitation?

10) With the emerging datasets, DL models can be built and trained to learn features, organizational patterns and relationships and predicts outputs given new input instances (P-3 LN-28).However, the authors are not advocating a whole transition to DL as some of the problems, specifically the problems with just not enough data to train DL-based models, could be best tackled by specifically designed earlier-generation models. I think, it would be more appropriate to show an example (may be in hydrology) of how to use DL models and how to use specifically designed earlier-generation models to avoid transition to DL.

11) Considering the number of authors listed in the manuscript and the quantity of the work carried out in the manuscript, I think, it becomes vital to list each author's contribution in the manuscript.

12) What is meant by citizen "scientists"? What is the minimum required qualification? Does the definition of citizen scientist vary spatially and temporally?

Minor Comments P-3 LN-19: should it be Srinivasan, 2013 P-4 LN-5 to P-4 LN-9: The language needs to be checked
* * *

---

## Referee Comment (RC1) · Anonymous Referee #1 · 6 May 2018

General comments

This article discusses the potential benefit of deep learning models to let emerge knowledge about water science systems from hydrological data. The paper is well written, the opinion is clearly stated and the authors present their arguments based on their expertise and their understanding of deep learning techniques. I'm wondering to what extent this is new and original compared to the opinion paper of Marçais & Dreuzy (2017 - see reference below). For example, the figure presented in this former article expressly conveys the idea that DL methods could enhance the unraveling of hydrological properties from data which is the core of this current article.

[Figure]

Marçais, J., & de Dreuzy, J. R. (2017). Prospective interest of deep learning for hydrological inference. Groundwater, 55(5), 688-692.

I also feel that DL techniques and especially why it does work so well is still not understood by computer scientists and mathematicians. However, this article can give the impression that the "DL reasons of success" are now understood (see specific comments) paving the way for knowledge discovery in water sciences through its use. I would consider being more cautious about that as the understanding of the specific properties of DL models compared to more traditional statistical learning models is still an active area of research. This does not mean that DL has not to be widely tested for hydrologic purposes.

Specific comments

Page 1 L.20 Could you specify articles where DL shows capacities for scientific discovery?

Page 2 L.9-16. The paragraph gives the impression that DL is a "plug and play" model whereas to my knowledge building a DL model still requires intensive computer scientists' knowledges and requires use of GPUs.

Page 2 L.24. I don't think that generalization capacities of DL come from its interpolation capability. Indeed, classical neural networks have been proven (see citation below) to be universal interpolators but they do not generalize well.

Hornik, K., Stinchcombe, M., & White, H. (1989). Multilayer feedforward networks are universal approximators. Neural networks, 2(5), 359-366.

Page 3 L.25. I agree that increase in environmental data opens new opportunities for data-driven techniques in general and particularly for DL techniques. Along with the development of spatialized, remote sensing data, I would also insist on the development of environmental observatories that collect a lot of time series, monitoring data even though they are site specific. These two types of data are complementary to

advance through knowledge discovery in hydrology.

Page 6. L.6-17. This paragraph is intended to bridge the link between interrogative techniques brought in DL by the "AI neuroscience" subdiscipline and the potential of DL for knowledge discovery in water sciences. If the arguments tend to prove that such interrogative techniques enlighten the way the architecture of DL works, it does not explain the success of DL in itself. For example, the sentence L.13: "activations of recurrent neural networks can be visualized to show the control domain of certain cells, which explains its functioning" is not correct. This only explains the functioning of the architecture of the DL, not the reason of success of such a method.

There is some literature exploring the need for explanation of DL techniques. For Convolutional Neural Networks (CNNs), their understanding can be linked with wavelet theory (see reference below). Especially their capacity to extract invariants through a lot of different scale in high dimensional datasets but this is still a subject of active research. This capacity could explain their generalization capabilities especially for image datasets.

Mallat, S. (2016). Understanding deep convolutional networks. Phil. Trans. R. Soc. A, 374(2065), 20150203.

Page 7 L.8-18. It could be interesting to explore how DL techniques can improve hypothesis testing through an exploration of competing process-based models? The Structure for Unifying Multiple Modeling Alternatives (SUMMA) (see reference below) could be a start to generate process-based models with alternative hypotheses. For example, process-based models could be used to feed DL models with numerical generated data.

Clark, M. P., Nijssen, B., Lundquist, J. D., Kavetski, D., Rupp, D. E., Woods, R. A., ... & Arnold, J. R. (2015). A unified approach for process‐based hydrologic modeling: 1. Modeling concept. Water Resources Research, 51(4), 2498-2514.

[Figure]

Page 10 L.20-Page 11 L.24. I would add to this list the fact that water sciences provide to DL a unique challenge because hydrologic data are intrinsically heterogeneous. Building a model able to integrate these heterogeneous data might be the key toward knowledge discovery in water sciences and toward big progresses in AI.

---

## Referee Comment (RC2) · MF McCabe (Referee) · 10 May 2018

Review of Shen et al. (2018) "HESS Opinions: Deep learning as a promising avenue towards knowledge discovery in water sciences"

Overview

The opinion article by Shen et al. (2018) aims to provide a perspective on the opportunities that deep learning may provide to the water sciences discipline. This is certainly a topic of much current interest and relevance to the community, as it offers possible new pathways for system interpretation and understanding. As such, I was very keen

to read and review this contribution, with hopes of 1) learning more about deep learning applications in the "water sciences", and 2) identify some practical outcomes that could be relevant to my own (relatively broad) research interests.

Since I was not previously aware of this type of "Opinions" forum in HESS, I was expecting something more akin to a "Review Article", where the advances in machine learning (being delivered in this case by deep learning) would be illustrated through some relevant applications and examples. As such, I was a little disappointed that this was not the intent of this paper. The paper is precisely as the title dictates: an opinion article. Having, and expressing, an opinion is great: but for it to appear as a published article, it should ideally be supported by a strong, reasoned and defensible position that counters competing arguments via illustration of its superiority (or at least equivalence). Deep learning may be (and I believe it is) "a promising tool toward knowledge discovery in [the] water sciences". But simply stating it and illustrating with some examples where it has worked before is not the way to convince a new audience.

What is presented is a brief description of deep learning, a rather concise historical review of "machine learning" applications in hydrology (e.g. SVM, CART, RFs; which actually have quite an extensive history in hydrology, and especially remote sensing that could be detailed further), an expression of the need for data-driven science in contrast to a more classical (physics-based) approach, and an overview of some of the unique challenges that the water sciences present (which do not seem particularly unique if posed across the earth sciences). However, none of the expressed opinions are particularly revolutionary ideas: hydrology (and related fields) already provides many examples of data-driven science, black-box modeling applications, and novel statistical approaches to divine process insights. What would be good to see is how deep learning transcends these, or at the least, builds upon them to provide an avenue for new insights and investigation into "hydrological" processes.

Overall, I think there is a missed opportunity here to provide a perspective that could potentially garner significant interest in the community. To do this, the authors could

expand on a possible road-map on future directions (and obstacles) for deep learning applications, and also provide a demonstration of some analogous examples (perhaps from other disciplines, if not from hydrology directly) that could be relevant to "water science" applications. It's my hope that the authors can consider some of my comments in adapting their opinion piece – and ultimately attract the impact such a topic deserves.

Comments (in no specific order of importance or logical sequence).

* The title is very broad, with "water sciences" encapsulating a wide range of possible research avenues. I guess this is fine, as I agree that deep learning has broad application, but I wonder whether it might help to focus this discussion on "hydrological" sciences instead, and illustrate with some demonstrations of where this approach might deliver upon its potential. If the title is retained, it would need a much broader description of approaches and applications that could be explored. The authors might wish to review the recent work of Marcais and de Dreuzy (2017), who present a brief introduction to deep learning, focused on some more specific applications (calibration, hypothesis testing, etc.).

* I would remove the repeated statement (see line 24 as an example) of "...we lay out several opinions shared by the authors". In fact, I'd remove the use of "opinions" throughout the manuscript (Pg3-L4; Pg3-L9; Pg8-L25; Pg11-L26 etc.) completely and just focus on the presentation of ideas. As an alternative, use instead "Here we propose...". However, it is assumed that all co-authors are in agreement with the content of the paper, so there's no need to remind the reader of this.

* The five points listed in the abstract lean a little towards motherhood statements. Some specificity here would be great. Outlining what "may" happen seems a bit counterproductive. If this is a strongly held "opinion", this should be reflected in the content of the paper. For instance, "Deep learning will revolutionize our understanding of XYZ..." or "Deep learning offers an entirely new approach to ABC...". At the least, these statements need to be supported throughout the manuscript by a clear and rational review of how (precisely) deep learning will deliver upon them. Point 4 is probably the most important here, and the manuscript could really be built up around this (a point I will discuss below). I do not really understand Point 5 i.e. we need hydrology-customized methods for interpreting knowledge provided by deep learning? Isn't one of the points of deep learning to provide new knowledge for interpreting hydrological processes? Are you suggesting that it can do this, but we aren't able to understand it? Perhaps it's just me, but I find this a bit confusing.

* Regarding Point 4. To me, this represents the key issue that much of the paper can be built around. Deep learning has potential, but there are some specific challenges that hydrological sciences present that need to overcome or addressed. These are detailed somewhat in Section 4, but so much more could be written and the ideas expanded upon. For deep learning to have an impact in "water science", it is precisely issues like these (and this list is not comprehensive) that need to be considered. It would be great if you could structure your paper to examine these in more detail (if not provide possible solutions or avenues to address them). At the moment, the paper basically says that deep learning is a great technique that has much potential to provide new insights and understanding – BUT – there are some pretty serious roadblocks and challenges (not unique) to hydrological sciences that need to be addressed first. It's a big "but", especially if no attempt to provide a pathway to addressing them is offered. The real value of this opinion piece could be to provide some roadmaps towards these. At the least, a number of the "questions" presented in this section can be examined in greater detail, with examples drawn from the existing literature to showcase earlier or preliminary efforts.

* The paper could really use a review of the structure combined with a sharper focus on the deep learning applications (to hydrology/water science) in general. The entire Overview section reads as a Deep Learning review, rather than an exploration of its application to water sciences. Section 2.2. could probably be incorporated into the Overview/Introductory section instead of standing alone. Further, while an introduction to the technical concept is certainly required (and also needs attention), there's not much in the way of expounding on knowledge discovery. Just as illustrating some examples in other disciplines is relevant and required, so too is exploring those applications already examined in the "water sciences" through some recent literature (see your own listed examples on Pg3-L3 as well as on Pg5-L22-28). Providing some brief review of these applications may serve to demonstrate the value of your opinion. There are also quite a few others (see Agana and Homaifar, 2017 10.1109/SECON.2017.7925314)

* Following this point, the companion paper of Shen (2017), purports to provide a more comprehensive technical background (it is not listed in the bibliography). I was able to find this on arxiv (https://arxiv.org/ftp/arxiv/papers/1712/1712.02162.pdf) with the title "A trans-disciplinary review of deep learning research for water resources scientists". While only skimming that paper, I can see that it addresses many of my criticisms of this manuscript, in that it provides the needed level of technical background, disciplinary context and demonstration via examples that I was hoping for. The obvious question then is what additional value this manuscript offers in light of that work? I will leave it up to the authors (and editor) to make that assessment [but in the same vein, the EOS article by Shen, 2018, https://doi.org/10.1029/2018EO095649 seems another example of an opinion article on this topic?).

* Page 2, Line 15-16. This sentence is unclear to me.

* Some of the short-comings of GANS should also be mentioned: especially their ability to be "easily fooled" (see https://arxiv.org/pdf/1801.00553.pdf, https://arxiv.org/abs/1710.09762 and many other similar papers). Are there implications to water sciences in this – especially for automated approaches used in prediction systems? What other drawbacks of deep learning may impair their uptake and development?

* Other papers that might be of interest to the authors (indeed, see Volume 55, Issue 5

of Groundwater):

Chen and Wang (2018) "Recent advance in earth observation big data for hydrology" https://doi.org/10.1080/20964471.2018.1435072

Frere (2017) "Revisiting the Relationship Between Data, Models, and Decision‐-Making" https://doi.org/10.1111/gwat.12574

Lary et al. (2016) "Machine learning in geosciences and remote sensing" https://doi.org/10.1016/j.gsf.2015.07.003

Marshall (2017) "Creativity, Uncertainty, and Automated Model Building", https://doi.org/10.1111/gwat.12552

Lidard et al. (2017) "Scaling, similarity, and the fourth paradigm for hydrology", https://doi.org/10.5194/hess-21-3701-2017

Anderson (2008) "The end of theory: the data deluge makes the scientific method obsolete" https://www.wired. com/2008/06/pb-theory/

McCabe et al. (2017) "The future of Earth observation in hydrology", https://doi.org/10.5194/hess-21-3879-2017

* Since I'm familiar with that last reference, I highlight some of the discussion therein on machine learning approaches in general, particularly on Page 3902 (n.b. it may also be worth reviewing some of the mentioned references in an attempt to provide context of machine learning based hydrological applications - and where deep learning will fit into that): "Despite this remarkable confluence of data science and remote sensing, one can still resist the narrative that there is no problem that a sufficiently complex machine-learning algorithm cannot unravel given enough data (Anderson, 2008). If this were the case, there would be no need for domain expertise to understand current and future challenges in hydrology: the dilettante will have prevailed (Klemeš, 1986). Indeed, there remain several obstacles to any predicted ascension of a completely data-driven approach to hydrology. Observations of the hydrosphere often have

a spatio-temporal structure that emerges in the form of correlations between variables, but this correlation may not necessarily imply causality. Therefore, being able to draw strong deterministic conclusions about the behaviour of hydrologic systems based on data-driven methods often requires prior knowledge (and understanding) of the physical processes (Faghmous and Kumar, 2014)." This is relevant to your Section 2.4 and elsewhere.

* Your Section 4 provides an excellent launching point to really expand on some of these ideas and challenges (see above), and I would encourage you to use these (and build upon them) to structure this opinion piece around. Of course, it should be recognized that the problems highlighted here are not particular to deep learning, but to hydrological inference and understanding broadly, and that there has been much effort directed towards novel statistical approaches to address some of these (which would be worth mentioning, or at least providing some context).

* I'm not convinced that Section 2.4 is essential to this paper – or at least it can be presented differently. Advocating the role of data-driven approaches is not a new concept in hydrology (see some of the papers above for reviews) – nor is it especially controversial. It is not like modelers act in isolation – data is an integral part of that process. As with the use of machine learning approaches, data-driven knowledge discovery has a rich history in hydrology, which may be worth reviewing. Certainly there are many examples of ANN type models outperforming their physically-based counterparts. But I'm not sure what the intent of this section is? Either way, it is also not immediately clear (or demonstrated) that deep learning offers a better path towards achieving this "goal" than the myriad of techniques already being used.

* Likewise, I'm not sure what the purpose of Section 3.2 is? The last paragraph in particular (Pg10-L6-15) invokes a lot of hand-waving.

There are a number of other questions I have and handwritten annotations I have made on the paper that are not included in this review. My overall impression is that the

paper needs some considered thought not just on its structure, but on how it attempts to present the "opinion" that deep learning is a promising tool in hydrology. While I'm an advocate of your perspective here, in reading the manuscript, I found little to convince me that this approach presents a radical new angle to anything that has come before it. I hope that the authors can address some of these comments and further refine the contribution, as I think it is a topic that will be of considerable interest to the community.

Matthew McCabe
* * *

---

## Author Comment (AC1) · 4 Jun 2018

Dear Sivarajah Mylevaganam,

Thanks for the online interaction discussion. Please see attached response to your comment.

"1) Even though the word "hydrology" is meant for science ("logy") of water ("hydro"), in the current version of the manuscript, the boundary between water sciences and hydrological/hydrologic sciences is not visible. As per the title of the manuscript, the manuscript is about deep learning in water sciences. However, the content of the

manuscript is merely constrained to hydrological sciences (e.g., see P-1 LN-22)."

After some collective deliberation, we indeed plan to change the title to hydrology rather than water sciences. Thanks.

"2) What is meant by "HESS" opinions? Are the opinions echoed in the manuscript represented by the journal office? Do the authors represent the journal office? Do the authors represent the editorial board? What is the expected outcome of the review process of this manuscript? If the null hypothesis is that the HESS opinions are always published regardless of the review process, what should be the appropriate alternate hypothesis that needs to be tested for a given significance level? Are the available data sufficient to conduct this hypothesis testing?"

HESS Opinions papers do not represent journal opinions. The reviewer is referred to some previous HESS Opinion papers. https://www.hydrol-earth-syst-sci.net/special_issue62.html https://www.hydrol-earth-syst-sci.net/18/2615/2014/hess-18-2615-2014.pdf https://www.hydrol-earth-syst-sci.net/13/157/2009/hess-13-157-2009.pdf https://www.hydrol-earth-syst-sci.net/20/3739/2016/hess-20-3739-2016.pdf Per HESS website https://www.hydrology-and-earth-system-sciences.net/about/manuscript_types.html ïĆğ Opinion articles discuss a topical aspect of hydrology. These articles are not peer reviewed in the traditional sense, but they are discussed openly in HESSD so as to stimulate an open debate among peers on new ideas, views, or perceptions in hydrology. Opinion articles will be published under the heading "HESS Opinions" and are handled by one of the executive editors. Opinion articles are generally invited, but authors with ideas for an opinion paper are encouraged to contact an executive editor. The manuscript title should start with "HESS Opinions:".

"3) As per the authors, deep learning, which has gained widespread attention since 2012(see P-2 LN-1), is a suite of tools centering on artificial neural networks. Is there a specific reason for the authors to prefer the year of 2012? In my opinion, the fusion

of information theory and deep learning in hydrological sciences was well rooted even before 2012. Therefore, an appropriate reference is needed to support the authors' statement."

2012 is when DL first won a major competition and started to garner attention, including the Merck competition and the ImageNet. Although there have always been researchers in this area, the winning of these two competitions added big fuels to the fire. Yes this point will be explained more in the revised version.

"4) The titles of some of the subsections are not acceptable at a significance level of 5%. For example, the tile of subsection 2.1 is with more data, opportunities arise. What is the HESS opinion on writing titles for sections/subsections?"

It is not certain what was referred to here. This sections are named as summaries of the section. We'd appreciate if more clarity is given on what is wrong with this subsection title. With or without this comment, this subsection title may be revised as part of the whole revision endeavor

"5) As per the authors, compared to classical DL problems, hydrology has a unique set of challenges that are research opportunities for DL (See P-10 LN-16). In the subsequent sentence, the authors state that DL research has not cover these questions extensively. What are those questions? I think, the paragraph (P-10 LN-16) needs to be re-written."

"These questions" means the points that immediately follow this paragraph. Yes, we will revise the wording to avoid this confusion.

"6) As per the authors, DL models have already been used as surrogate models for PBMs, but many novel ways that couple the two (i.e., PBMs and DLs) should be investigated (see P-11 LN-18). In my opinion, this has already been investigated in one of the PBMS (SWAT) in hydrology."

There have been limited studies that integrate ML with PBM. One of such examples

is the following reference, which is co-authored by one of the co-authors of the current paper. It is also perhaps what was referred to here. Mekonnen, B. A., Nazemi, A., Mazurek, K. A., Elshorbagy, A., & Putz, G. (2015). Hybrid modelling approach to prairie hydrology: fusing data-driven and process-based hydrological models. Hydrological Sciences Journal, 60(9), 1473–1489. https://doi.org/10.1080/02626667.2014.935778 In the above paper, SWAT is used to represent the runoff generation from contributing areas and the ANN model is used to represent the nonlinear overflow generation from the non-contributing areas. However, this is only one way such coupling is done. There are many other ways in many different applications, such as partially replace the functions for each other, use PBM as inputs to DL, use PBM as constraints for DL, etc. We will include this paper as a citation, but there are so many other things to do in this regard. We will add some examples of this in the revision.

"7) As per the authors, the evidence is mounting that when given "enough data", DL can provide the "unique ability" to automatically extract features, sometimes "better than human experts" do(see P-4 LN-28). Subsequent to this statement, the authors provide few bulleted points. What are meant with those bulleted points? Are the bulleted points meant to show that when given "enough data", DL can provide the "unique ability" to automatically extract features, sometimes "better than human experts" do?"

Indeed here we can use a sentence to transition more naturally. We will add the following sentence: 'The performance gain by DL can be witnessed by an increasing number of competition wins by DL-based models and adoption in the mainstream information technology industry.'

"8) Should the abbreviation ML (P-5 LN-33) be introduced in one of the previous pages (see P-3 LN-6)?"

We will remove this abbreviation as it is not needed. Thanks for pointing it out.

"9) On P-3 LN-20, except for satellite-based data products of precipitation, references are given for all other large available datasets (e.g., soil moisture, evapotranspiration,

and streamflows) mentioned in the manuscript. Is there a specific reason for not citing a research paper for satellite-based data products of precipitation?"

The reason is there were many precipitation products. However, we will add some references there, too.

"10) With the emerging datasets, DL models can be built and trained to learn features, organizational patterns and relationships and predicts outputs given new input instances (P-3 LN-28).However, the authors are not advocating a whole transition to DL as some of the problems, specifically the problems with just not enough data to train DL-based models, could be best tackled by specifically designed earlier-generation models. I think, it would be more appropriate to show an example (may be in hydrology) of how to use DL models and how to use specifically designed earlier-generation models to avoid transition to DL."

Yes this is a good suggestion. We can add an example in the revision.

"11) Considering the number of authors listed in the manuscript and the quantity of the work carried out in the manuscript, I think, it becomes vital to list each author's contribution in the manuscript."

This would not be necessary. We again refer the reviewer to earlier HESS opinion papers. None of those papers did this.

"12) What is meant by citizen "scientists"? What is the minimum required qualification? Does the definition of citizen scientist vary spatially and temporally? Minor Comments P-3 LN-19: should it be Srinivasan, 2013 P-4 LN-5 to P-4 LN-9: The language needs to be checked"

Citizen scientists mean volunteers who are willing to spend their time to help provide relevant measurements and data. Good point that we should not assume this term is generally known or accepted. More clarification will be added.

---

## Author Comment (AC2) · 4 Jun 2018

*AR1*
*General comments This article discusses the potential benefit of deep learning models to let emerge knowledge about water science systems from hydrological data. The paper is well written, the opinion is clearly stated and the authors present their arguments based on their expertise and their understanding of deep learning techniques. I'm wondering to what extent this is new and original compared to the opinion paper of Marçais & Dreuzy (2017 - see reference below). For example, the figure presented in this former article expressly conveys the idea that DL methods could enhance the unraveling of hydrological properties from data which is the core of this current article.*
*Marçais, J., & de Dreuzy, J. R. (2017). Prospective interest of deep learning for hydrological inference. Groundwater, 55(5), 688-692.*

The roles taken by these papers are very different. We are cheerful to see others echoing the same enthusiasm for DL in hydrology. We would welcome others to have discussion and work together on this topic. In terms of the paper, though, we see significant differences between these articles, which is summarized in the table below. In our opinion, the Marcasis and de Dreuzy 2017 (MD17) paper was a timely and welcomed first "call into the wild" (despite that one of the co-authors of this Opinion paper has had a DL paper, Tao et al. in 2016). While it has fulfilled the mission, that paper was very brief and had a different focus. It did not explain why DL could unravel hydrologic properties. It did not mention interrogative studies which is a crucial part of our argument. It also did not discuss what we need to do as a community to incubate such research. In this article we gather from our past working to voice some enthusiasm as well as challenges.

However, prompted by the reviewer (and other comments), we will revise this Opinion paper significantly to emphasize our main points, which are: (1) DL+interrogative study is a valuable research avenue; (2) what challenges face the community and what we can do together to incubate DL research; (3) water resources present unique challenges and opportunities for DL.

Table. Difference between papers

| Paper | Unique ideas |
|---|---|
| This HESS Opinion | (As its title indicates, this is truly an opinion paper. We need to assume readers have access to Shen's review paper)
1. Opinion: DL is not a hype. Supported by a review of its solid progress, winnings of competitions and adoption in daily uses.
2. Proposition of the complementary, data-driven scientific avenue: the integration of interrogative studies into the avenue.
3. Following unique Opinions are about what we can do as a community:
a.     scientific methods: hypotheses come from machine learning. We do not pose an opinion before doing data mining. Difference from earlier ML: now we have DL to automatically extract features.
b.     call for open competition of DL in hydrology with criteria focusing on both performance and explainability
c.     collecting big data through data sharing and citizen scientists
6 (**to be enhanced**). Water science provide unique challenges and opportunities for DL.
7 (**to be added**): Roadmap toward DL-supported science discovery. & Practice challenges and research thrust as a community |
| Marcasis and Dreuzy 2017 | Main points: DL can be used for prediction issues; it may contribute to initial choice and alternatives of physical model structures; model reduction; emergent system properties; calibration. (however, each was only mentioned in one sentence).
Test on hydrologic numerical data; benchmarks |
| Shen. 2018 Review | 1. Technical details on ML and DL
2. Trans-disciplinary review of DL applications and experiences in sciences
3. Technical details of progress: interpreting DL and GANs
4. (revision) prospects for DL to help tackling grand challenges facing water sciences: inter-disciplinarity, human dynamics, data deluge (from novel sources), scaling and equifinality issues, non-unique inversions and high-dimensional, multi-modal data. |

*I also feel that DL techniques and especially why it does work so well is still not understood by computer scientists and mathematicians. However, this article can give the impression that the "DL reasons of success" are now understood (see specific comments) paving the way for knowledge discovery in water sciences through its use. I would consider being more cautious about that as the understanding of the specific properties of DL models compared to more traditional statistical learning models is still an active area of research. This does not mean that DL has not to be widely tested for hydrologic purposes.*

There have been some studies that looked at why DL is powerful, but the point is taken. We will add clarification to this regard on the lines of 'why DL works so well is not fully understood. there are some suggestions… However in water research it needs to widely before trusted.'

*Specific comments Page 1 L.20 Could you specify articles where DL shows capacities for scientific discovery?*

It was included in the review paper Shen2018. There were quite a few examples. Here, we will include some summary of these in the revised Opinion article.

*Page 2 L.9-16. The paragraph gives the impression that DL is a "plug and play" model whereas to my knowledge building a DL model still requires intensive computer scientists' knowledges and requires use of GPUs.*

Good point. We will add the following sentence to this sentence:

"*While showing many advantages, DL models will require substantial amount of computing expertise. The tuning of hyper-parameters, e.g. network size, learning rate, batch size, etc., often require a priori experiences and trial and error. The computational paradigm is also substantially different from ordinary hydrologists' educational background.*"

*Page 2 L.24. I don't think that generalization capacities of DL come from its interpolation capability. Indeed, classical neural networks have been proven (see citation below) to be universal interpolators but they do not generalize well. Hornik, K., Stinchcombe, M., & White, H. (1989). Multilayer feedforward networks are universal approximators. Neural networks, 2(5), 359-366. Page 3 L.25. I agree that increase in environmental data opens new opportunities for data-driven techniques in general and particularly for DL techniques. Along with the development of spatialized, remote sensing data, I would also insist on the development of environmental observatories that collect a lot of time series, monitoring data even though they are site specific. These two types of data are complementary to advance through knowledge discovery in hydrology.*

The original sentence was "*Moreover, the differentiable nature allows for greater success for interpolation and mild extrapolation, contributing to the strong generalization capability of DL.*". The differentiability "contributes" to the generalization ability, but that is not the sole reason. Other factors include improved architecture, regularization, big data, weights sharing etc., which were mentioned earlier. To avoid any confusion, this sentence will be revised as

"*Moreover, the differentiable nature allows for greater success for interpolation and mild extrapolation, partially contributing to the strong generalization capability of DL.*"

*Page 6. L.6-17. This paragraph is intended to bridge the link between interrogative techniques brought in DL by the "AI neuroscience" subdiscipline and the potential of DL for knowledge discovery in water sciences. If the arguments tend to prove that such interrogative techniques enlighten the way the architecture of DL works, it does not explain the success of DL in itself. For example, the sentence*

*L.13: "activations of recurrent neural networks can be visualized to show the control domain of certain cells, which explains its functioning" is not correct. This only explains the functioning of the architecture of the DL, not the reason of success of such a method. There is some literature exploring the need for explanation of DL techniques. For Convolutional Neural Networks (CNNs), their understanding can be linked with wavelet theory (see reference below). Especially their capacity to extract invariants through a lot of different scale in high dimensional datasets but this is still a subject of active research. This capacity could explain their generalization capabilities especially for image datasets.*

The interpretive study does not solely focus on "why DL was successful", and this is not really the point. The interpretive studies answer "what has DL learned". We would argue that for scientists the second question is more important than the first. To clarify, we will revise this part to separate out the two question separately.

*Mallat, S. (2016). Understanding deep convolutional networks. Phil. Trans. R. Soc. A, 374(2065), 20150203. Page 7 L.8-18. It could be interesting to explore how DL techniques can improve hypothesis testing through an exploration of competing process-based models? The Structure for Unifying Multiple Modeling Alternatives (SUMMA) (see reference below) could be a start to generate process-based models with alternative hypotheses. For example, process-based models could be used to feed DL models with numerical generated data.*
*Clark, M. P., Nijssen, B., Lundquist, J. D., Kavetski, D., Rupp, D. E., Woods, R. A., ... & Arnold, J. R. (2015). A unified approach for process-based hydrologic modeling: 1. ˘ Modeling concept. Water Resources Research, 51(4), 2498-2514*

*Page 10 L.20-Page 11 L.24. I would add to this list the fact that water sciences provide to DL a unique challenge because hydrologic data are intrinsically heterogeneous. Building a model able to integrate these heterogeneous data might be the key toward knowledge discovery in water sciences and toward big progresses in AI.*

Good point! This part will be expanded to include this point.

Our plan of re-organization:

Section 1. Overview

current overview, with more discussion about promising attributes of DL

Section 2. The emergence of a complementary research avenue

More  examples of DL success, some potential uses of DL in hydrology. Some possible interrogative study methods to show the promise.

Section 3. Challenges and opportunities

Expand on original section 4. There are many old and new challenges, many of which cannot be resolved by individual research groups: regionally-imbalanced dataset; strong heterogeneity and contextual variables; partial observations; computational challenges; data access; myriad configurations and "tricks"; lacking training data, especially unlabeled data; problem complexity; missing dynamics; large variation in performance based on DL configurations; Non-stationary world and increasing extremes are beyond previous observations.

Section 4. A community roadmap to DL-powered scientific advances in hydrology. How to solve challenges raised in Section 3 with a community-based approach

(i) synergy between PBM and DL
(ii) readily **accessible large dataset** with uniform formats: earth observations and monitoring networks. assimilate large amount of data to learn true patterns.
(iii) community-shared baseline DL models and data-processing pipelines
(iv) Open and transparent modeling **competitions** in water to facilitate algorithm comparisons, with evaluation on both **performance** and **interpretation** → we need to recognize the significant roles played by competitions in the development of DL research.
(v) Develop a baseline suite of DL interpretation and visualization software that support mainstream DL models, especially those that interpret the hidden layers.

Sections 3 & 4 will be greatly enhanced.

---

## Author Comment (AC3) · 4 Jun 2018

*AR3 McCabe*
*Review of Shen et al. (2018) "HESS Opinions: Deep learning as a promising avenue towards knowledge discovery in water sciences" Overview The opinion article by Shen et al. (2018) aims to provide a perspective on the opportunities that deep learning may provide to the water sciences discipline. This is certainly a topic of much current interest and relevance to the community, as it offers possible new pathways for system interpretation and understanding. As such, I was very keen to read and review this contribution, with hopes of 1) learning more about deep learning applications in the "water sciences", and 2) identify some practical outcomes that could be relevant to my own (relatively broad) research interests.*

*Since I was not previously aware of this type of "Opinions" forum in HESS, I was expecting something more akin to a "Review Article", where the advances in machine learning (being delivered in this case by deep learning) would be illustrated through some relevant applications and examples. As such, I was a little disappointed that this was not the intent of this paper. The paper is precisely as the title dictates: an opinion article. Having, and expressing, an opinion is great: but for it to appear as a published article, it should ideally be supported by a strong, reasoned and defensible position that counters competing arguments via illustration of its superiority (or at least equivalence).*

Many thanks to Dr. McCabe who gave very constructive criticism. Indeed this is an opinion paper which does not normally assume the role of a full review. Of course, some concise arguments are provided, but the function of a full review paper has been achieved in another open-access paper on arxiv. Few opinion papers carry a full review and we would like the paper to be shorter. We must take elements (in the form of summaries and abridged examples) from the review paper to support the arguments. In addition, this paper also has the important task fo discussing what the community has to do as a whole

During revision (if allowed), we plan to reorganize the paper in the following fashion:

Section 1. Overview

current overview, with more discussion about promising attributes of DL

Section 2. The emergence of a complementary research avenue

More examples of DL success, some potential uses of DL in hydrology. Some possible interrogative study methods to show the promise.

Section 3. Challenges and opportunities

Expand on original section 4. There are many old and new challenges, many of which cannot be resolved by individual research groups: regionally-imbalanced dataset; strong heterogeneity and contextual variables; partial observations; computational challenges; data access; myriad configurations and "tricks"; lacking training data, especially unlabeled data; problem complexity; missing dynamics; large variation in performance based on DL configurations; Non-stationary world and increasing extremes are beyond previous observations.

Section 4. A community roadmap to DL-powered scientific advances in hydrology. How to solve challenges raised in Section 3 with a community-based approach

(i) synergy between PBM and DL
(ii) readily **accessible large dataset** with uniform formats: earth observations and monitoring networks. assimilate large amount of data to learn true patterns.

(iii) community-shared baseline DL models and data-processing pipelines
(iv) Open and transparent modeling **competitions** in water to facilitate algorithm comparisons, with evaluation on both **performance** and **interpretation** → we need to recognize the significant roles played by competitions in the development of DL research.
(v) Develop a baseline suite of DL interpretation and visualization software that support mainstream DL models, especially those that interpret the hidden layers.

Sections 3 & 4 will be greatly enhanced.

*Deep learning may be (and I believe it is) "a promising tool toward knowledge discovery in [the] water sciences". But simply stating it and illustrating with some examples where it has worked before is not the way to convince a new audience*
*What is presented is a brief description of deep learning, a rather concise historical review of "machine learning" applications in hydrology (e.g. SVM, CART, RFs; which actually have quite an extensive history in hydrology, and especially remote sensing that could be detailed further), an expression of the need for data-driven science in contrast to a more classical (physics-based) approach, and an overview of some of the unique challenges that the water sciences present (which do not seem particularly unique if posed across the earth sciences). However, none of the expressed opinions are particularly revolutionary ideas: hydrology (and related fields) already provides many examples of data-driven science, black-box modeling applications, and novel statistical approaches to divine process insights. What would be good to see is how deep learning transcends these, or at the least, builds upon them to provide an avenue for new insights and investigation into "hydrological" processes.*

We do not think interrogative studies was part of the historical studies or even an aspect that was ever argued for in hydrology. These studies were motivated by the criticism that DL models are black boxes, and the computer science community and the general scientific community only recently have worked together to develop these interpretive methods.

DL are non-deep machine learning are substantially different in their capability, data demands and scope. Their differences have been detailed in some DL papers such as Goh et al., 2017, Tao et al., 2016, Fang et al. 2017 andis described at lengths in Shen 2017 (and a revised version of Shen's review paper that is to be posted on arxiv soon). First, the ability to automatically extract or engineer features is a new characteristic of DL that was not available before. The hidden layers provide a new way of providing scientific understanding. Second, the expressive power and the abilities to capture spatio-temporal dependences and high-dimensional data distributions are new possibilities that were not possible before. Transfer learning reduces the demand for data and is more widely exploited with DL.

These characteristics leads to a new plausible way of doing science as we are advocating here. However, it does seem like this has not come out very apparently in the manuscript. We will enhance this part of the discussion with summaries from Shen's review as well as examples not found from there.

*Overall, I think there is a missed opportunity here to provide a perspective that could potentially garner significant interest in the community. To do this, the authors could expand on a possible road-map on future directions (and obstacles) for deep learning applications,*

*and also provide a demonstration of some analogous examples (perhaps from other disciplines, if not from hydrology directly) that could be relevant to "water science" applications. It's my hope that the authors can consider some of my comments in adapting their opinion piece – and ultimately attract the impact such a topic deserves*
*Comments (in no specific order of importance or logical sequence).*

If a revision is permitted we will indeed add a New Section 4 as indicated above, which includes a roadmap of several important pieces. Some of this was covered in the original manuscript, but was not presented in the format of a road-map. Please see our first reply to Dr. McCabe. On the other hand, in this paper we would like to stress the importance of a community-coordinated approach towards solving these obstacles.

*\* The title is very broad, with "water sciences" encapsulating a wide range of possible research avenues. I guess this is fine, as I agree that deep learning has broad application, but I wonder whether it might help to focus this discussion on "hydrological" sciences instead, and illustrate with some demonstrations of where this approach might deliver upon its potential. If the title is retained, it would need a much broader description of approaches and applications that could be explored. The authors might wish to review the recent work of Marcais and de Dreuzy (2017), who present a brief introduction to deep learning, focused on some more specific applications (calibration, hypothesis testing, etc.).*

We will indeed change the title to focus on hydrology.

Marcasis and de Dreuzy will be added to the literature review. As described in our reply to reviewer #1, it is a brief "call into the wild".

*vv*
*\* I would remove the repeated statement (see line 24 as an example) of ". . .we lay out several opinions shared by the authors". In fact, I'd remove the use of "opinions" throughout the manuscript (Pg3-L4; Pg3-L9; Pg8-L25; Pg11-L26 etc.) completely and just focus on the presentation of ideas. As an alternative, use instead "Here we propose. . .". However, it is assumed that all co-authors are in agreement with the content of the paper, so there's no need to remind the reader of this.*

Good point. This sentence will be changed.

*\* The five points listed in the abstract lean a little towards motherhood statements. Some specificity here would be great. Outlining what "may" happen seems a bit counterproductive. If this is a strongly held "opinion", this should be reflected in the content of the paper. For instance, "Deep learning will revolutionize our understanding of XYZ. . ." or "Deep learning offers an entirely new approach to ABC. . .". At the least, these statements need to be supported throughout the manuscript by a clear and ratio nal review of how (precisely) deep learning will deliver upon them. Point 4 is probably the most important here, and the manuscript could really be built up around this (a point I will discuss below). I do not really understand Point 5 i.e. we need hydrologycustomized methods for interpreting knowledge provided by deep learning? Isn't one of the points of deep learning to provide new knowledge for interpreting hydrological processes? Are you suggesting that it can do this, but we aren't able to understand it? Perhaps it's just me, but I find this a bit confusing.*

We will revised our abstract with more clearly-defined statements.

In terms of the hydrology-customized interpretation method:

According to our summary of Shen 2017's review paper, there are several methods that have already been developed for standard image recognition problems (relevance backpropagation, approximation using interpretable models, and correlation-based analysis). These methods can be ported to water applications. However, scientists in other domains have been creative in devising problem-specific ways in interpreting the results. We believe water scientists need to do both. Especially, considering we will also need customized network structure and our applications will be diverse, some of the method will not work out of the box. Therefore, we expect customized interpretive methods to be necessary and will be an active area of research. Nevertheless, we will re-write this sentence to make it more clear:

(5) An important research need is to customize DL interpretive methods for hydrologic research.

*Regarding Point 4. To me, this represents the key issue that much of the paper can be built around. Deep learning has potential, but there are some specific challenges that hydrological sciences present that need to overcome or addressed. These are detailed somewhat in Section 4, but so much more could be written and the ideas expanded upon. For deep learning to have an impact in "water science", it is precisely issues like these (and this list is not comprehensive) that need to be considered. It would be great if you could structure your paper to examine these in more detail (if not provide possible solutions or avenues to address them). At the moment, the paper basically says that deep learning is a great technique that has much potential to provide new insights and understanding – BUT – there are some pretty serious roadblocks and challenges (not unique) to hydrological sciences that need to be addressed first. It's a big "but", especially if no attempt to provide a pathway to addressing them is offered. The real value of this opinion piece could be to provide some roadmaps towards these. At the least, a number of the "questions" presented in this section can be examined in greater detail, with examples drawn from the existing literature to showcase earlier or preliminary efforts.*

This is a wonderful suggestion. We believe the manuscript would benefit from such a re-structuring. As indicated by the previous comment, we are in the process of revising the manuscript and we anticipate the structure will look like this:

Section 3. Challenges and opportunities

Section 4. A community roadmap to DL-powered scientific advances in hydrology. How to solve challenges raised in Section 3 with a community-based approach

*The paper could really use a review of the structure combined with a sharper focus on the deep learning applications (to hydrology/water science) in general. The entire Overview section reads as a Deep Learning review, rather than an exploration of its application to water sciences. Section 2.2. could probably be incorporated into the Overview/Introductory section instead of standing alone. Further, while an in troduction to the technical concept is certainly required (and also needs attention), there's not much in the way of expounding on knowledge discovery. Just as illustrating some examples in other disciplines is relevant and required, so too is exploring those applications already examined in the "water sciences" through some*

*recent literature (see your own listed examples on Pg3-L3 as well as on Pg5-L22-28). Providing some brief review of these applications may serve to demonstrate the value of your opinion. There are also quite a few others (see Agana and Homaifar, 2017 10.1109/SECON.2017.7925314)*

As summarized in Shen 2017, there really are not a great deal of water applications for DL, and hence this paper. Those have been collected into that review paper. Here we will add a paragraph to summarize the findings, and add some examples. Again please note this paper moves beyond the review: we raise several opinions on how, together as a community, tackle the obstacles. Selling DL is not the only focus of this piece.

*\* Following this point, the companion paper of Shen (2017), purports to provide a more comprehensive technical background (it is not listed in the bibliography). I was able to find this on arxiv (https://arxiv.org/ftp/arxiv/papers/1712/1712.02162.pdf) with the title "A trans-disciplinary review of deep learning research for water resources scientists". While only skimming that paper, I can see that it addresses many of my criticisms of this manuscript, in that it provides the needed level of technical background, disciplinary context and demonstration via examples that I was hoping for. The obvious question then is what additional value this manuscript offers in light of that work? I will leave it up to the authors (and editor) to make that assessment [but in the same vein, the EOS article by Shen, 2018, https://doi.org/10.1029/2018EO095649 seems another example of an opinion article on this topic?).*

Please see the table in the reply to AR1 about the differences between these two papers. The review paper provides basics, literature review, and discussion of the hydrologic science challenges that DL could help address. The current Opinions paper focuses on the complementary scientific avenue, the challenges facing application of DL, and what we can do as a community to tackle these challenges. This paper is more forward looking and it is, as titled, an opinion paper. We agree this should be made more clear in the abstract and in the paper. On the other hand, the EOS article, constrained by 600 words limit, was to serve as an opener of a special issue in WRR. It stresses the argument that DL may one day become part of hydrology. There was barely any discussion in that one so it should not really compared.

Table. Difference between papers

| Paper | Unique ideas |
|-------|--------------|
| This HESS Opinion | (As its title indicates, this is truly an opinion paper. We need to assume readers have access to Shen's review paper)
1. Opinion: DL is not a hype. Supported by a review of its solid progress, winnings of competitions and adoption in daily uses.
2. Proposition of the complementary, data-driven scientific avenue: the integration of interrogative studies into the avenue.
3. Following unique Opinions are about what we can do as a community:
a.    (**to be enhanced**) incorporate PBM and DL
b.    scientific methods: hypotheses come from machine learning. We do not pose an opinion before doing data mining. Difference from earlier ML: now we have DL to automatically extract features.
c.    call for open competition of DL in hydrology with criteria focusing on both performance and explainability
d.    collecting big data through data sharing and citizen scientists
5 (**to be enhanced**). Water science provide unique challenges and opportunities for DL.
6 (**to be added**): Roadmap toward DL-supported science discovery. & Practice challenges and research thrust as a community |
| Marcasis and Dreuzy 2017 | Main points: DL can be used for prediction issues; it may contribute to initial choice and alternatives of physical model structures; model reduction; emergent system properties; calibration. (however, each was only mentioned in one sentence).
Test on hydrologic numerical data; benchmarks |
| Shen. 2018 Review | 1. Technical details on ML and DL, regularization, etc.
2. Trans-disciplinary review of DL applications and experiences in various disciplines of sciences
3. Technical details of progress: interpretive DL and GANs
4. (revision) prospects for DL to help tackling grand challenges facing water sciences: inter-disciplinarity, human dynamics, data deluge (from novel sources), scaling and equifinality issues, non-unique inversions and high-dimensional, multi-modal data. |

*\* Page 2, Line 15-16. This sentence is unclear to me*
*\* Some of the short-comings of GANS should also be mentioned: especially their ability to be*

*"easily fooled" (see https://arxiv.org/pdf/1801.00553.pdf, https://arxiv.org/abs/1710.09762 and many other similar papers). Are there implications to water sciences in this – especially for automated approaches used in prediction systems? What other drawbacks of deep learning may impair their uptake and development?*
*\* Other papers that might be of interest to the authors (indeed, see Volume 55, Issue 5 of Groundwater):*

We will add some discussion here. It is not that GANs are easily fooled, but deep networks can be fooled by small changes in inputs. GANs can actually be used to train networks more robustly, which is one of the co-authors focus of research [Ororbia].

*Chen and Wang (2018) "Recent advance in earth observation big data for hydrology" https://doi.org/10.1080/20964471.2018.1435072 Frere (2017) "Revisiting the Relationship Between Data, Models, and Decision R-˘ Making" https://doi.org/10.1111/gwat.12574 Lary et al. (2016) "Machine learning in geosciences and remote sensing" https://doi.org/10.1016/j.gsf.2015.07.003 Marshall (2017) "Creativity, Uncertainty, and Automated Model Building", https://doi.org/10.1111/gwat.12552 Lidard et al. (2017) "Scaling, similarity, and the fourth paradigm for hydrology", https://doi.org/10.5194/hess-21-3701-2017 Anderson (2008) "The end of theory: the data deluge makes the scientific method obsolete" https://www.wired. com/2008/06/pb-theory/ McCabe et al. (2017) "The future of Earth observation in hydrology", https://doi.org/10.5194/hess-21-3879-2017*

*\* Since I'm familiar with that last reference, I highlight some of the discussion therein on machine learning approaches in general, particularly on Page 3902 (n.b. it may also be worth reviewing some of the mentioned references in an attempt to provide context of machine learning based hydrological applications - and where deep learning will fit into that): "Despite this remarkable confluence of data science and remote sensing, one can still resist the narrative that there is no problem that a sufficiently complex machine-learning algorithm cannot unravel given enough data (Anderson, 2008). If this were the case, there would be no need for domain expertise to understand current and future challenges in hydrology: the dilettante will have prevailed (Klemeš, 1986). Indeed, there remain several obstacles to any predicted ascension of a completely data-driven approach to hydrology. Observations of the hydrosphere often have a spatio-temporal structure that emerges in the form of correlations between variables, but this correlation may not necessarily imply causality. Therefore, being able to draw strong deterministic conclusions about the behaviour of hydrologic systems based on data-driven methods often requires prior knowledge (and understanding) of the physical processes (Faghmous and Kumar, 2014)." This is relevant to your Section 2.4 and elsewhere*

Some of these points indeed resonate with what we have put forth in old Section 4 where we mentioned PBMs and DL models complement each other. We in fact do not argue for a completely data-driven scenario where domain expertise is of no use. This is not from a personal interest point of view, but we do not see water science presents so much data that can cover every aspect of the hydrologic and human water cycle. If we humor ourselves and imagine that such a scenario does occur, it may indeed be possible for DL models to predict everything more accurately than process-based models, but still PBMs are required for us, humans, to understand the causal relationships. Even data in the world may be not be sufficient to distinguish between causal and associative relationships.

15

In our revision we will make these points more clear, and will cite the above-mentioned literature so our viewpoints can be put into context.

*Your Section 4 provides an excellent launching point to really expand on some of these ideas and challenges (see above), and I would encourage you to use these (and build upon them) to structure this opinion piece around. Of course, it should be recognized that the problems highlighted here are not particular to deep learning, but to hydrological inference and understanding broadly, and that there has been much effort directed towards novel statistical approaches to address some of these (which would be worth mentioning, or at least providing some context)*

Thanks and should a revision be allowed, we will expand this section into a much bigger section. We will update this online response as revisions are done.

*I'm not convinced that Section 2.4 is essential to this paper – or at least it can be presented differently. Advocating the role of data-driven approaches is not a new concept in hydrology (see some of the papers above for reviews) – nor is it especially controversial. It is not like modelers act in isolation – data is an integral part of that process. As with the use of machine learning approaches, data-driven knowledge discovery has a rich history in hydrology, which may be worth reviewing. Certainly there are many examples of ANN type models outperforming their physically-based counterparts. But I'm not sure what the intent of this section is? Either way, it is also not immediately clear (or demonstrated) that deep learning offers a better path towards achieving this "goal" than the myriad of techniques already being used.*

As mentioned previously, we do not believe interrogative studies were mentioned before, as they are essentially a new sub-discipline. Therefore, while we will take most of the suggestions offered by the reviewer, here we beg to differ that this section is important. Indeed, we will outline the novelty more clearly.

*Likewise, I'm not sure what the purpose of Section 3.2 is? The last paragraph in particular (Pg10-L6-15) invokes a lot of hand-waving*

This is actually one of the roadmap toward collecting more data. Citizen scientists can help collect data about precipitation, groundwater depths and pressure, surface water stages, soil texture and other observable variables. Previously it was not possible to consume lot of the sub-research-grade data they produce. Now, with lots of data, the noise inherent in these data can be averaged out. With DL, we can infer concepts not possible before.

*There are a number of other questions I have and handwritten annotations I have made on the paper that are not included in this review. My overall impression is that the paper needs some considered thought not just on its structure, but on how it attempts to present the "opinion" that deep learning is a promising tool in hydrology. While I'm an advocate of your perspective here, in reading the manuscript, I found little to convince me that this approach presents a radical new angle to anything that has come before it. I hope that the authors can*

*address some of these comments and further refine the contribution, as I think it is a topic that will be of considerable interest to the community.*

Again we thank Dr. McCabe for constructive criticism. I think the main focus may be a little different from what Dr. McCabe had anticipated. Here not only do we want to argue the usefulness of DL (partially accomplished elsewhere), we want to address how the community as a whole can incubate DL research.

---

## Referee Comment (RC3) · K. A. Sawicz (Referee) · 15 Jun 2018

The manuscript titled "Deep learning as a promising avenue toward knowledge discover in water sciences" conveys the opinion that hydrology is a field well suited for deep learning and that deep learning techniques should be widely applied to increase our understanding of hydrologic systems.

While I agree with the authors' general premise and believe the article can have a great impact on the direction of the technical analysis of future hydrological studies, I also was disappointed to not see two primary topics. The first of which is a summary of how deep learning has been integral in other fields to increase knowledge discovery of other

fields of science. The second and more important topic would be the presentation of a general framework of how to apply the techniques of deep learning to open or poorly understood problems within the hydrologic sciences. Section 4 does present some ideas of areas that deep learning may be applied, but a framework of how to apply DL techniques to these problems would help convince the reader of their utility.

The manuscript is generally well written, but I would recommend more attention to simplifying the verbiage and clarify the message of the paper. To help communicate this further, I included examples within the specific comments below as a guide. With thoughtful revision, I believe that this paper can serve the community well to help show the utility of deep learning techniques. I also think that the inclusion of a companion paper to explain the more technical aspects of deep learning was a very good decision.

Specific Comments: I have included some specific comments that should be revised and used as examples to help guide the authors in their overall revision. Page 2 Line 19: "deep networks are differentiable from outputs to inputs, giving them practical advantages in efficient parameter optimization via backpropagation (training)." It is not clear to me what is meant by differentiable from outputs to inputs. I believe that the concept the authors are trying to communicate here is simple, but it is not done so effectively.

Page 2 Line 23-24: "Moreover, the differentiable nature allows for greater success for interpolation and mild extrapolation, contributing to the strong generalization capability of DL." This sentence is very thick in jargon. I would suggest simplifying the verbiage to improve readability and connection to the rest of the paper.

Page 4 Line 9-10: "As a result, over time, some may have grown dispassionate about progress in machine 10 learning, and some may have concerns about whether DL is a real progress or just a "hype." While I believe that the authors do reflect the sentiment of some within the community to the promise of machine learning, the opinion paper does not present much to dispel these feelings either. In accordance with the mention

of my first point in General Comments, it would serve the paper to include some proof as to the utility of machine learning and deep learning.

Page 4 Line 14-15: "The progress brought forth by DL to the information technology industry is revolutionary (Section 4 in Shen17) and can no longer be ignored." While a companion paper should compliment this paper, this opinion manuscript should also provide evidence to the point. This could even include a small summary of findings. It is natural to have some overlap between the papers, and I believe that this suggested overlap has purpose.

Page 10 Line 20: "Observations in hydrology and water sciences..." Some would consider hydrology to be a subset of water science and others may say that hydrology and water sciences are the same field named differently. While cleaning up the language used in the manuscript, I would also suggest using either hydrology or water science. This may be a small point but is one that should be echoed through the paper.

In addition to these specific comments, I would also encourage the authors to include the various references listed by reviewers 1 and 2. I do not personally have anything to add to these references, but they would serve to present a fuller picture of machine learning applications within hydrology.

---

## Short Comment (SC2) · 20 Jun 2018

Thanks for the online comment. Here, due to time constraint, I can make a quick personal reply. Dr. Sawicz mentioned two themes: 1. "how deep learning has been integral in other fields to increase knowledge discovery of other fields of science". 2. "the presentation of a general framework of how to apply the techniques of deep learning to open or poorly understood problems within the hydrologic sciences." Both are good points and would be useful for the community. As our reply to the other reviewers, a review of how DL is applied in other sciences is provided in our companion review paper. However, it is indeed a good idea to present a good summary here in the revision,

and perhaps a couple of illuminating examples.

"The manuscript is generally well written, but I would recommend more attention to simplifying the verbiage and clarify the message of the paper. To help communicate this further, I included examples within the specific comments below as a guide. With thoughtful revision, I believe that this paper can serve the community well to help show the utility of deep learning techniques. I also think that the inclusion of a companion paper to explain the more technical aspects of deep learning was a very good decision"

Agreed. We shall make some drastic effort to simplify the paper in terms of terminology. Many hydrologists would not be able to interpret some of the language. We shall use simpler terms or make some clear definitions.

"I have included some specific comments that should be revised and used as examples to help guide the authors in their overall revision. Page 2 Line 19: "deep networks are differentiable from outputs to inputs, giving them practical advantages in efficient parameter optimization via backpropagation (training)." It is not clear to me what is meant by differentiable from outputs to inputs. I believe that the concept the authors are trying to communicate here is simple, but it is not done so effectively."

Yes this is mostly DL jargon. DL models essentially accept some inputs and propagate through a chain of matrix multiplications and nonlinear transformations. Unlike classification trees which may be discontinuous, DL models produce a mapping that is differentiable, except at final decision stage for a discrete classification problem, so a small change in inputs normally results in a small change in output. Nevertheless, we shall revise these sentences to be more interpretable.

Page 10 Line 20: "Observations in hydrology and water sciences. . ." Some would consider hydrology to be a subset of water science and others may say that hydrology and water sciences are the same field named differently. While cleaning up the language used in the manuscript, I would also suggest using either hydrology or water science. This may be a small point but is one that should be echoed through the paper."

We will stick to hydrologists for our revision.

"In addition to these specific comments, I would also encourage the authors to include the various references listed by reviewers 1 and 2. I do not personally have anything to add to these references, but they would serve to present a fuller picture of machine learning applications within hydrology"

will do. thanks.

---

## Short Comment (SC3) · 21 Jun 2018

Dear reviewers and editors,

At the point of this writing, a revision has not been requested by the editor yet. However, to embrace HESSD's spirit of online discussion and faster communication, we are uploading a document here to demonstrate the direction we would like to take in terms of revising this manuscript. Due to time constraint, the revision here is incomplete, early, and has been consented to by only a few co-authors. It will take a lot more time to work with so many authors to develop the formal revision, but the open discussion window will be closed shortly. However, we do think it might be valuable to post this

preview so that reviewers, as well as the editor, can see where the plan to take the revision. And there might potentially be more communications than what a traditional review process would have allowed.

Mainly, we re-organized the flow of the manuscript. We will put "challenges" first, in Section 3, and discuss how we can address these challenges as a community in Section 4. Please find the revision draft. We will add more examples where DL showed advantages, more examples of DL interpretation, a roadmap toward DL-powered advances in hydrologic science, and discussion of possible research directions. There will be some overlap with the companion review paper including some summaries of the content there, but there will also be new elements. We will add discussion of shared baseline models and a grass-roots revamp of education. This manuscript differs resoundingly from previous related papers. The revision will make it even more so.

Please pay more attention to the new Sections 3 and 4.

Please also note the supplement to this comment:
https://www.hydrol-earth-syst-sci-discuss.net/hess-2018-168/hess-2018-168-SC3-supplement.pdf

**Supplement:**

*At the point of this writing, a revision has not been requested by the editor yet. This document is uploaded only to demonstrate the direction we would like to take in terms of revising this manuscript. Due to time constraint, the revision here is incomplete, immature, not-grammar-checked, and has been consented to by only a few co-authors. However, it is indeed in line with the spirit of the HESS online discussion system to give a preview of where a revision could be taken. We embrace this model and*
5 *would like to communicate more than what conventional review process would have permitted.*

*Track change has been enabled. **Please pay more attention to the new Sections 3 and 4**.* ==Yellow highlight== *indicates author comments.*

**HESS Opinions: Incubating deep-learning-powered hydrologic science advances as a community**

Chaopeng Shen[1], Eric Laloy[2], Adrian Albert[3], Fi-John Chang[4], Amin Elshorbagy[5], Sangram Ganguly[6], Kuo-lin Hsu[7], Daniel Kifer[8], Zheng Fang[9], Kuai Fang[1], Dongfeng Li[9], Xiaodong Li[10], and Wen-Ping Tsai[1]

15  *1. Civil and Environmental Engineering, Pennsylvania State University, University Park, PA 16802*
   *2. Institute for Environment, Health and Safety, Belgian Nuclear Research Centre, Mol, Belgium*
   *3. Energy Technologies Area, Lawrence Berkeley National Laboratory, Berkeley, CA 94720*
   *4. Department of Bioenvironmental Systems Engineering, National Taiwan University, Taipei, 10617, Taiwan*
   *5. Dept. of Civil, Geological, and Environmental Engineering, University of Saskatchewan, Saskatoon, Canada*
20  *6. NASA Ames Research Center/ BAER Institute, Moffett Field, CA 94035*
   *7. Civil and Environmental Engineering, University of California, Irvine, Irvine, CA 92697*
   *8. Computer Science and Engineering, Pennsylvania State University, University Park, PA 16802*
   *9. Civil Engineering, University of Texas at Arlington, Arlington, TX 76013*
   *10. State Key Laboratory of Hydraulics and Mountain River Engineering, Sichuan University, Sichuan, China*

25  *Correspondence to*: Chaopeng Shen ([cshen@engr.psu.edu](mailto:cshen@engr.psu.edu))

**Abstract.** Recently, deep learning (DL) has emerged as a revolutionary and versatile tool transforming industry applications and generating new and improved capabilities for scientific discovery and model building. The adoption of DL in water science has so far been gradual, but the related fields are now ripe for breakthroughs. This paper proposes that DL-based methods can open up a viable, complementary avenue toward knowledge discovery in hydrologic sciences. In the new avenue, machine-
30  learning algorithms present competing hypotheses that are consistent with data for scientists to further evaluate. Interrogative studies are then invoked to interpret DL models. However, hydrology presents many challenges to DL-power scientific advances, such as data limitations, model diversity and variability, and the general inexperience of the hydrologic field with

DL. The roadmap toward DL-powered scientific advances will need the coordinated effort from a large community involving scientists and citizens. Integrating process-based models with DL ones will help alleviate data limitations. The sharing of data, data pipelines, and baseline models will improve the efficiency of the community as a whole. Open competitions will greatly propel growth in hydrology and. Grass-root collaboration could overcome barriers on data science education. There are a great number of research opportunities in this new area which may stimulate advances in machine learning as well.

**1. Overview**

Deep learning (DL), which has gained widespread attention since 2012, is a suite of tools centering around artfully-designed large-size artificial neural networks. Compared to non-deep networks, DL is characterized by the large size to accommodate the complexities of information contained in big data, multiple levels of hidden representations, the addition of unsupervised learning units, and effective, large-scale regularization techniques. As a foundational component of modern artificial intelligence (AI), DL has made substantial strides in recent years and helped solve problems that have resisted AI for decades (LeCun et al., 2015). DL models have repeatedly been shown to outperform simpler models by large margins and generalize better to unseen instances (Schmidhuber, 2015; Shen, 2017).

Deep networks may be more robust than simpler models despite their large size, if they are regularized properly and are chosen based on validation errors in a two-stage approach (Kawaguchi et al., 2017). Effective regularization techniques include (i) early stopping: monitor the training progress on a separate validation set and stop the training once validation metrics start to deteriorate; and/or (ii) novel regularization techniques such as dropout (Srivastava et al., 2014). DL models can be easier to train than previous networks, as their architectures and new stochastic gradient techniques (Kingma and Ba, 2014) address issues like vanishing gradient (Hochreiter, 1998). Training large networks as used today was computationally implausible until scientists started to exploit the parallel processing power of graphical processing units (GPUs). Nowadays new application-specific integrated circuits have also been created to specifically tackle DL, although DL architectures are rapidly evolving.

To be expanded:  more discussion of the attractive features of DL. Reduce jargons and improve readability. Citations

[revised manuscript text omitted]

(1) Observations in hydrology and water science, in general, are often regionally imbalanced. For example, while streamflow data are relatively dense in the United States, it is very sparse in many other parts of the world. In some parts of the world, observations have been made, but data are not made available to the public. Even for variables that can be remotely sensed, e.g., soil moisture, dense canopies often prevent uniform observations of the variable. For many hydrologic applications, there may be a dearth of observations that can be used as the supervising data. Few applications have the magnitude of data on the order of training datasets for AI tasks. A body of literature studying this problem between different geographic regions can be loosely summarized under the topic of "prediction in ungauged basins" (PUB) (Hrachowitz et al., 2013). However, PUB problems pose a significant challenge to data-driven methods.

(2) Global change is altering the hydrologic and related cycles, and hydrologists must now make predictions in anticipation of changes, beyond previously observed ranges (Wagener et al., 2010). Especially, more frequent extremes have been observed for many parts of the world and have been projected to occur in the future. Data-driven methods often face a higher chance of failure when applied out of the range of training dataset.

5 (3) Hydrologic observations also tend to be incomplete in space and time, but there are multiple sources of observations focusing on different aspects of the water cycle. For example, top 5-cm surface soil moisture only reflects a very small fraction of the water cycle, but we can also observe terrestrial water storage, which is related to soil moisture. The most prevalent observation, streamflow, integrates the signal of the whole landmass. Thus, how to merge inter-related information from different sources and to improve the prediction of each other is an important question that DL have not studied extensively.

10 (4) Compared to standard IT applications, such as speech recognition or image recognition, water data are accompanied by a large amount of strongly heterogeneous "contextual variables" such as land use, climate, geology, and soil. Heterogeneity needs to be adequately represented without radically bloating the parameter space of the models. They covary and exert complicated controls on hydrologic responses, but we have limited knowledge of some of them, especially subsurface properties like geology. There are significant uncertainties with respect to input datasets, . In addition, these heterogeneous 15 factors co-vary due to co-evolution (Troch et al., 2013), which makes it difficult for data-driven models to distinguish between causal and associative relationships. Especially, training with insufficient data may result in many alternative DL models that cannot be rejected.

(5) Hydrologic problems fit poorly into the template of problems that the standard network structures (Section 3.2 in Shen17) are designed for. While some direct applications such as soil moisture hindcasting (Fang et al., 2017) and precipitation retrieval 20 from images (Tao et al., 2016) are possible, we envision many new types of problems may require customized structures. For example, catchment hydrologic problems have both spatial but static (topography and groundwater flow) and temporal (atmospheric forcing) dimensions.

(6) Because large and diverse datasets are needed, the access to datasets, their pre-processing, and appropriate formatting present practical challenges. These steps often occupy too much unnecessary time for researchers. Many of the processing 25 tasks for images cannot be handled by a single research group. Compared to the deep learning community in AI and chemistry, etc., the machine learning in hydrology community is not sufficiently coordinated, resulting in significant waste of effort and "recreation of wheels".

(7) DL model performances can vary widely depending on model architecture, modification of network designs, training methods, use of data, data preparation, hyper-parameter setups, etc. There are a large variety of different configurations, with 30 many options beyond what could be explored by automated algorithms. Individual research groups are often limited in only exploring part of these possibilities. Thus, it is difficult to reliably reproduce reported results and learn the advantages and

disadvantages of each model design. There are also often training "tricks" that were critical in terms of achieving the desired performance.

(8) under-coordinated community

(9) Multi-point physics challenge: needs more flexible objective function

**4. A community roadmap toward DL-powered scientific advances in hydrology**

Facing the above challenges, we share the vision of a community-shared roadmap toward advancing hydrologic sciences using DL. A well-coordinated community is much more efficient and powerful in resolving the abovementioned challenges. We see that several steps are crucial in this roadmap: devising ways to integrate physical knowledge, PBMs and DL, community approaches in sharing and accessing data, open and transparent model competitions, and baseline models and visualization packages. (a Figure to illustrate the roadmap)

**4.1. Integrating physical knowledge, process-based models, and DL models**

To address data limitations mentioned in the last section (Points 1 through 3), we envision that an inevitable step is to more organically integrate hydrologic knowledge, process-based models, and deep learning. Process-based models, as they are derived to from underlying physics, require less data for calibration and can fill the gaps in different regions and for unobservable hydrologic processes. Given well-constructed, fundamentally-sound PBMs, they should also be able to represent the temporal changes and trends. However, because data-driven models directly target observations, they may have higher accuracy where data are available. Also, as discussed earlier, they are less prone to *a priori* model structural error. We should aim to maximally utilize the best features of each type of models.

This integration will undoubtedly be highly diversified as there can be many ways it can occur. Karpatne et al., (2017) compiled a list of approaches in the literature they collectively call "theory-guided data science" : (i) using knowledge to design data-driven model; (ii) using knowledge to initialize network states; (iii) using physical knowledge to construct priors to constrain the data-driven models; (iv) using knowledge-based constrained optimization (although this may be difficult to implement in practice); (v) using theory as regularization terms for the data-driven model, which will force the model to respect these constraints; (vi) learn hybrid models, where data-driven method is used as surrogate for certain part of the physical model. One may also impose multiple learning objectives based on the knowledge of the problem. d

This list can be further expanded to accommodate varied objectives. First, we can focus on PBM errors (difference between PBM simulation and observations). Non-deep machine learning has already shown promise in correcting PBM errors. *Abramowitz et al.* (2006) developed an ANN to predict the error in net ecosystem exchange from a land surface model, and achieved 95% reduction in annual error. More importantly, an ANN trained to correct the error at one biome completely

corrects the PBM for another, which is in a different temperature regime (Abramowitz et al., 2007). In the context of weather forecasts, machine learning methods were used to learn the patterns from past forecasting errors (Delle Monache et al., 2011, 2013). Then, through looking for similarity between the present situation and the past, error correction is advised, leading to 20% gain in performance (Junk et al., 2015). Their results suggest PBMs make structural errors that are independent of the state-variable regimes they operate in. We envision that PBMs can better resolve the impacts of regime changes, while DL can better capture state-independent error patterns and do mild state-dependent extrapolations. A co-benefits of modelling PBM error is insights about the PBM: if we are able to use interrogative methods to reverse engineer what DL has learned about these errors, it provides possible explanations to when and where our PBMs are wrong. It also provides clues as to how to fix these errors mechanistically. However, there lacks a theoretical framework for separately estimating aleatory uncertainty (resulting from data noise), and epistemic uncertainty (resulting from PBM error and training data paucity) and uncertainty due to regime-shift. There are significant research opportunities in this regard.

Second, PBMs can provide training data for DL models, alleviating point 1 raised in Section 3. PBMs can be used to either directly create supervising data or apply perturbations to augment existing data. Furthermore, if the DL training is limited by available data, there would be many alternative DL models that could not be rejected (point 4 in the last section). Some of these alternative DL models generate unphysical outputs. Providing PBM simulations as either training data or regularization terms help to nudge DL models to generate physically meaningful outputs.

Here, two additional ways of PBM-DL integration will be proposed.

In summary, there is substantial potential in combining the benefits of DL and PBMs. There are myriad possible approaches, yet guiding theories are lacking. On top of these alternatives, different hydrologic problems, e.g., soil moisture or streamflow, have different system properties. The advantages and disadvantages of these approaches could be systematically and efficiently evaluated in community-coordinated fashion.

**4.2. Community-coordinated hydrologic modeling competitions to pursue both performance and explainability**

As mentioned in Section 4.1 and points 6 and 7 in Section 3, there are many possible approaches and many alternative model structures. In the light of these challenges, Wwe argue that open, fast and standardized competitions are a very effective way of accelerating the progress in an area. In addition to commonly employed metrics, we can formulate competitions that are evaluated based on the attainment of understanding.

As discussed earlier, the effectiveness of competitions is best demonstrated in the community-coordinated challenges in computer science. These competitions have strongly propelled advances in artificial intelligence. New methods can be evaluated objectively and disseminated rapidly, with reduced subjectivity in the evaluation. Because the problems are standardized, they remove significant variability in terms of data sources and pre-processing. In the case of deep learning, DL models have emerged as a dominant force in almost every contest where it was applicable since 2012. Despite substantial

[revised manuscript text omitted]

**4.4. Develop a base suite of shared models, interpretation and visualization software**

To be expanded: these shared models, analogical to GoogleNet, etc., could greatly facilitate newcomers in getting started. In addition, they improve reproducibility and the effectiveness of comparisons. The interpretation and visualization effort seem fragmented and adhoc. Compiling and collecting them into community-shared resources will greatly improve our growth.

**4.5. Education**

To be expanded: a huge barrier is our educational background. There is very little preparation for big data. Grass-root collaboration could overcome barriers e data science education.

[revised manuscript text omitted]

---

## Author Comment (AC5) · 18 Jul 2018

Thanks to all reviewers and commentators for their comments. Overall, although there are some criticisms, we feel three RCs all expressed agreement with the big message to some extent. They have encouraged us, not without some reservations, to refine the message and improve content, rather than rejecting the messages. For example, in RC3, Dr. Sawicz mentioned "With thoughtful revision, I believe that this paper can serve the community well to help show the utility of deep learning techniques". In RC2, Dr. McCabe mentioned "It's my hope that the authors can consider some of my comments in adapting their opinion piece – and ultimately attract the impact such a

topic deserves". RC1 mentioned that "I would consider being more cautious about that as the understanding of the specific properties of DL models compared to more traditional statistical learning models is still an active area of research."

We agree with many of the reviewers' comments. We will do our due diligence to revise very carefully. Our earlier response in SC3 and its attachment has summarized how we would like to make the revision. Therefore, our hope is that the editor grant us the opportunity to revise.

---

## Author Response (AR1)

Dear reviewers and editors,

We have completed a thorough revision of the manuscript. Major section re-organization have taken place. New Figures and new sections have been added to address the issues. We believe the paper has improved significantly from the first version. In the following, line numbers in the curly brackets refers to the starting position of the excerpts in the Track Change document.

*AR1*
*General comments This article discusses the potential benefit of deep learning models to let emerge knowledge about water science systems from hydrological data. The paper is well written, the opinion is clearly stated and the authors present their arguments based on their expertise and their understanding of deep learning techniques. I'm wondering to what extent this is new and original compared to the opinion paper of Marçais & Dreuzy (2017 - see reference below). For example, the figure presented in this former article expressly conveys the idea that DL methods could enhance the unraveling of hydrological properties from data which is the core of this current article.*
*Marçais, J., & de Dreuzy, J. R. (2017). Prospective interest of deep learning for hydrological inference. Groundwater, 55(5), 688-692.*

The roles taken by these papers are very different. We are cheerful to see others echoing the same enthusiasm for DL in hydrology. We would welcome others to have discussion and work together on this topic. In terms of the paper, though, we see significant differences between these articles, which is summarized in the table below. In our opinion, the Marcasis and de Dreuzy 2017 (MD17) paper was a timely and welcomed first "call into the wild" (with the need to mention that one of the co-authors of this Opinion paper has had a DL paper, Tao et al. in 2016). While it is a great addition, that paper was very brief and had a different focus. It did not explain why DL could unravel hydrologic properties. It did not mention interrogative method which is a crucial part of our argument. It also did not discuss what we need to do as a community to incubate such research. In this article we gather from our past working to voice some enthusiasm as well as challenges.

However, prompted by the reviewer (and other comments), we will revise this Opinion paper significantly to emphasize our main points, which are: (1) DL+interrogative study is a valuable research avenue; (2) what challenges face the community and what we can do together to incubate DL research; (3) water resources present unique challenges and opportunities for DL.

Table. Difference between papers

| Table. Difference between papers

Paper | Unique ideas |
|---|---|
| This HESS Opinion | (As its title indicates, this is truly an opinion paper. We need to assume readers have access to Shen's review paper)
1. Opinion: DL is not a hype. Supported by a review of its solid progress, winnings of competitions and adoption in daily uses.
2. Proposition of the complementary, data-driven scientific avenue: the integration of interrogative studies into the avenue.
3. Following unique Opinions are about what we can do as a community:
a.     scientific methods: hypotheses come from machine learning. We do not pose an opinion before doing data mining. Difference from earlier ML: now we have DL to automatically extract features.
b.     call for open competition of DL in hydrology with criteria focusing on both performance and explainability
c.     collecting big data through data sharing and citizen scientists
6. Water science provide unique challenges and opportunities for DL.
7. Roadmap toward DL-supported science discovery
8. Integration model and DL
9. Education |
| Marcasis and Dreuzy 2017 | Main points: DL can be used for prediction issues; it may contribute to initial choice and alternatives of physical model structures; model reduction; emergent system properties; calibration. (however, each was only mentioned in one sentence).
Test on hydrologic numerical data; benchmarks |
| Shen. 2018 Review | 1. Technical details on ML and DL
2. Trans-disciplinary review of DL applications and experiences in sciences
3. Technical details of progress: interpreting DL and GANs
4. prospects for DL to help tackling grand challenges facing water sciences: inter-disciplinarity, human dynamics, data deluge (from novel sources), scaling and equifinality issues, non-unique inversions and high-dimensional, multi-modal data, inverse emergence |

*I also feel that DL techniques and especially why it does work so well is still not understood by computer scientists and mathematicians. However, this article can give the impression that the "DL reasons of success" are now understood (see specific comments) paving the way for knowledge discovery in water sciences through its use. I would consider being more cautious about that as the understanding of the specific properties of DL models compared to more traditional statistical learning models is still an active area of research. This does not mean that DL has not to be widely tested for hydrologic purposes.*

There have been some studies that looked at why DL is powerful, but the point is taken. We added the following paragraph

"*The fundamental theories on why DL generalizes so well have not been maturely developed (Section 2.7 in Shen18). In the ongoing debates, some argued that a large part of DL's power comes from memorization while others countered that DL prioritizes learning simple patterns* [*Arpit et al.*, 2017; *Krueger et al.*, 2017] *and a two-stage procedure (training and testing) also helped* [*Kawaguchi et al.*, 2017]. *Despite these explanations, it has been found in vision DL that deep networks can be fooled by adversarial examples where small, unperceivable perturbations to input images sometimes cause large changes in predictions, leading to incorrect outcomes* [*Szegedy et al.*, 2013; *Goodfellow et al.*, 2015]. *It remains to be seen whether such adversarial examples exist for hydrologic DL applications.*" {p8L8}

*Specific comments Page 1 L.20 Could you specify articles where DL shows capacities for scientific discovery?*

This is based on example and reasoning. It was included in the review paper Shen2018. There were quite a few examples. Here, we will include some summary of these in the revised Opinion article:

"*Given that deep networks can identify features without human guide, it follows that they may extract features that the algorithm designers were unaware of, or did not intentionally encode the network to do. If we could believe that there is latent knowledge about the hydrologic system that humans are not yet aware of, but can be determined from data, the automatic extraction of features leads to a potential pathway toward knowledge discovery. For example, deep networks recently showed that grid-like neuron response structures automatically emerge at intermediate network layers for a network trained to imitate how mammals perform navigation, providing strong support to a Nobel-winning neuroscience theory about the functioning of these structures (Banino et al., 2018).*" {p7L10}

*Page 2 L.9-16. The paragraph gives the impression that DL is a "plug and play" model whereas to my knowledge building a DL model still requires intensive computer scientists' knowledges and requires use of GPUs.*

Good point. We added the following sentence to this sentence:

"*While showing many advantages, DL models will require substantial amount of computing expertise. The tuning of hyper-parameters, e.g. network size, learning rate, batch size, etc., often require a priori experiences and trial and error. The computational paradigm, e.g., computing on graphical processing units, is also substantially different from ordinary hydrologists' educational background.*" {P8L3}

*Page 2 L.24. I don't think that generalization capacities of DL come from its interpolation capability. Indeed, classical neural networks have been proven (see citation below) to be universal interpolators but they do not generalize well. Hornik, K., Stinchcombe, M., & White, H. (1989). Multilayer feedforward networks are universal approximators. Neural networks, 2(5), 359-366. Page 3 L.25. I agree that increase in environmental data opens new opportunities for data-driven techniques in general and particularly for DL techniques. Along with the development of spatialized, remote sensing data, I would also insist on the development of environmental observatories that collect a lot of time series, monitoring data even though they are site specific. These two types of data are complementary to advance through knowledge discovery in hydrology.*

The original sentence was "*Moreover, the differentiable nature allows for greater success for interpolation and mild extrapolation, contributing to the strong generalization capability of DL.*". The differentiability "contributes" to the generalization ability, but that is not the sole reason. Other factors include improved architecture, regularization, big data, weights sharing etc., which were mentioned earlier. To avoid any confusion, this sentence has been revised as

"*Compared to earlier models like classification trees, most of the deep networks are differentiable, meaning that we can calculate derivatives of outputs with respect to inputs or the parameters in the network. This feature enables highly efficient training algorithms that exploit these derivatives.*" {P6L24}

*Page 6. L.6-17. This paragraph is intended to bridge the link between interrogative techniques brought in DL by the "AI neuroscience" subdiscipline and the potential of DL for knowledge discovery in water sciences. If the arguments tend to prove that such interrogative techniques enlighten the way the architecture of DL works, it does not explain the success of DL in itself. For example, the sentence*

*L.13: "activations of recurrent neural networks can be visualized to show the control domain of certain cells, which explains its functioning" is not correct. This only explains the functioning of the architecture of the DL, not the reason of success of such a method. There is some literature exploring the need for explanation of DL techniques. For Convolutional Neural Networks (CNNs), their understanding can be linked with wavelet theory (see reference below). Especially their capacity to extract invariants through a lot of different scale in high dimensional datasets but this is still a subject of active research. This capacity could explain their generalization capabilities especially for image datasets.*

The interpretive study does not solely focus on "why DL was successful", and this is not really the point. The interpretive studies answer "what has DL learned".

The revision has separated the two questions. At the end of Section 2.2, we mention present theories about "why DL generalizes so well". The whole 2.3 explains "how to extract knowledge from deep networks". Thus there should not be confusion now.

*Mallat, S. (2016). Understanding deep convolutional networks. Phil. Trans. R. Soc. A, 374(2065), 20150203. Page 7 L.8-18. It could be interesting to explore how DL techniques can improve hypothesis testing through an exploration of competing process-based models? The*

*Structure for Unifying Multiple Modeling Alternatives (SUMMA) (see reference below) could be a start to generate process-based models with alternative hypotheses. For example, process-based models could be used to feed DL models with numerical generated data.*
*Clark, M. P., Nijssen, B., Lundquist, J. D., Kavetski, D., Rupp, D. E., Woods, R. A., ... & Arnold, J. R. (2015). A unified approach for process-based hydrologic modeling: 1. ˘ Modeling concept. Water Resources Research, 51(4), 2498-2514*

The reviewer is spot-on that the SUMMA framework is relevant and could be integrated with DL models in several ways. In fact it has been cited in Shen18 in a more relevant reference to equifinality.

Here, the new Section 4.1 mentions this

"*Second, PBMs can augment input data for DL models. PBMs can be used to increase supervising data for DLs, for example, for climate or land-use scenarios that have not existed presently, to augment existing data. Given model structural uncertainty (uncertainty with hydrologic processes), frameworks like the Structure for Unifying Multiple Modeling Alternatives (Clark et al., 2015) could be employed to generate a range of inputs.*" {P16L23}

*Page 10 L.20-Page 11 L.24. I would add to this list the fact that water sciences provide to DL a unique challenge because hydrologic data are intrinsically heterogeneous. Building a model able to integrate these heterogeneous data might be the key toward knowledge discovery in water sciences and toward big progresses in AI.*

Good point! This part has been explained to include this point:

"*Hydrologic data are accompanied by a large amount of strongly heterogeneous [Blöschl, 2006] "contextual variables" such as land use, climate, geology, and soil properties. The proper scale at which to represent heterogeneity in natural systems is a vexing problem [Archfield et al., 2015], as micro-scale of soil heterogeneity, for example, is not computationally realistic in hydrologic models. The scale at which heterogeneity should be represented varies with setting and elements of the water cycle [Ajami et al., 2016]. Moreover, while we recognize that heterogeneity exists in contextual features, many of these features, such as soil properties and hydrogeology, are poorly characterized across landscapes, but both features play important role in controlling water movement. Heterogeneity needs to be adequately represented without radically bloating the parameter space of the models. Furthermore, the heterogeneous physiographic factors covary [Troch et al., 2013] and exert complicated causal and non-causal connections, but we have limited knowledge of their covariation. Consequently, training with insufficient data may result in many alternative DL models that cannot be rejected.*" {P13L28}

*AR3 McCabe*
*Review of Shen et al. (2018) "HESS Opinions: Deep learning as a promising avenue towards knowledge discovery in water sciences" Overview The opinion article by Shen et al. (2018) aims to provide a perspective on the opportunities that deep learning may provide to the water*

*sciences discipline. This is certainly a topic of much current interest and relevance to the community, as it offers possible new pathways for system interpretation and understanding. As such, I was very keen to read and review this contribution, with hopes of 1) learning more about deep learning applications in the "water sciences", and 2) identify some practical outcomes that could be relevant to my own (relatively broad) research interests.*

*Since I was not previously aware of this type of "Opinions" forum in HESS, I was expecting something more akin to a "Review Article", where the advances in machine learning (being delivered in this case by deep learning) would be illustrated through some relevant applications and examples. As such, I was a little disappointed that this was not the intent of this paper. The paper is precisely as the title dictates: an opinion article. Having, and expressing, an opinion is great: but for it to appear as a published article, it should ideally be supported by a strong, reasoned and defensible position that counters competing arguments via illustration of its superiority (or at least equivalence).*

Many thanks to Dr. McCabe who gave very constructive criticism. Indeed this is an opinion paper which does not normally assume the role of a full review. Of course, some concise arguments have now been provided, but the function of a full review paper has been achieved in another open-access paper on arxiv.

We have taken elements (in the form of summaries and abridged examples) from the review paper Shen18 to support the arguments. In addition, this paper also has the important task of discussing what the community has to do as a whole

*Deep learning may be (and I believe it is) "a promising tool toward knowledge discovery in [the] water sciences". But simply stating it and illustrating with some examples where it has worked before is not the way to convince a new audience*
*What is presented is a brief description of deep learning, a rather concise historical review of "machine learning" applications in hydrology (e.g. SVM, CART, RFs; which actually have quite an extensive history in hydrology, and especially remote sensing that could be detailed further), an expression of the need for data-driven science in contrast to a more classical (physics-based) approach, and an overview of some of the unique challenges that the water sciences present (which do not seem particularly unique if posed across the earth sciences). However, none of the expressed opinions are particularly revolutionary ideas: hydrology (and related fields) already provides many examples of data-driven science, black-box modeling applications, and novel statistical approaches to divine process insights. What would be good to see is how deep learning transcends these, or at the least, builds upon them to provide an avenue for new insights and investigation into "hydrological" processes.*

With respect to convincing people of the power of DL, the re-organized manuscript added the following:

[revised manuscript text omitted]

These characteristics leads to a new plausible way of doing science as we are advocating here.

"*Given that deep networks can identify features without guide, it follows that they may extract features that the algorithm designers were unaware of, or did not intentionally encode the network to do, leading to a potential pathway toward knowledge discovery. For example, deep networks recently showed that grid-like neuron response structures automatically emerge at intermediate network layers for a network trained to imitate how mammals perform navigation, providing strong support to a Nobel-winning neuroscience theory about the functioning of these structures (Banino et al., 2018)*" {P7L6}

We do not think interrogative studies was part of the historical studies or even an aspect that was ever argued for in hydrology. These studies were motivated by the criticism that DL models are black boxes, and the computer science community and the general scientific community only recently have worked together to develop these interpretive methods.

*Overall, I think there is a missed opportunity here to provide a perspective that could potentially garner significant interest in the community. To do this, the authors could expand on a possible road-map on future directions (and obstacles) for deep learning applications, and also provide a demonstration of some analogous examples (perhaps from other disciplines, if not from hydrology directly) that could be relevant to "water science" applications. It's my hope that the authors can consider some of my comments in adapting their opinion piece – and ultimately attract the impact such a topic deserves*
*Comments (in no specific order of importance or logical sequence).*

We think that the reviewer's suggestion is excellent and have adopted the suggestion. The major re-organization in the revision should have been a significant improvement. The new Section 4 and a new Figure expands on the roadmap.

[Figure]

Figure 4. A roadmap toward DL-powered scientific discovery in hydrologic science. Data availability can be increased by (green arrows) collecting and compiling existing data, incorporate novel data sources such as those collected by citizen scientists, remote sensing and modelled dataset. DL can be employed to predict data that are currently difficult to observe. The modelling competitions and the integration between PBM and DL will build important shared computing and analytic infrastructure, which, together

with data sources, support a wide range of hydrologic applications. Interpretive methods should be attempted to extract knowledge from trained deep networks (orange arrows). Underpinning these activities is the enhanced, community-based educational program for machine learning in hydrology (purple arrows). However, these activities, especially the modelling competitions, might in turn feedback to the educational activity.

*\* The title is very broad, with "water sciences" encapsulating a wide range of possible research avenues. I guess this is fine, as I agree that deep learning has broad application, but I wonder whether it might help to focus this discussion on "hydrological" sciences instead, and illustrate with some demonstrations of where this approach might deliver upon its potential. If the title is retained, it would need a much broader description of approaches and applications that could be explored. The authors might wish to review the recent work of Marcais and de Dreuzy (2017), who present a brief introduction to deep learning, focused on some more specific applications (calibration, hypothesis testing, etc.).*

The title has been changed to "*HESS Opinions: Incubating deep-learning-powered hydrologic science advances as a community*"

Marcasis and de Dreuzy will be added to the literature review. As described in our reply to reviewer #1, it is a brief "call into the wild".

*\* I would remove the repeated statement (see line 24 as an example) of ". . .we lay out several opinions shared by the authors". In fact, I'd remove the use of "opinions" throughout the manuscript (Pg3-L4; Pg3-L9; Pg8-L25; Pg11-L26 etc.) completely and just focus on the presentation of ideas. As an alternative, use instead "Here we propose. . .". However, it is assumed that all co-authors are in agreement with the content of the paper, so there's no need to remind the reader of this.*

This sentence has been removed.

*\* The five points listed in the abstract lean a little towards motherhood statements. Some specificity here would be great. Outlining what "may" happen seems a bit counterproductive. If this is a strongly held "opinion", this should be reflected in the content of the paper. For instance, "Deep learning will revolutionize our understanding of XYZ. . ." or "Deep learning offers an entirely new approach to ABC. . .". At the least, these statements need to be supported throughout the manuscript by a clear and ratio nal review of how (precisely) deep learning will deliver upon them. Point 4 is probably the most important here, and the manuscript could really be built up around this (a point I will discuss below). I do not really understand Point 5 i.e. we need hydrologycustomized methods for interpreting knowledge provided by deep learning? Isn't one of the points of deep learning to provide new knowledge for interpreting hydrological processes? Are you suggesting that it can do this, but we aren't able to understand it? Perhaps it's just me, but I find this a bit confusing.*

We will revised our abstract with more clearly-defined statements.

In terms of the hydrology-customized interpretation method:

According to our summary of Shen 2017's review paper, there are several methods that have already been developed for standard image recognition problems (relevance backpropagation, approximation using interpretable models, and correlation-based analysis). These methods can be

ported to water applications. However, scientists in other domains have been creative in devising problem-specific ways in interpreting the results. We believe water scientists need to do both. Especially, considering we will also need customized network structure and our applications will be diverse, some of the method will not work out of the box. Therefore, we expect customized interpretive methods to be necessary and will be an active area of research. Nevertheless, this sentence has been removed due to total re-writing of the abstract.

*Regarding Point 4. To me, this represents the key issue that much of the paper can be built around. Deep learning has potential, but there are some specific challenges that hydrological sciences present that need to overcome or addressed. These are detailed somewhat in Section 4, but so much more could be written and the ideas expanded upon. For deep learning to have an impact in "water science", it is precisely issues like these (and this list is not comprehensive) that need to be considered. It would be great if you could structure your paper to examine these in more detail (if not provide possible solutions or avenues to address them). At the moment, the paper basically says that deep learning is a great technique that has much potential to provide new insights and understanding – BUT – there are some pretty serious roadblocks and challenges (not unique) to hydrological sciences that need to be addressed first. It's a big "but", especially if no attempt to provide a pathway to addressing them is offered. The real value of this opinion piece could be to provide some roadmaps towards these. At the least, a number of the "questions" presented in this section can be examined in greater detail, with examples drawn from the existing literature to showcase earlier or preliminary efforts.*

This is a wonderful suggestion. We believe the manuscript would benefit from such a re-structuring. As indicated by the previous comment, we are in the process of revising the manuscript and the revised structure is:

**3. Challenges and opportunities**

**4. A community roadmap toward DL-powered scientific advances in hydrology**

**4.1. Integrating physical knowledge, process-based models, and DL models**

**4.2. Multi-faced, community-coordinated hydrologic modelling competitions**

**4.3. Community-shared resources and broader involvement**

**4.4. Education**

*The paper could really use a review of the structure combined with a sharper focus on the deep learning applications (to hydrology/water science) in general. The entire Overview section reads as a Deep Learning review, rather than an exploration of its application to water sciences. Section 2.2. could probably be incorporated into the Overview/Introductory section instead of standing alone. Further, while an introduction to the technical concept is certainly required (and also needs attention), there's not much in the way of expounding on knowledge discovery. Just as illustrating some examples in other disciplines is relevant and required, so too is exploring those applications already examined in the "water sciences" through some recent literature (see your own listed examples on Pg3-L3 as well as on Pg5-L22-28). Providing some brief review of these applications may serve to demonstrate the value of your*

*opinion. There are also quite a few others (see Agana and Homaifar, 2017 10.1109/SECON.2017.7925314)*

As summarized in Shen 2017, there really are not a great deal of water applications for DL, and hence this paper. Those have been collected into that review paper. Here we have added a paragraph as a summary:

"*In contrast, only a handful of applications of big data DL could be found in hydrology, but they already demonstrated great promise. Vision DL has been employed to retrieve precipitation from satellite images, where exihibited a materially-superior performance than earlier-generation neural networks (Tao et al., 2017, 2018). GAN was used to imitate and generate scanning images of geologic media (Laloy et al., 2018), where the authors showed realistic replication of training image patterns. Time-series deep learning network was employed to temporally extend satellite-sensed soil moisture observations (Fang et al., 2017) and was found to be more reliable than simpler methods. Regionalized time series DL rainfall-runoff models have been created (Kratzert et al., 2018). There are also DL studies, based on smaller dataset, to help predict water flows in the urban environment (Assem et al., 2017) and water infrastructure (Zhang et al., 2018). In addition to utilizing big data, DL was able to create valuable, big datasets that could not have been otherwise possible. For example, utilizing DL, researchers were able to generate new datasets for Tropical Cyclones, Atmospheric Rivers and Weather Fronts (Liu et al., 2016; Matsuoka et al., 2017) by tracking them. Machine learning has also been harnessed to tackled the convection parameterization issue in climate modelling (Gentine et al., 2018).*" {P6L6}

Again please note this paper moves beyond the review: we raise several opinions on how, together as a community, tackle the obstacles. Selling DL is not the only focus of this piece.

*\* Following this point, the companion paper of Shen (2017), purports to provide a more comprehensive technical background (it is not listed in the bibliography). I was able to find this on arxiv (https://arxiv.org/ftp/arxiv/papers/1712/1712.02162.pdf) with the title "A trans-disciplinary review of deep learning research for water resources scientists". While only skimming that paper, I can see that it addresses many of my criticisms of this manuscript, in that it provides the needed level of technical background, disciplinary context and demonstration via examples that I was hoping for. The obvious question then is what additional value this manuscript offers in light of that work? I will leave it up to the authors (and editor) to make that assessment [but in the same vein, the EOS article by Shen, 2018, https://doi.org/10.1029/2018EO095649 seems another example of an opinion article on this topic?).*

Please see the table in the reply to AR1 about the differences between these two papers. The review paper provides basics, literature review, and discussion of the hydrologic science challenges that DL could help address. The current Opinions paper focuses on the complementary scientific avenue, the challenges facing application of DL, and what we can do as a community to tackle these challenges. This paper is more forward looking and it is, as titled, an opinion paper. We agree this should be made more clear in the abstract and in the paper. On the other hand, the EOS article, constrained by 600 words limit, was to serve as an opener of a special issue in WRR. It stresses the argument that DL may one day become part of hydrology. There was barely any discussion in that one so it should not really compared.

*\* Page 2, Line 15-16. This sentence is unclear to me*
*\* Some of the short-comings of GANS should also be mentioned: especially their ability to be "easily fooled" (see https://arxiv.org/pdf/1801.00553.pdf, https://arxiv.org/abs/1710.09762 and many other similar papers). Are there implications to water sciences in this – especially for automated approaches used in prediction systems? What other drawbacks of deep learning may impair their uptake and development?*

It is not that GANs are easily fooled, but deep networks can be fooled by small changes in inputs. GANs can actually be used to train networks more robustly, which is one of the co-authors focus of research [Ororbia].

We added the following

"*Despite these explanations, it has been found in vision DL that deep networks can be fooled by adversarial examples where small, unperceivable perturbations to input images sometimes cause large changes in predictions, leading to incorrect outcomes (Goodfellow et al., 2015; Szegedy et al., 2013). It remains to be seen whether such adversarial examples exist for hydrologic DL applications. If we can recreate adversarial examples, they can be added into the training dataset to improve the robustness of the model (Ororbia et al., 2016).*" {P8L9}

*\* Other papers that might be of interest to the authors (indeed, see Volume 55, Issue 5 of Groundwater):*

*Chen and Wang (2018) "Recent advance in earth observation big data for hydrology" https://doi.org/10.1080/20964471.2018.1435072 Frere (2017) "Revisiting the Relationship Between Data, Models, and Decision­A˘R-˘Making" https://doi.org/10.1111/gwat.12574 Lary et al. (2016) "Machine learning in geosciences and remote sensing" https://doi.org/10.1016/j.gsf.2015.07.003 Marshall (2017) "Creativity, Uncertainty, and Automated Model Building", https://doi.org/10.1111/gwat.12552 Lidard et al. (2017) "Scaling, similarity, and the fourth paradigm for hydrology", https://doi.org/10.5194/hess-21-3701-2017 Anderson (2008) "The end of theory: the data deluge makes the scientific method obsolete" https://www.wired. com/2008/06/pb-theory/ McCabe et al. (2017) "The future of Earth observation in hydrology", https://doi.org/10.5194/hess-21-3879-2017*

*\* Since I'm familiar with that last reference, I highlight some of the discussion therein on machine learning approaches in general, particularly on Page 3902 (n.b. it may also be worth reviewing some of the mentioned references in an attempt to provide context of machine learning based hydrological applications - and where deep learning will fit into that): "Despite this remarkable confluence of data science and remote sensing, one can still resist the narrative that there is no problem that a sufficiently complex machine-learning algorithm cannot unravel given enough data (Anderson, 2008). If this were the case, there would be no need for domain expertise to understand current and future challenges in hydrology: the dilettante will have prevailed (Klemeš, 1986). Indeed, there remain several obstacles to any predicted ascension of a completely data-driven approach to hydrology. Observations of the hydrosphere often have a spatio-temporal structure that emerges in the form of correlations between variables, but this correlation may not necessarily imply causality. Therefore, being able to draw strong deterministic conclusions about the behaviour of hydrologic systems based on data-driven methods often requires prior knowledge (and understanding) of the physical processes (Faghmous and Kumar, 2014)." This is relevant to your Section 2.4 and elsewhere*

Some of these points indeed resonate with what we have put forth in old Section 4 where we mentioned PBMs and DL models complement each other. We in fact argue in the revised paper that DL alone is not going to "do it" in hydrology, in Section 3. This is not from a personal interest point of view, but we do not see water science presents so much data that can cover every aspect of the hydrologic and human water cycle. If we humor ourselves and imagine that such a scenario does occur, it may indeed be possible for DL models to predict everything more accurately than process-based models, but still PBMs are required for us, humans, to understand the causal relationships. Even data in the world may be not be sufficient to distinguish between causal and associative relationships.

Below are our revised statements. We have included citations suggested by the reviewer.

"*Observations in hydrology and water science generally are regionally and temporally imbalanced*." {P13L8}

"*Global change is altering the hydrologic and related cycles, and hydrologists must now make predictions in anticipation of changes, beyond previously observed ranges*" {P13L18}

"*Observations of the water cycle tend to focus on one aspect of the water cycle, and seldom offer a complete description*." {P13L23}

"*Furthermore, the heterogeneous physiographic factors co-evolve and covary (Troch et al., 2013) with complicated causal and non-causal connections (Faghmous and Kumar, 2014). The relationships of soil, terrain and vegetation are further conditioned on geologic and climate history and often do not transfer to other regions (Thompson et al., 2006). Consequently, training with insufficient data may result in overfitted data-driven models or many alternative DL models that cannot be rejected. On the flip side, such complexity due to co-evolution also nearly precludes a reductionist approach where all or most of these relationships are clearly described from fundamental laws*" {P14L6}

As a solution to these issues, we actually promote integrating process-based models and the data-driven one, as described in the New Section 4.1:

"***4.1. Integrating physical knowledge, process-based models, and DL models***

*To address the challenge of data limitations (data quantity), we envision that a critical and necessary step is to more organically integrate hydrologic knowledge, process-based models, and DL. Process-based models, as they are derived from underlying physics, require less data for calibration than data-driven models. They can provide estimates for spatial and temporal data gaps and unobserved hydrologic processes. Well-constructed PBMs should also be able to represent temporal changes and trends. However, because data-driven models directly target observations, these models may have better performance in locations and periods where data are available. Also, as discussed earlier, data-driven models are less prone to a priori model structural error than are PBMs. We should aim to maximally utilize the best features of each type of models.*

*….*

*"*

*\* **Your Section 4 provides an excellent launching point to really expand on some of these ideas and challenges (see above), and I would encourage you to use these (and build upon them) to***

*structure this opinion piece around. Of course, it should be recognized that the problems highlighted here are not particular to deep learning, but to hydrological inference and understanding broadly, and that there has been much effort directed towards novel statistical approaches to address some of these (which would be worth mentioning, or at least providing some context)*

Thanks. We have expanded this section into two much bigger sections, the new Section 3 and 4.

*\* I'm not convinced that Section 2.4 is essential to this paper – or at least it can be presented differently. Advocating the role of data-driven approaches is not a new concept in hydrology (see some of the papers above for reviews) – nor is it especially controversial. It is not like modelers act in isolation – data is an integral part of that process. As with the use of machine learning approaches, data-driven knowledge discovery has a rich history in hydrology, which may be worth reviewing. Certainly there are many examples of ANN type models outperforming their physically-based counterparts. But I'm not sure what the intent of this section is? Either way, it is also not immediately clear (or demonstrated) that deep learning offers a better path towards achieving this "goal" than the myriad of techniques already being used.*

As mentioned previously, we do not believe interrogative studies were mentioned before, as they are essentially a new sub-discipline. Therefore, while we will take most of the suggestions offered by the reviewer, here we beg to differ that this section is important. We believe the new section 2.4 is .

*\* Likewise, I'm not sure what the purpose of Section 3.2 is? The last paragraph in particular (Pg10-L6-15) invokes a lot of hand-waving*

This is actually one of the roadmap toward collecting more data. Citizen scientists can help collect data about precipitation, groundwater depths and pressure, surface water stages, soil texture and other observable variables. Previously it was not possible to consume lot of the sub-research-grade data they produce. Now, with lots of data, the noise inherent in these data can be averaged out. With DL, we can infer concepts not possible before.

*There are a number of other questions I have and handwritten annotations I have made on the paper that are not included in this review. My overall impression is that the paper needs some considered thought not just on its structure, but on how it attempts to present the "opinion" that deep learning is a promising tool in hydrology. While I'm an advocate of your perspective here, in reading the manuscript, I found little to convince me that this approach presents a radical new angle to anything that has come before it. I hope that the authors can address some of these comments and further refine the contribution, as I think it is a topic that will be of considerable interest to the community.*

Again we thank Dr. McCabe for constructive criticism. I think the main focus may be a little different from what Dr. McCabe had anticipated. Hopefully the revised version has more clarity and usefulness.

Reviewer #3. Dr. Sawicz

*The manuscript titled "Deep learning as a promising avenue toward knowledge discover in water sciences" conveys the opinion that hydrology is a field well suited for deep learning and that deep learning techniques should be widely applied to increase our understanding of hydrologic systems. While I agree with the authors' general premise and believe the article can have a great impact on the direction of the technical analysis of future hydrological studies, I also was disappointed to not see two primary topics. The first of which is a summary of how deep learning has been integral in other fields to increase knowledge discovery of other fields of science. The second and more important topic would be the presentation of a general framework of how to apply the techniques of deep learning to open or poorly understood problems within the hydrologic sciences. Section 4 does present some ideas of areas that deep learning may be applied, but a framework of how to apply DL techniques to these problems would help convince the reader of their utility*

*The manuscript is generally well written, but I would recommend more attention to simplifying the verbiage and clarify the message of the paper. To help communicate this further, I included examples within the specific comments below as a guide. With thoughtful revision, I believe that this paper can serve the community well to help show the utility of deep learning techniques. I also think that the inclusion of a companion paper to explain the more technical aspects of deep learning was a very good decision*

Agreed. We have made effort to simplify the paper in terms of terminology. We shall use simpler terms or make some clear definitions. Examples are given below as response to detailed comments.

*Specific Comments: I have included some specific comments that should be revised and used as examples to help guide the authors in their overall revision. Page 2 Line 19: "deep networks are differentiable from outputs to inputs, giving them practical advantages in efficient parameter optimization via backpropagation (training)." It is not clear to me what is meant by differentiable from outputs to inputs. I believe that the concept the authors are trying to communicate here is simple, but it is not done so effectively*

*Page 2 Line 23-24: "Moreover, the differentiable nature allows for greater success for interpolation and mild extrapolation, contributing to the strong generalization capability of DL." This sentence is very thick in jargon. I would suggest simplifying the verbiage to improve readability and connection to the rest of the paper*

*"Compared to earlier models like classification trees, most of the deep networks are differentiable, meaning that we can calculate derivatives of outputs with respect to inputs or the parameters in the network."* {P6L24} We believe this sentence works better.

For another example, the main technical parts are now simplified:

*"Underpinning the powerful performance of DL are its technical advances. The deep architectures have several distinctive advantages: (1) deep networks are designed with the capacity to represent extremely complex functions. (2) After training, the intermediate layers can perform modular functions which can be migrated to other tasks, in a process called transfer learning, and extend the value of the training*

*data. (3) The hidden layer structures have been designed to automatically extract features, which helps dramatically reducing labour, expertise and the trial and error time needed for feature engineering. (4) Compared to earlier models like classification trees, most of the deep networks are differentiable, meaning that we can calculate derivatives of outputs with respect to inputs or the parameters in the network. This feature enables highly efficient training algorithms that exploit these derivatives. Moreover, the differentiability of neural networks enables querying DL models for sensitivity analysis of outputs to input parameters, a task of key importance in hydrology."* {P6L20}

*"Some recent progress in DL research focused on addressing these concerns. Notably, a new sub-discipline, known as "AI neuroscience" has produced useful interrogative techniques to help scientists interpret the DL model (see literature in Section 3.2 in Shen18). The main classes of interpretive methods include (i) reverse-engineer the hidden layers: attributing deep network decisions to input features or a subset of inputs; (ii) transferring knowledge from deep networks to interpretable, reduced-order models. (iii) visualization of network activations. Many scientists have also devised case-by-case adhoc methods, e.g., to investigate the correlation between inputs and cell activations (Shen, 2018; Voosen, 2017)."* {P8L25}

***Page 4 Line 9-10: "As a result, over time, some may have grown dispassionate about progress in machine 10 learning, and some may have concerns about whether DL is a real progress or just a "hype." While I believe that the authors do reflect the sentiment of some within the community to the promise of machine learning, the opinion paper does not present much to dispel these feelings either. In accordance with the mention of my first point in General Comments, it would serve the paper to include some proof as to the utility of machine learning and deep learning***

***Page 4 Line 14-15: "The progress brought forth by DL to the information technology industry is revolutionary (Section 4 in Shen17) and can no longer be ignored." While a companion paper should compliment this paper, this opinion manuscript should also provide evidence to the point. This could even include a small summary of findings. It is natural to have some overlap between the papers, and I believe that this suggested overlap has purpose.***

The revised Section 2.2 largely fills this role. This section begins with industry uptake, then goes into uptake of DL in computer science and non-computer science research and then list computer science competitions which are now dominated by DL.

We think that some statistics may be convincing in terms of how useful DL is. "*Major technology firms have rapidly adopted and commercialized DL-powered AI (Evans et al., 2018). For example, Google has re-oriented its research priority from "mobile-first" to "AI-first" (Dignan, 2018). The benefits of these industrial investments can now be felt by ordinary users of their services such as machine translation and digital assistants who can engage in conversations sounding like a human (Leviathan and Matias, 2018). Moreover, AI patents of industries and scientific disciplines grew at a 34% compound annual growth rate between 2013 and 2017, apparently after DL's breakthroughs in 2012 (Columbus, 2018). Also reported in (Columbus, 2018), more than 65% of data professionals responded to a survey indicating AI as their company's most significant data initiative for next year.*" {P4L13}

"*The fast growth is clearly witnessed from literature searches. Since 2011, the number of entries with DL as a topic increased almost exponentially, showing around 100% compound annual growth rate before 2017 (Table 1). DL evolved from occupying less than 1% of machine learning (ML) entries in computer science (CS) in 2011 to 46% in 2017. This change showcases massive conversion from traditional machine learning to DL within computer science. Other disciplines lagged slightly behind, but also experienced exponential increase. They also saw the DL/ML ratio jumping from 0% in 2011 to 33% in 2017.*" {P4L20}

Then we gave some concrete examples, but more examples are given in Shen18.

"*As reviewed in Shen18, DL has enhanced the statistical power of data in high energy physics, and the use of DL can be considered to be equivalent to a 25% increase in the experimental dataset (Baldi et al., 2015). In biology, DL has been used to predict potential pathological implications from genetic sequences (Angermueller et al., 2016). DL models in computational fed with raw-level data have been shown to outperform those using expert-defined features when they predict high-level outcomes, e.g., toxicity, from molecular compositions (Goh et al., 2017).*" {P4L15}

*Page 10 Line 20: "Observations in hydrology and water sciences. . ." Some would consider hydrology to be a subset of water science and others may say that hydrology and water sciences are the same field named differently. While cleaning up the language used in the manuscript, I would also suggest using either hydrology or water science. This may be a small point but is one that should be echoed through the paper.*

Yes. now we stick to "hydrology".

*In addition to these specific comments, I would also encourage the authors to include the various references listed by reviewers 1 and 2. I do not personally have anything to add to these references, but they would serve to present a fuller picture of machine learning applications within hydrology.*

We have included many of those references. Thanks.

---

## Author Response (AR2)

Dear editor,

Many thanks to you for your comments and handling.  We have addressed the minor comments. In addition, we have gone through the manuscript a final round between several authors and fixed some grammar issues.

Regarding your points

Some minor comments:
- In Figure 3, panels a and b are not mentioned in the caption and it is unclear to what extent panel b is really necessary to understand the points made in the caption.

We modified Figure 3 caption as the following. It should be clear from panel (b) that RockDepth become the criterion.

[Figure]

**Figure 1. (adapted from Fang and Shen 2017. Reprint permission obtained). We calculated storage-streamflow correlation patterns over contiguous United States (CONUS) and divided small or mesoscale basins into multiple classes. We studied what physical factors most cleanly separate different correlation patterns. Panel (a) shows the distribution of basin classes on and south to the Appalachian ranges. Panel (b) shows a one-level CART model that finds the most effective criterion to split basins into clusters with different distances to class #1 (storage and streamflow are highly correlated across all flow regimes) correlation patterns. This CART model does not achieve complete separation of class #1 basins because it is based on criteria that could be established using input attributes, and there is also resolution limitation with terrestrial water storage observations, which are from the GRACE satellite. However, on a large spatial scale, what separates the blue class and the green class (low correlation between storage and high flow bands) turned out to be soil thickness. It suggests the blue basins in the south have high correlation *because* they have thick soils, which facilitates infiltration, water storage, and increases the importance of groundwater-contributed streamflow.**

- Some readers might argue that artificial neural networks are a form of DL that has been around for decades (hundreds of papers) and that this part of the history of DL in hydrology might merit a little more acknowledgment than the brief reference in section 2.2.1..?

True. We added the following sentence

"*There are especially numerous hydrologic applications of ANNs including modeling of rainfall-runoff, groundwater, water quality, salinity, and rainfall estimation, and soil parameterization (Govindaraju, 2000; Maier et al., 2010) (see a brief discussion in Appendix A in Shen18).*" {page 4 line 2-4}

- It is interesting that you use a classic table (Table 1) to highlight the growth of DL papers rather than a visualisation (minor point)!

We have deliberated over this question, and tried presenting the information using a figure. The issue with this table is that there are six columns, and it will lead to quite a messy figure. Also, there is a huge contrast in numbers (from 0 to 6125) so that log-scale would be involved. But by the end of the day, in a more philosophical paper like this we hoped to reduce the strain on the reader. Plus the numbers are very clear and readers will perhaps be able to run some back-of-envelop calculations with these numbers. Therefore, while we highly respect your suggestion here, we think it might be ok to keep it as a table. Again, thanks for chipping in the opinion.

[revised manuscript text omitted]